# Carbon-Based Nanofluids and Their Advances towards Heat Transfer Applications—A Review

**DOI:** 10.3390/nano11061628

**Published:** 2021-06-21

**Authors:** Naser Ali, Ammar M. Bahman, Nawaf F. Aljuwayhel, Shikha A. Ebrahim, Sayantan Mukherjee, Ali Alsayegh

**Affiliations:** 1Nanotechnology and Advanced Materials Program, Energy and Building Research Center, Kuwait Institute for Scientific Research, Safat 13109, Kuwait; nmali@kisr.edu.kw; 2Mechanical Engineering Department, College of Engineering and Petroleum, Kuwait University, P.O. Box 5969, Safat 13060, Kuwait; a.bahman@ku.edu.kw (A.M.B.); shikha.ebrahim@ku.edu.kw (S.A.E.); 3Thermal Research Laboratory (TRL), School of Mechanical Engineering, Kalinga Institute of Industrial Technology, Bhubaneswar, Odisha 751024, India; 1881148@kiit.ac.in; 4School of Aerospace, Transport and Manufacturing (SATM), Cranfield University, Cranfield MK43 0AL, UK; a.alsayegh@cranfield.ac.uk

**Keywords:** carbon nanotubes, graphene, nanodiamond, parabolic trough solar collector, nuclear reactor, air conditioning and refrigeration

## Abstract

Nanofluids have opened the doors towards the enhancement of many of today’s existing thermal applications performance. This is because these advanced working fluids exhibit exceptional thermophysical properties, and thus making them excellent candidates for replacing conventional working fluids. On the other hand, nanomaterials of carbon-base were proven throughout the literature to have the highest thermal conductivity among all other types of nanoscaled materials. Therefore, when these materials are homogeneously dispersed in a base fluid, the resulting suspension will theoretically attain orders of magnitude higher effective thermal conductivity than its counterpart. Despite this fact, there are still some challenges that are associated with these types of fluids. The main obstacle is the dispersion stability of the nanomaterials, which can lead the attractive properties of the nanofluid to degrade with time, up to the point where they lose their effectiveness. For such reason, this work has been devoted towards providing a systematic review on nanofluids of carbon-base, precisely; carbon nanotubes, graphene, and nanodiamonds, and their employment in thermal systems commonly used in the energy sectors. Firstly, this work reviews the synthesis approaches of the carbon-based feedstock. Then, it explains the different nanofluids fabrication methods. The dispersion stability is also discussed in terms of measuring techniques, enhancement methods, and its effect on the suspension thermophysical properties. The study summarizes the development in the correlations used to predict the thermophysical properties of the dispersion. Furthermore, it assesses the influence of these advanced working fluids on parabolic trough solar collectors, nuclear reactor systems, and air conditioning and refrigeration systems. Lastly, the current gap in scientific knowledge is provided to set up future research directions.

## 1. Introduction

Since the 20th century, scientists have been working with considerable effort to develop fluids that can surpass those conventionally known by the scientific society and industry in terms of thermal and physical performance. The idea of dispersing solid particles of millimeter (mm) and micrometer (µm) in size is the milestone, which was physically initiated by Ahuja [1,2] in 1975, Liu et al. [3] in 1988, and other researchers at Argonne National Laboratory (ANL) [4,5,6] in 1992 on the bases of Maxwell theoretical work [7]. Such suspensions have shown tremendous improvements in heat transfer characteristics compared to their base fluids. This is due to the dispersed solid particles’ significantly higher thermal conductivity compared to their hosting fluid, which would enhance the effective thermal conductivity of the colloidal. The term ‘effective’ is generally used when referring to the net property of a solid–liquid suspension [8]. However, it was found that in flow areas of low velocities, the particles hosted by the suspension tended to deposit from its carrier liquid. Additionally, hence the fluid starts to lose its tuned properties. Furthermore, clogging of small passages was also experienced due to the significant level of agglomeration between the dispersed particles, and therefore making it extremely challenging to employ in heat transfer devices containing small channels. This is when, in 1993, Masuda et al. [9] conceived the idea of fabricating suspensions with ultrafine particles of silica, alumina, and titanium dioxide, where these dispersions were afterward given the name ‘Nanofluids’ by Choi and Eastman [10], in 1995, as a result of their extensive research work at ANL. According to the founders, a nanofluid can be generally defined as an advanced category of fluid that is produced by homogeneously dispersing low concentrations (preferably ≤1 vol. %) of particles of less than 100 nanometers (nm) in size within a non-dissolving base fluid [11]. Both Masuda et al.’s [9] and Choi and Eastman’s [10] primary motivation at that time was to overcome the limitations associated with suspensions made by their counterparts (i.e., colloidal containing millimeter or micrometer sized particles). In addition, Choi and Eastman [10] have theoretically known beforehand that reducing the size of the dispersed particles to the nanoscale would greatly enlarge the particle exposed surface area to the surrounding, and thus increasing the suspension overall thermal conductivity [12]. The significant variation in thermal conductivity between solid particles and liquids can be clearly seen in Figure 1 for some of the most commonly used particles and base fluids, at room temperature and atmospheric pressure, for fabricating nanofluids [13,14,15,16,17]. It is worth noticing that CuO, MgO, Al_2_O_3_, ZnO, TiO_2_, Fe_2_O_3_, SiO_2_, Ag, Cu, Au, Al, Fe, carbon nanotubes (CNTs), and multiwalled carbon nanotubes (MWCNTs) stands for cupric oxide, magnesium oxide, aluminum oxide, zinc oxide, titanium dioxide, iron(III) oxide, silicon dioxide, silver, copper, gold, aluminum, iron, carbon nanotubes, and multiwalled carbon nanotubes, respectively. Furthermore, the thermal conductivity of some of the materials shown in Figure 1 was seen to have a significant scatter of data across the literature, which can be linked to several factors such as the purity, crystallinity, particle size, and the determination approach used to find this thermal property. In addition, the thermal conductivity of graphene after being subjected to oxidization (i.e., having the form of graphene oxide) gets highly reduced, where it can reach values between 1000 and 2 W/m·K [18,19,20].

Following their success, many researchers started to explore and develop this class of engineered fluid via modifying their production route, enhancing the suspension stability, and improving the colloidal thermal conductivity [13,21,22]. As of today, nanofluids are seen to have potential usage in a wide range of areas, including the energy sector, construction and building, transportation, oil and gas, medical sector, etc. [23,24,25,26,27,28,29,30,31,32,33,34]. Figure 2a shows the increasing trend in scientific publications in the field of nanofluids from 1995 to 2020, while Figure 2b illustrates the different types of these published documents that are available in the same database. It is worth mentioning that the data in Figure 2 was obtained from Elsevier’s abstract and citation database, Scopus, via searching through the word ‘Nanofluid’ [35].

Despite the promising achievements that nanofluids could deliver to the scientific community, there are still some obstacles that need to be overcome before this category of fluids can be industrially accepted. For example, the colloidal preparation phase is still considered one of the most significant challenges, as this stage can strongly influence the fluid physical stability and effective thermophysical properties [13,36]. Meaning that if the fabrication process used was not well structured before being executed, the chances of an unstable nanofluid being produced is likely to occur. As a result, some of the suspension’s thermophysical properties will gradually degrade with time due to the separation of particles from the hosting base fluid. Almurtaji et al. [37] have illustrated in their published work the relationship between the effective thermal conductivity and the physical stability of suspensions. They showed that the effective thermal conductivity of a nanofluid could reach its optimum possible value when the dispersion is physically stable, and vice versa. In addition, the commonly employed two-step fabrication method that relies on an ultrasonic bath type device, was reported to raise the as-prepared nanofluid temperature and that the surrounding atmospheric conditions govern this increase in temperature along with the sonicator working power. Thus, it is highly unlikely that similar nanofluids can be produced through the conventional two-step route without simultaneously fabricating the products at the same preparation conditions. A more convenient two-step method employed for nanofluid production would be the two-step controlled sonicator bath temperature approach, as was reported by Ali et al. [8,11] and Song et al. [38]. The aforementioned approach would eliminate the rise in bath temperature obstacle, and hence will ensure an optimum level of nanofluids reproducibility to the manufacturer at any surrounding atmospheric conditions, and even when using different types of bath sonicators. Furthermore, as the thermal properties of a nanofluid are influenced mainly by the dispersed particles compared to its base fluid, researchers have been focusing more on carbon-based materials. This is because some of these materials, in the nanoscale, have exceptional thermophysical properties compared to other commonly used materials (e.g., metals and oxides) [39,40,41]. For instance, CNTs and graphene have significantly elevated thermal conductivity [42,43], large aspect ratio [44], lower density [45,46], lower erosion and corrosion surface effects [47], higher stability [43], and lower pressure drop and pumping power requirement in comparison to other types of nanomaterials [48,49]. Figure 3 demonstrates common allotropes of carbon nanomaterials.

Many published numerical and experimental studies on nanofluids fabricated with particles of carbon-based materials were found in the literature, which show the continued growth of interest in such materials [35,51,52,53,54]. Figure 4 classifies these documents in terms of the number of available publications at the Scopus database for each type of carbon-based material used in nanofluids production. The single-walled carbon nanotube (SWCNT) and double-walled carbon nanotube (DWCNT) abbreviations in Figure 4 refer to the single-walled carbon nanotube and double-walled carbon nanotube, respectively. During the reviewing process, which led to the formation of Figure 4, the authors remarkably recognized that the researchers had used different sonication duration and intensities to fabricate their nanofluids. However, some of the suspensions had the same particles type, size, and hosting base fluid. This shows that, up to today, there is no standard fabrication method for the production of the colloidal. The authors have also found that dispersing carbon-based materials, such as walled carbon nanotubes (MCNTs) and graphene, can tremendously enhance the quality of biofuels blends, in specific biodiesel [55,56]. This includes lowering the brake specific fuel consumption, stabilizing the fuel consumption rate and brake thermal efficiency, and improving the diesel engine performance and the resulting emissions from the combustion process.

This review paper provides an overview of three types of carbon-based nanofluids: CNT, nanodiamond (ND), and graphene. The selection reason for these three carbon-based particles is due to their outstanding thermal properties compared to any other sort of nanoscaled solids. Hence, they can be considered promising candidates for fabricating nanofluids targeted towards heat transfer applications. The main contribution of the present review study is that this work starts from the synthesis stage of these three carbon-based materials, followed by their dispersed form, and up to their employment in selected energy applications. Furthermore, recommendations on the different nanofluids production methods used are shown along with the colloidal stability and its effect on the thermophysical properties. Moreover, the experimental measuring devices and theoretical equations used to determine and predict the thermophysical properties are provided. In addition, the research work done on utilizing these carbon-based suspensions are presented for three thermal applications, namely, parabolic trough solar collectors (PTSCs), nuclear reactors, and air conditioning and refrigeration (AC&R) systems, with a comparison to those of conventional working fluids. Finally, the gaps in present scientific knowledge that scientists need to tackle are highlighted in order to promote these advanced types of heat transfer fluids commercially.

## 2. Synthesis of Nanoscaled Carbon-Based Materials

Carbon ranks as the 4th most common element after hydrogen, helium, and oxygen in our solar system, and the 17th in the crust of our planet [57]. Remarkably, this element is distinctive so that when the crystal structure of carbon atoms is changed into deferent arrangements, the material properties significantly differ [58,59,60,61,62,63]. For example, both ND and graphene are made of carbon but of different atomic bounds arrangement. While the first is an electrical isolator and transparent towards visible light waves, the second has excellent electrical conductivity with complete visible light blockage. Such materials that contain various arrangements of carbon atoms are known as ‘allotropes of carbon’, which means that the material has chemically identical elements but with different atomic arrangements, and hence different physical properties. Due to this fact, many allotropes of carbon exist or have been discovered by scientists, e.g., diamond, graphene, and CNTs. The following Section 2.1, Section 2.2 and Section 2.3. will provide a short overview of the fabrication of three allotropes of carbon in the nanoscale, namely ND, graphene, and CNTs. Knowing the production methods of these materials is essential and will, later on, help us understand which nanofluid fabrication route is suitable to conduct.

### 2.1. Nanodiamonds

NDs have existed for billions of years in nature within meteorites, crude oil, interstellar dust protoplanetary nebulae, and different sediment layers of the Earth’s crust. Nevertheless, the synthetization process of this valuable material only started in the second half of the nineteenth century through either exposing graphite to high pressure and high temperature conditions, or by the explosive detonation of bulk graphite [64,65,66]. The first is known as the high-pressure and high-temperature (HPHT) approach, whereas the second route is known as the detonation technique. In the literature, it was reported that the first study conducted on the preparation of NDs was performed by Bovenkerk et al. [67], in 1959, after which Danilenko [68] used the detonation technique as part of his synthesis approach. Furthermore, many approaches were developed afterward for fabricating ND, such as the microplasma-assisted formation [69], chemical vapor deposition (CVD) method [70], laser ablation [71], high energy ball milling of microdiamonds produced from high pressure and high temperature conditions [72], high energy ball milling of ultra-fine graphite powder [73,74], ultrasound cavitation [75], chlorination of carbides [76], carbon onions irradiated by electron [77], and irradiation of graphite by ion beam [78]. In addition to the previous synthesizing methods, El-Eskandarany has proposed a novel approach for producing superfine NDs from commercial graphite powders and SWCNTs under ambient temperature and atmospheric pressure conditions, using a high-energy ball mill technique [79]. It is important to note that, according to Ali et al. [66] and Mochalin et al. [80], the most common types of NDs seen today are the detonation NDs (DNDs) and the HPHT-NDs. From the aforementioned production routes, it can be concluded that the synthesized NDs can only be produced as independent solid particles, and therefore cannot be grown within liquids through chemical and/or physical approaches. Regardless of the method used, the production of NDs usually involves three major phases, which are 1—synthesis (methods mentioned earlier), 2—processing, and 3—modification. The processing stage, which follows the synthesis phase, enhances the as-produced NDs purity by removing the metals and metals oxides along with the non-diamond carbons that remain attached to the ND surface. Hence, a high level of sp^3^ carbon bonded diamond nanoparticles can be obtained. This can be done by using oxidants such as nitric acid (HNO_3_), perchloric acid (HCLO_4_), or hydrochloric acid (HCL) [81]. Furthermore, the modification phase is essential so that the fabricated NDs can meet the requirements of their targeted application. Modification can be performed using either surface functionalization (widely used) or doping of the NDs particles. It is important to note that some researchers have recently started focusing on the doping technique due to the distinct optical properties gained from this NDs modification approach [82,83]. Figure 5 shows the three phases involved in the production of NDs [84].

### 2.2. Graphene

Graphene is a type of carbon material that originates from bulk graphite. It has the shape of a 2-dimensional (2D) (i.e., monolayer) sheet of one-atom thickness and lattice of hexagonally arranged sp^2^ bonded carbon atoms [85]. The material itself was successfully synthesized for the first time in 2004 by Novoselov et al. [86], through mechanical exfoliating graphite with Scotch tape. Furthermore, the development in the field has resulted in categorizing graphene by the materials architecture structure, which ranges from zero-dimensional (0D) graphene quantum dots, one-dimensional (1D) graphene fibers and nanoribbons, and 2D graphene nanomesh, rippled/wrinkled and multisheet [87]. Figure 6 shows an illustration of the different categories of graphene based on their dimensionality and bandgap opening. Regarding 2D graphene sheets, few suitable techniques are commonly employed for producing such material, which are mechanical exfoliation [86], sublimation of silicon carbide (SiC) [88], laser-induced graphene [89,90], covalent [91,92] or non-covalent [93] exfoliation of graphite in liquids, and CVD growth [94]. These fabrication methods produce graphene in a solid form except for the liquid-phase exfoliation, which delivers the material as part of a suspension.

The mechanical exfoliation method was the first approach for obtaining graphene. In this method, the small mesas of highly oriented pyrolytic graphite are repeatedly peeled out with a Scotch tape, and hence the attached thin films on the tape are of monolayer graphene. This production route is highly reliable and allowed the preparation of high-quality graphene sheets of up to 100 µm in thickness [86]. Other less common types of mechanical exfoliation are also available, such as ball milling of graphite nanoparticles [95] and hammering graphite [96]. Furthermore, the high temperature sublimation of SiC, which was developed initially for the electronics industry, relies on the thermal decomposition of a SiC substrate via either an electron beam or resistive heating to epitaxial graphene under ultrahigh vacuum condition. This results in the desorption of the silicon (Si) on the wafer surface, and therefore causing the surface atoms to arrange into forming hexagonal lattice. Moreover, fabricating graphene through laser-inducement is performed under ambient atmosphere by subjecting carbon dioxide (CO_2_) pulsed laser to a substrate containing carbon-based materials. This approach combines 3-dimensional (3D) graphene fabrication and patterning into a single step without having to use wet chemical steps. In addition, exfoliation of graphite in liquids or liquid-phase exfoliation depends on the employment of external peeling force, such as an ultrasonic horn sonicator, to separate the graphene sheets from the immersed bulk graphite in a solvent of suitable surface tension. The solvent used in the process is usually a non-aqueous solution, such as N-methyl-2-pyrrolidone (NMP), but aqueous solutions can also be employed if surfactant was added. It is important to note that the yield of the liquid-phase exfoliation process is relatively low, and thus centrifugation is used to gain a significant fraction of monolayer and few-layer graphene flakes in the final dispersion [97]. On the other hand, the CVD production route uses hydrocarbon gases to grow graphene on a targeted substrate by carbon diffusion and segregation of high carbon solubility metallic substrates, such as nickel (Ni), or by surface adsorption of low carbon solubility metals (e.g., Cu) [98,99]. From all of the previous methods, CVD has shown to be the most successful, promising, and feasible approach in the field for producing monolayer graphene of high quality and large area [94]. For deeper insight into the various graphene synthesis methods, the reader is referred to the published work of Rao et al. [100].

### 2.3. Carbon Nanotubes

Although carbon is known as a ubiquitous material in nature, CNTs are not, where this allotrope material is a human-made seamless cylindrical form of carbon. It is believed that the oldest CNTs existed on damascene swords [101]. Still, their first proof of presence was in 1952 through the transmission electron microscopy (TEM) images published by Radushkevich and Lukyanovich [102], after which Boehm [103] and Oberlin et al. [104] obtained similar images along with describing the currently widely accepted CNTs growth model. Conceptually, CNTs are graphene sheets rolled into cylindrical tubes, of less than 1 nm in diameter, with a half fullerenes caped end. Based on the number of consistent tubes (i.e., rolled-up graphene sheets), CNTs can be classified as SWCNTs, DWCNTs, and MWCNTs. As the terms suggest, the SWCNTs consist of only one tube, whereas DWCNTs and MWCNTs comprise two and three (or more) tubes, respectively [105,106]. Figure 7 shows the mechanism in which CNTs are formed and their three different types. It is important to note that some researchers distinguished between the three tubes form of CNTs and those of a higher number of tubes, where they have categorized the first as the triple-walled carbon nanotubes (TWCNTs) and the second as MWCNTs [107,108].

There are three main synthesis methods for producing CNTs, which are the arc discharge, laser ablation, and CVD [109,110,111]. Other approaches, such as diffusion and premised flame method, can be used for CNTs fabrication but are less frequently utilized [112]. All three primary production methods depend on the carbon feedstock, either as a solid phased carbon source (arc discharge and laser ablation) or carbonaceous gases (CVD method). An example of the gases employed in the CVD process include carbon monoxide (CO), ethanol, and acetylene. Moreover, the final product is always delivered in a dried form; thus, CNTs cannot be grown within liquids as dispersions. In the arc discharge process, doped graphite rods or two catalysts loaded are vaporized at 4000–5000 K, within a closed chamber, by an electric arc placed between them, after which the resulting deposit is of CNTs. Like the arc discharge method, the laser ablation route relies on the evaporation of a carbon feedstock, usually a graphite rod with a metallic based catalyst, to obtain the CNTs. The difference between this approach and the previous one is that the laser ablation uses high energy laser irradiation to heat the carbon source, and thus causing the phase transformation (i.e., from the solid to gaseous phase). Additionally, the final product gets accumulated in a cold trap located within the chamber. Therefore, this technique is much more efficient than the arc discharge process in terms of the losses in the as-produced CNTs. On the other hand, the CVD, which was mentioned earlier in Section 2.2, decomposes carbonaceous gases on catalytic nanoparticles to produce the CNTs. The catalytic nanoparticles used for this purpose are either grown while conducting the process or are initially fabricated through a separate procedure. Furthermore, the advantage associated with this production technique is the high level of control over the synthesis process parameters such as carbon supply rate, growth temperature, catalyst particles size, and type of substrate used for the CNTs growth.

## 3. Preparation of Nanofluids

Nanofluids can be formed by dispersion particles made of single elements (e.g., Cu and Fe), single element oxides (e.g., CuO and Al_2_O_3_), alloys (e.g., stainless steel), metal carbides (e.g., silicon carbide and zirconium carbide), metal nitrides (e.g., silicon nitride and titanium nitride), or carbon-based materials in a none dissolving base fluid such as water, methanol, glycol, ethylene glycol (EG), transformer oil, kerosene, and/or different types of refrigerants with or without the use of surfactant/s [13,113]. The nanosuspension is given the name ‘nanofluid’ when one type of nanoparticles is used in the fabrication process; in contrast to the previous category, dispersions formed by employing two or more types of nanoparticles are classified as ‘hybrid nanofluids’ [114,115]. To the best of the authors knowledge, unlike the previous two nanofluids categories that are subjected to the number of different particles used in the process, there does not exist a specific classification for nanofluids made of more than one type of base fluid. However, researchers could have used the terms ‘Bi-liquid nanofluid’ or ‘Tri-liquid nanofluid’ to refer to their nanofluid that is made from two or three base fluids, respectively. Figure 8 shows an illustration of the conventional nanofluid and the hybrid nanofluid. In addition, the homogeneity and physical stability of the dispersion depend significantly on the implemented preparation approach, which can substantially influence the effective thermophysical properties of the as-prepared suspension. Knowing the aforementioned is essential when selecting the appropriate type of nanofluid for any targeted application [116]. In general, two known fabrication processes are currently used for producing nanofluids, namely, the one-step (also referred to as the single-step) method and the two-step approach [37]. It is important to note that some researchers prefer to classify the one-step production processes into two categories, which are the one-step physical technique and the one-step chemical approach, resulting in three types of methods of nanofluid fabrication for these groups [117,118]. A summary of the two fabrication schemes (i.e., the one-step and two-step methods) is presented in the following subsections.

### 3.1. One-Step Method

The production of nanofluids by the one-step method is conducted by simultaneous synthesizing and dispersing the nanoparticles in the base fluid. Thus, the storage, drying, and transportation of nanoparticles are unnecessary [119]. Furthermore, the dispersed particles in this bottom-up process avoid oxidization from their surrounding environment. In addition, this technique is well known to highly eliminate clustering and agglomeration of dispersed particles within the hosting fluid, and hence coagulation of nanoparticles in real-life applications that uses microchannels can be minimized with an increase in the level of the physical stability of the colloidal compared to the two-step production approach. Moreover, this method allows greater control over the size and shape of the dispersed nanoparticles during the fabrication process. Nevertheless, the presence of residual reactants as a result of uncompleted reactions has always been a major drawback of such a production route. Other disadvantages can also be experienced when following the single-step synthesis approach, such as the inconsistency of the scale for industrial applications, which can only be used with base liquids of low pressure, high production cost, and limitation in the types of nanofluids that can be fabricated compared to the two-step route [120,121,122,123].

One of the most common one-step approaches is the one that was established by Eastman et al. [21]. In this method, nanofluids are synthesized by evaporating a bulk material, after which the evaporated particles get deposited then condensed in a thin film of base fluid attached to a vessel wall due to centrifugation. Figure 9 demonstrates the aforementioned one-step approach. Many researchers have continuously worked on developing the one-step fabrication approach through physical and/or chemical means. Today, different methods have been acknowledged to be in the one-step nanofluid production category [36,120,124]. Figure 10 shows some of the commonly known one-step nanofluid fabrication routes in the field where their method of conduct can be found fully explained in the published work of Ali et al. [13] and Mukherjee et al. [36].

### 3.2. Two-Step Method

Unlike the one-step method, the two-step approach is a top-down process that uses dried nanoparticles that were initially prepared, through physical or chemical processes, after which these particles get dispersed in a base fluid through ultrasonic agitation [125,126,127,128], magnetic stirring [129,130,131,132], homogenizing [131,133,134], or ball milling (least commonly used) [16,135,136] with or without adding surfactant(s) to the mixture. Other less common dispersion routes can also be used, such as dissolver, kneader, three roller mill, stirred media mill, and disc mill [137]. Figure 11 demonstrates an example of the two-step method, where a bath type ultrasonic device is used to form the suspension. In addition to the bath type sonicators, some researchers have employed the probe/horn type sonicators to fabricate their nanofluids. They have reported higher particles dispersion capability and enhanced suspensions thermal properties using this type of device compared to the bath type dispersers [138]. The reason behind the previously achieved improvements in the suspension is that the probe device provides focused and intense ultrasonication effects, reaching up to 20 kW/L, to the mixture in an evenly distributed manner [139]. This is something that the bath type sonicators cannot provide due to its low relative intensity (i.e., 20–40 W/L) and non-uniform distribution of the ultrasonication effect on the fabricated nanofluid. It is important to note that the bath type ultrasonicator is more applicable for commercial scale production of nanofluids. In contrast, the probe type is better suited for synthesis at the lab scale. Regardless of the type of two-step mixing approach used, this method is still considered as a cost-effective process that is appropriate for both small- and large-scale production of any type of nanofluids, which is seen as a favorable approach to many researchers in the field [140]. However, some of the critical issues associated with this method during nanofluids fabrication are the agglomeration of the nanoparticles due to the very high surface energy between the particles, and the notable increase in the process temperature with fabrication time when using some of the mixing devices (e.g., bath type ultrasonic device) [8,13]. The first obstacle causes the suspension to be in a weak physical stability state that results from the nanoparticles undergoing agglomeration, which is followed by separation of the particles from the base fluid in the form of sediments. Thus, the nanofluid thermophysical properties degrade with time. As for the raise in fabrication process temperature problem, the reproducibility of similar nanofluids (i.e., obtaining suspensions with the same thermophysical properties) would be impossible to achieve. This is because different bath type ultrasonic devices and/or surrounding atmospheric conditions lead to varying the thermophysical properties and physical stability of the fabricated colloidal [8,38]. There are several ways to overcome the aforementioned limitations in the two-step method. For example, surfactants can be added to the mixture to reduce the level of particles agglomeration, and the sonicator bath temperature could be controlled throughout the fabrication process by equipping the device with a temperature regulator. Other approaches used to physically stabilize the as-prepared dispersions are mentioned afterward in the nanofluid stability enhancement section (Section 4.2). When preparing nanofluids, the nanoparticles and surfactants (if required) are added to the base fluid with respect to either volume (vol.) or weight (wt.) percentage (%). Most researchers tend to use the vol. % to calculate the added nanopowder to the base fluid, which can be estimated through the appropriate formulae presented in Table 1.

Where V, m, ρ, np, np1, np2, bf, bf1, and bf2 represent the volume, mass, density, single type of nanoparticles, first type of nanoparticles, second type of nanoparticles, single type of base fluid, first type of base fluid, and second type of base fluid, respectively. In addition to the equations shown in Table 1, one can use the following three equations to determine the vol. % for their nanofluids when having two different particles and/or two base fluids concentration ratio(s).

For single type of nanoparticles and two different types of base fluids:(5)$$\frac{{\left(\frac{m}{\rho }\right)}_{np}\;}{{\left(\frac{m}{\rho }\right)}_{np}+\left[{\left(\frac{m}{\rho }\right)}_{bf1}\times \frac{A}{A+B}+{\left(\frac{m}{\rho }\right)}_{bf2}\times \frac{B}{A+B}\right]\;}\times 100$$
where the ratio of bf1:bf2 is equal to A:B.

For two different types of nanoparticles and single type of base fluid:(6)$$\frac{{\left(\frac{m}{\rho }\right)}_{np1}\times \frac{A}{A+B}+{\left(\frac{m}{\rho }\right)}_{np2}\times \frac{B}{A+B}\;}{\left[{\left(\frac{m}{\rho }\right)}_{np1}\times \frac{C}{C+D}+{\left(\frac{m}{\rho }\right)}_{np2}\times \frac{D}{C+D}\right]+{\left(\frac{m}{\rho }\right)}_{bf}}\times 100$$
where the ratio of np1:np2 is equal to C:D.

For two different types of nanoparticles and two types of base fluid:(7)$$\frac{{\left(\frac{m}{\rho }\right)}_{np1}+{\left(\frac{m}{\rho }\right)}_{np2}\;}{\left[{\left(\frac{m}{\rho }\right)}_{np1}\times \frac{C}{C+D}+{\left(\frac{m}{\rho }\right)}_{np2}\times \frac{D}{C+D}\right]+\left[{\left(\frac{m}{\rho }\right)}_{bf1}\times \frac{A}{A+B}+{\left(\frac{m}{\rho }\right)}_{bf2}\times \frac{B}{A+B}\right]\;}\times 100$$
where the ratio of np1:np2 and bf1:bf2 are equal to C:D and A:B, respectively.

### 3.3. Carbon-Based Nanofluids Fabrication

As was explained previously in Section 2, carbon allotropes, whether ND, graphene, or CNT, have their own production routes and final product form. For instance, it was shown that both NDs and CNTs could only be produced in the form of dried particles, whereas graphene can be fabricated as dried sheets or as part of a dispersion. Therefore, depending on the type of nanoscaled carbon allotrope and base fluid desired for synthesizing the nanofluid, the production process can be constrained by only the two-step method or the manufacturer can be left with the freedom of selecting any of the two approaches. In general, the two-step method is the only approach that can be employed for fabricating dispersions containing NDs or CNTs, while both one- and two-step routes can be used for producing graphene nanofluids. Nevertheless, the majority of the studies have shown the adaptation of the two-step method for producing graphene nanofluids, which can be justified by the difficulties associated with the single-step route of fabrication and the limitations in the type of base fluid that can be used (see Section 3.1) [145,146]. Some of the research work published on fabricating NDs, graphene, and CNTs nanofluids using the two-step method are listed in Table 2. Note that the single-step graphene nanofluid production was excluded from Table 2 because it is precisely the same as liquid-phase exfoliation of graphene; thus, the reader can find further information’s within the sources provided previously in Section 2.2 and the work published by Texter [147]. Nevertheless, it is worth mentioning that the common base fluids used in the graphene suspension one-step (or liquid-phase exfoliation) approach are n-methyl-2-pyrrolidone (NMP), γ-butyrolactone (GBL), n,n-dimethylacetamide (DMAC), n,n-dimethylformamide (DMF), dimethylsulfoxide (DMSO), ortho-dichlorobenzene (ODCB), acetonitrile (ACN), and water with the aid of surfactant [148].

## 4. Nanofluids Stability

### 4.1. Stability Mechanism and Evaluation

The stability of nanofluids is of major concern for maintaining the thermophysical properties of the mixture [169]. Specifically, the stability of the suspension combines several aspects such as dispersion stability, kinetic stability, and chemical stability [120,170]. The dispersion stability deals with nanoparticles aggregation within the colloidal, while the kinetic stability describes the Brownian motion of nanoparticles hosted by the base fluid (i.e., sedimentation of randomly agglomerated particles due to gravity). As for the chemical stability, it is associated with the chemical reactions that occur between the nanoparticles themselves and between the nanoparticles and the surrounding base fluid. However, it is essential to note that chemical reactions in a nanofluid are minimized or halted at low temperature conditions (i.e., below the temperature point of a chemical reaction). Hence, agglomeration and sedimentation of nanoparticles would be the primary aspects concerned with suspension stability. When a nanofluid is physically unstable, the formed sedimentation can have one of three behaviors, namely; 1—dispersed sedimentation, 2—flocculated sedimentation, or 3—mixed sedimentation [8]. Figure 12 shows a schematic illustration of the realistic reflection for the three types of sedimentation behaviors. In addition, the speed at which the sediment forms and settles within an unstable suspension can be classified into two main regions. The first is known as the rapid settling region, which occurs at the beginning stage of the separation of the particles from the hosting base fluid; and the following stage is called the slow settling region, where the changes in sediment formation and settling becomes insignificant along the shelving lifetime [171]. Figure 13 demonstrates an example of the two sedimentation speed formation regions from Witharana et al. [171] investigation. Furthermore, there are about eight techniques that can be used to evaluate the stability of nanofluids, such as 1—sedimentation photographical capturing method, 2—dynamic light scattering (DLS) approach, 3—zeta potential analysis, 4—3-ω approach, 5—scanning electron microscopy (SEM) analysis, 6—TEM characterization, 7—spectral analysis, and 8—centrifugation method. From the previous stability evaluation methods, the sedimentation photographical capturing approach is considered as the most reliable route between them all, but at the expense of time (i.e., it takes a very long time to conduct and analyze). The DLS approach usually over-predicts the size of the particles, especially when using a non-ionized base fluid (e.g., deionized water), where the analysis can show larger values (from 2 to 10 nm more) than the actual particle size [172]. Such results are very problematic and misleading when analyzing nanofluids, especially when the dispersed particles are 10 nm or less in size, where the oversized prediction can incorrectly indicate an instability state. On the other hand, the zeta potential analysis should only be used as a supportive characterization tool. This is because if the nanoparticles and/or the base fluid are non-polar or even of low polarity, there may be other mechanisms affecting the suspension stability [172]. Thus, it is highly recommended to use multiple approaches (e.g., three methods) to determine the stability of the nanofluid. A detailed description of each of the experimental stability evaluation approaches, and their advantages and limitations can be found in the work published by Ali et al. [13]. Other than the previous stability evaluation approaches, Carrillo-Berdugo et al. [173] have proposed a novel theory-based design framework for determining the polarity between the solid and liquid interface, which can be used to adjust the interface tension by adding the required number of dispersive components to meet those of the dispersed nanomaterial.

### 4.2. Stability Enhancements

Several approaches have been shown to improve the stability of nanofluids successfully. These methods are subdivided into two main categories, which are in the form of physical and chemical routes. The physical approach involves the employment of high energy forces such as ultrasonication, magnetic stirring, homogenizer (or probe sonicator), or even ball milling, which is rarely reported [117,175]. Figure 14 shows the four previous physical stability methods. Unlike the ultrasonication and homogenization methods, the magnetic stirring approach is considered as the most basic route that can be applied to break-down clusters of nanoparticles, within the suspension, with very low performance effectiveness when compared to the other two physical methods [176]. Furthermore, in the literature [177], high pressure homogenization was shown to provide better stability characteristics than ultrasonication to the as-produced nanofluids. In addition, the mixing duration and intensity used in the sonicator device were commonly seen to vary from one research work to another in an attempt to physically stabilize the nanofluid. A good explanation for the aforementioned method is that the mixing power cannot be maintained constant throughout the process due to the voltage fluctuation that the device experienced. Therefore, Yu et al. [178] suggested relying on the relation between the suspension absorption spectra against the total energy supplied to the mixture as a relative solution to the sonication time.

On the other hand, the chemical route stabilizes the suspension by declustering the agglomerated nanoparticles by alternating the pH value of the base fluid or the mixture, adding surfactant(s) to the solid–liquid matrix, or modifying the surface of the nanoparticles. Nanofluids pH alteration affects the level of free cations or anions charges in the media surrounding the dispersed particles, and hence the hydrophilicity or hydrophobicity nature of the particles changes causing the colloidal to either stabilize or destabilize [179,180]. The disadvantage of the previous method is that fabricating suspensions of high or low pH values may be corrosive for high heat flux applications. In addition, surfactants are essential when dispersing nanomaterials of hydrophobic nature (e.g., CNTs and graphene) in a polar base fluid (e.g., water), and vice versa [181,182]. This is because the added surfactant would act as a bridge between the nanoparticles and the hosting fluid, and therefore would improve the dispersion stability of the particles through increasing the repulsive force between the particles themselves and reducing the interfacial tension between the base fluid and the hosted particles. Surfactants are categorized based on their head group charge as cationic, non-ionic, anionic, and amphoteric. Table 3 shows some of the surfactants used in the nanofluids preparation process according to their head group charge [118].

The downside from using surfactants is that the nanofluid becomes more viscous; starts to generate foam when being heated or cooled down; can be lost at high temperatures, and would reduce the overall thermal conductivity of the suspension. As for the nanoparticles surface modification technique, the particles are either initially functionalized (before the dispersion process), or the functionalized materials themselves are added to the colloidal (where they get grafted to the surface of the segregated particles), and therefore forming a new particle surface exposure to the hosting base fluid [183,184]. The drawback of using functionalized materials as stabilizers is that they tend to reduce the overall thermal conductivity of the produced nanofluid due to having a significantly lower thermal conductivity than the dispersed nanoparticles. Figure 15 recaps all of the nanofluid stability improvement methods that were mentioned earlier in this section.

## 5. Stability Effect on Thermophysical Properties

The thermophysical properties govern the heat transfer rate that the nanofluid can provide to the system in which it is employed as a working fluid. Nanofluids thermal properties, such as the thermal conductivity, greatly depend on the type of base fluid, nanoparticles material, morphological characteristics of the particles, nanoparticles concentration, and homogeneity of nanoparticles dispersion in the hosting base fluid. The dispersion characteristics of the suspension are subjected to alteration with the change in stability of the particles in their surrounding environment (i.e., base fluid). For such reason, the stability of a nanofluid is considered as a significant factor to maintain the heat transfer rate from and to the colloidal. This section covers the influence of stability on nanofluids effective thermal conductivity and effective viscosity. It is important to highlight that the effect of suspension stability, as a parameter, on the effective density was not reported across the literature, but rather the added surfactants and particles concentration were seen responsible for the changes caused to nanofluids densities [185,186,187]. This is because nanofluids effective density ($\rho_{nf}$) is constrained by its overall volume and mass, where it can be directly calculated from extending the rule of mixtures (i.e., Equation (8)):(8)$$\rho_{nf}={\;f}_{V}\times \rho_{np}+\left(1-{\;f}_{V}\right)\times \rho_{bf}$$
where $f_{V}$ is the particles volumetric fraction, $\rho_{np}$ is the density of the nanoparticles, and $\rho_{bf}$ is the density of the base fluid. Similarly, the effective specific heat capacity of the colloidal was not shown to be linked to the dispersion stability. The main parameter that affects nanofluids effective specific heat capacity is the particles concentration included in the mixture. This is because increasing the nanoparticles concentration would result in enhancing the overall thermal performance of the suspension, and hence less heat would be required to raise the temperature of the fabricated nanofluid, and vice versa [188]. In general, nanofluids effective specific heat capacity is lower than their base fluids [126,189]. According to Ali et al. [13] and other researchers [190,191,192,193], the most accurate theoretical model for calculating the effective specific heat capacity of a nanofluid ($C_{p_{nf}}$) is the following equation:(9)$$C_{p_{nf}}=\frac{\rho_{bf}\times \left(1-f_{V}\right)}{\rho_{nf}}\times C_{p_{bf}}+\frac{\rho_{np}\times f_{V}}{\;\rho_{nf}}\times C_{p_{np}}$$
where $C_{p_{bf}}$ and $C_{p_{np}}$ are the specific heat capacities of the base fluid and the nanoparticles, respectively. Experimentally, the $C_{p_{nf}}$ can be determined using the differential scanning calorimetry (DSC) technique, which basically measures the amount of heat required to be delivered to both test sample and reference source, of well-known heat capacity, so that a temperature rise can be achieved [188].

### 5.1. Effective Thermal Conductivity

Thermal conductivity enhancement of heat transfer fluids has always been the main driving force that motivated researchers into developing nanofluids. This is because the solid particles added to the liquid have tremendously higher thermal conductivity compared to that of the base fluid, and thus cause the effective thermal conductivity of the mixture to improve significantly. At the early stages of their discovery, the claims on the enhancement caused by the dispersed particles on the hosting fluid were seen as a controversial topic because many published works across the literature reported divergence in the level of enhancement and measurement results were difficult to be replicated [194,195,196,197]. Nevertheless, a worldwide round-robin, including 33 research institutes, have demonstrated acceptable consistency in measuring the effective thermal conductivity of nanofluids, despite the fact that they unexplored any anomalous improvement in the effective thermal property [198]. Up to today, the effective thermal conductivity of the suspension remains a complicated topic, where it involves many vital elements such as the particles type and morphology, particles concentration, base fluid type and temperature, added surfactants (if any), and dispersion stability [13,199,200]. When constraining the first four parameters in fabricating a dispersion, the optimum effective thermal conductivity is usually reached when the particles are well distributed in the hosting fluid with minimum to no agglomerations/clustering between them. Since a stable state nanofluid reflects that its nanoparticles are homogeneously dispersed within the hosting base fluid, it should theoretically result in a superior overall suspension thermal conductivity to those of an unstable state. The potential influence of nanoparticles agglomeration on the thermal conduction emphasizes that colloid chemistry will play a significant role in enhancing the thermal conductivity of nanofluids. Scientists such as Yu et al. [201], Haghighi et al. [202], and Li et al. [203] have all proven, through their research work, that stabilized nanofluids have greater and steady effective thermal conductivity than their counterparts. Prasher et al. [204] and Wang et al. [205] explained this observation by analyzing the effect of nanoparticles aggregation on the thermal conductivity of nanofluids, where they assumed that solid liner and side chains get formed by particles clustering. Based on the researcher’s conclusion, these chains are mainly responsible for enhancing the suspensions thermal conductivity. Still, as more nanoparticles get accumulated, the cluster becomes heavier, and therefore separates from the base fluid due to the gravitational force. The aforementioned causes the thermal conductivity of the colloidal to degrade, with respect to settling time, until it decreased to a minimum possible value when total separation is attained. The previous claim was also supported by the work of Hong et al. [206], where they examined the effective thermal conductivity of SWCNTs—water dispersion with magnetic-field-sensitive nanoparticles (Fe_2_O_3_) under various magnetic field strengths. In their experiment, the researchers successfully interconnected the dispersed CNTs using Fe_2_O_3_ nanoparticles and the employed magnetic field, and thus forming a well aligned chains of nanomaterials. This resulted in the effective thermal conductivity to increase by 50% over that of the base fluid. However, as the holding time under the magnetic field increased, the nanomaterials started to form larger clumps that caused the suspension effective thermal conductivity to degrade. Other studies have also proven the enhancement in nanofluids thermal conductivity through the chain concept, such as the work of Wright et al. [207], Wensel et al. [208], and Hong et al. [209]. All three groups of scholars relied on the magnetic field to form the dispersed particles connected networks in the host fluid. However, the first used a novel alignment approach via coating the SWCNTs with Ni, whereas the other two achieved the interconnection with the aid of metal oxide nanoparticles (e.g., Fe_2_O_3_ and MgO). It is important to note that different types of base fluid and surfactants were used in the three previous studies. Younes et al. [210] have suggested an innovative nanoscale aggregation process that can be adopted to form nanosolids with an interconnect chain capability when dispersed in liquids. In their work, they coated the CNTs through their aggregation process with metal oxide nanoparticles and different types of surfactants. Afterwards, the scholars filtered and dried the aggregate to obtain their as-prepared CNTs-based nanosolids. These newly formed nanomaterial can interconnect when dispersed in a non-aqueous solution by applying a magnetic field. Figure 16 illustrates the effective thermal conductivity degradation theory, which describes the mechanism in which the particles separate from the base fluid due to the formation of both linear and side chains. Other aspects that have less influence on the effective thermal conductivity of nanofluids includes the liquid layering near the outer particles surface [211], Brownian motion of dispersed particles [212,213], thermophoresis [214,215], near-field radiation [216,217], and ballistic transport and nonlocal effects [218,219].

Many different techniques have been adopted for measuring the thermal conductivity of nanofluids, namely; 1—cylindrical cell method, 2—temperature oscillation approach, 3—steady state parallel-plate method, 4—3-ω method, 5—thermal constants analyzer approach, 6—thermal comparator method, 7—flash lamp method, 8—transient hot-wire method, 9—laser flash method, and 10—transient plane source. More details on the usage, advantages, and disadvantages of these thermal conductivity measurement techniques can be found in Mashali et al. [17], Paul et al. [220], Qiu et al. [170], and Tawfik [221] published works. Among the previously mentioned techniques, the transient hot-wire approach was mainly adopted across the literature for nanofluids effective thermal conductivity measurements, although it was the first measuring route for such property [17]. The reasons that attracted researchers into favoring the transient hot-wire method among other methods is due to its capability of eliminating measurements errors caused by the natural convection of the fluid, its minimal amount of time required to perform each measurement (i.e., within seconds), and its simple conceptual design compared to other available devices or apparatuses. One thing that needs to be emphasized here is that the high thermal conductivities of graphene, ND, and CNT found in the literature are based on theoretical calculations for a single particle, and that when attempting to measure this thermal property for a powder sample, the results will show tremendously lower values due to the presence of air along with the limitation associated with the measuring tool itself [222,223].

Furthermore, researchers have published numerous amounts of literature on improving nanofluids effective thermal conductivity over their base fluids [224,225,226,227,228]. For example, Yu et al. [224] compared the thermal conductivity of their stable graphene—EG nanofluids to that of pure EG. The researchers have dispersed 2.0 and 5.0 vol. % of graphene, of 0.7–1.3 nm in size, in EG to fabricate their nanofluids at a set of temperatures from 10 to 60 °C, using the two-step approach. They have found that the as-prepared 5.0 vol. % suspension had 86% enhanced thermal conductivity over its base fluid at 60 °C. Yarmand et al. [225] synthesized water based nanofluids from 0.02 to 0.1 wt % of functionalized graphene nanoplatelets using the two-step method at 20–40 °C. The functionalization process was conducted through an acidic treatment to the graphene powder by dispersing the as-received graphene in a 1:3 mixture of HNO_3_ and H_2_SO_4_, respectively. They found that the formation of sedimentation within their as-fabricated nanofluids was minimal throughout their 245 h test. The heat transfer coefficient improved by 19.68% compared to the base fluid when using the 0.1 wt % nanofluid. Furthermore, Yarmand et al. [225] concluded that the thermal property of the suspension is influenced by the temperature of fabrication and the dispersed solid concentration. Zhang et al. [226] compared the thermal conductivity of three ionic based nanofluids containing graphene sheets, graphite nanoparticles, and SWCNTs. All three types showed enhanced thermal conductivity with a partial increase in viscosity compared to their base fluids. Nevertheless, the nanofluid fabricated from graphene had a higher increase in thermal conductivity compared to the other two types of dispersions. Ghozatloo et al. [227] studied the effect of time, temperature, and concentration on the thermal conductivity of pure and functionalized CVD graphene–water nanofluids. The functionalizing process of graphene was conducted through an alkaline method, and the suspensions were fabricated using sonication (i.e., the two-step approach). Moreover, the concentration used in the production of the suspension was of 0.01–0.05 wt %, and the duration of the dispersion mechanism was 1 h. The authors found that the nanofluids samples containing pure graphene had promptly developed clusters between its solid content, whereas the functionalized suspensions were highly stable. Furthermore, the effective thermal conductivity was seen to reduce to a certain extent for all nanofluids after the time of production. In addition, the enhancement in the effective thermal conductivity using functionalized graphene showed to be 13.5% (0.05 wt %) and 17% (0.03 wt %) over 25 °C and 50 °C water, respectively. Askari et al. [46] experimentally investigated the thermal and rheological properties of 0.1–1.0 wt % CVD nanoporous graphene–water nanofluids along with heat transfer suspension effect on the thermal performance of a counter-flow arranged mechanical wet tower. The base fluid used in the two-step suspension fabrication was taken from one of the working cooling towers located in South Iran to reflect a real-life case scenario. Different types of surfactants were used to stabilize the dispersion of the as-prepared nanofluids, such as AG, Tween 80, CTAB, Triton X-100, and Acumer Terpolymer. The authors found through analyzing the physical stability of their nanofluids, utilizing the sedimentation capturing method and zeta potential measurements, that using Tween80 as a disperser resulted in a stabilized suspension that can last for up to two months. Furthermore, their 1.0 wt % nanofluid showed a 16% increase in the thermal conductivity at a dispersion temperature of 45 °C. At the same time, the low concentration suspensions would be appropriate for industrial applications because of their increasing effect on the effective density and viscosity. Moreover, the as-produced nanofluids enhanced the efficiency, cooling range, and tower characteristic compared to the conventional base fluid. For example, using a 0.1 wt % fabricated nanofluid had resulted in a 67% increase in the cooling range and a 19% decrease in the overall water consumption. Goodarzi et al. [229] studied the effective thermal conductivity, specific heat capacity, and viscosity of their as-prepared nitrogen-doped graphene–water nanofluids along with their convective heat transfer behavior when employed in a double-pipe type heat exchanger. The authors used 0.025 wt % of Triton X-100, as their surfactant, along with 0.01–0.06 wt % of graphene to prepare the suspensions using the two-step method. Their results showed that the examined thermophysical properties where very sensitive to both temperature and concentration. As an example, the effective thermal conductivity of their suspension showed an increase from 0.774 to 0.942 W/m·K with the increase in temperature (from 20 to 60 °C). The maximum effective thermal conductivity achieved by the scholars was 37% higher than that of the base fluid. Furthermore, they found that increasing the concentration of their nanosheets in the base fluid had caused the heat transfer coefficient of their working fluid to improve but at the same time results in increasing the pressure drop in the system and the pumping power requirement. Liu et al. [230] examined the effective thermal conductivity and physical stability of their synthesized graphene oxide–water nanofluids. Moreover, the mass fraction and temperature of the investigated samples were 1.0–4.5 mg/mL and 25–50 °C, respectively. The researchers found that they can achieve a homogeneously stable nanofluid for about 3 months using their preparation process. They also found that the effective thermal conductivity of their as-prepared nanofluid was 25.27% higher than the base fluid, at 4.5 mg/mL mass fraction, and a temperature of 50 °C. Ghozatloo et al. [228] explored the possibility of improving the convection heat transfer behavior of a shell and tube heat exchanger, under laminar flow conditions, using CVD graphene nanofluid of water base. The researchers also investigated the effect of temperature and solid dispersed concentration of the mixture on the thermal conductivity and convective heat transfer coefficients. The dispersions were prepared using 0.05, 0.075, and 0.1 wt % of treated CVD graphene and 15 min sonication in water. According to the authors outcomes, using 0.05, 0.075, and 0.1 wt % suspensions, at 25 °C, enhanced the thermal conductivity over pure water by 15.0%, 29.2%, and 12.6%, respectively. Moreover, the convective heat transfer coefficients of the as-produced mixtures depended on the flow conditions in which the working fluid undergoes but were in all cases higher than the base fluid. From the previously mentioned studies, it can be concluded that carbon-based nanomaterials can form stabilized nanofluids, either by selecting the appropriate base fluid–nanoscaled material matrix or through external physical and/or chemical approaches. Moreover, these suspensions have enhanced thermal properties compared to their conventional base fluids, but the level of enhancement gets affected by parameters such as concentration, temperature, physical stability, etc. Thus, such factors should be carefully considered to obtain the optimum suspension thermophysical condition.

Besides the experimental studies, many researchers have developed theoretical correlations to predict the effective thermal conductivity of nanofluids. Still, most of these formulas have shown conceptual limitations towards their experimental origin. Table 4 demonstrates the developments in the effective thermal conductivity equations.

### 5.2. Effective Viscosity

The effective viscosity of nanofluids is part of the chain that determines the applicability of using such a category of suspensions in heat transfer applications. Since it is a transport property directly related to the dynamic performance of the heat transfer system, where an increase in colloidal viscosity would lead to an increase in the friction coefficient and thus a raise in the pressure losses in the system. The heat transfer system then compensates for this pressure difference by increasing the pumping power, and accordingly, more electrical power gets consumed. For such a reason, many research studies have been devoted to investigating the link between the nanofluids effective viscosity and the different parameters associated with the suspension, such as nanoparticles shape, size, concentration, dispersion stability, and mixture temperature [247,248,249,250,251,252,253,254,255,256]. Mena et al. [257] suggested that nanofluids fabricated with nanoparticles concentration of up to 13 vol. % behaves as Newtonian fluids (i.e., their viscosity is independent of shear strain). In addition, many researchers proved that the stability of nanofluids has an inverse relationship with their effective viscosity. Meaning that well-dispersed suspensions tend to have lower effective viscosity than those of poor stability [202,258,259,260]. If the viscosity of a shelved nanofluid was to be categorized according to its stability condition, then there would exist one to three different viscosity regions. To be more precise, a well-dispersed suspension would roughly have a uniform viscosity along its column, while three different viscosity regions would exist in the semi-stable case, and two different viscosity regions would form in the unstable separation scenario. Figure 17 demonstrates the three stability cases with their different viscosity regions. As for the nanofluid in its dynamic form, these viscosity regions would most likely still exist within the suspension while flowing in the hosting system. Knowing this, one can explain why the unstable suspension would require higher pumping power compared to the stable form of the same dispersion. To calculate the percentage of viscosity increase that the dispersed particles cause to the base fluids, the following equation can be used [154]:(10)$$Viscosity\;increase\;\left(\%\right)=\left(\frac{{µ}_{eff}}{{µ}_{bf}}-1\right)\times 100$$
where ${µ}_{eff}$ and ${µ}_{bf}$ are the effective viscosities of the nanofluid and the base fluid, respectively. Furthermore, the most common and widely used approach for determining nanofluids viscosity is via the rotational viscosity measurement method [261]. In this method, a shaft is inserted in the sample, after which the rotational speed and the torque per unit length of the shaft are used to determine the viscosity of the nanofluid. Other measuring techniques are also used, such as the capillary viscometer, concentric cylinder viscometer, rheonuclear magnetic resonance, and rheoscope [262,263,264]. To be noted that, according to Prasher et al. [265], in order for a nanofluid to improve the performance of its hosting thermal application, the increase in effective viscosity should not exceed four times the mixture enhanced effective thermal conductivity, otherwise the working fluid would not serve the hosting system in terms of overall performance. This can be theoretically calculated through the following equations [266]:(11)$$\frac{\left(\frac{{µ}_{eff}}{{µ}_{bf}}\right)-1}{\left(\frac{k_{eff}}{k_{bf}}\right)-1}<4$$

In another study, Akhavan-Zanjani et al. [267] investigated the thermal conductivity and viscosity of nanofluids made of graphene, water, and polyvinyl alcohol (PVA) surfactant. The wt % used were of 0.005–0.02% and the mass ratio of the PVA employed was 3:1. The authors found a significant increase in the as-prepared fluid thermal conductivity with a moderate raise in the viscosity. The highest recorded percentages for the thermal conductivity and viscosity were 10.30% and 4.95%, respectively. Iranmanesh et al. [268] analyzed the effect of two preparation parameters, namely; the concentration and temperature, on aqueous graphene nanosheets nanofluids thermal conductivity and viscosity. They used 0.075–0.1 wt % to fabricate their nanofluids using the two-step method at 20–60 °C. The findings indicated that the wt % used had a clear effect on the viscosity and thermal conductivity on the prepared dispersion. Moreover, the temperature, as a parameter, was seen to have a larger influence on the level of the final product viscosity compared to the added solid concentration. Although the study avoided any employment of surfactants, the authors as-prepared suspension was stable for several days. Such an observation is not new and was also reported by other researchers, such as Mehrali et al. [159], where they successfully fabricated stabilized nanofluids made of the graphene–water mixture without the need for surfactants or graphene functionalization, but these are rare cases because of the attraction nature between the head groups of both particles and the base fluid molecules. Ghazatloo et al. [269] have developed a model that can predict the effective viscosity of CVD graphene–water and CVD graphene–EG nanofluids at ambient temperature. They used experimental measurements of the property and employed a commonly used model to form their correlation. Moreover, the researchers used 0.5–1.5 vol % of 60 nm graphene sheets with two separate base fluids (i.e., water and EG), after which the content was subjected to sonication for 45 min. For their water based nanofluids, a volume ratio of 1.5:1 of SDBS surfactant to solid content was used to physical stabilize the dispersion. The outcome of their research indicated that the effective viscosity remarkably increased with the raise of vol. %, and hence the concentration as a parameter had a significant effect on the property. Furthermore, the Batchelor model [270] showed some deviation from the experimental data of the as-prepared suspensions viscosity. This variation in results were reduced by the newly developed model, which, unlike the previous model, considered the solid additive geometry. The comparison between the authors model, Batchelor model, and experimental results is demonstrated in Figure 18.

On the other hand, in terms of the effective viscosity theoretical models developments, Table 5 lists these correlations with their year of development, dependent parameters, and limitations. From the formulas shown in Table 5, it can be concluded that most of the authors have used specific experimental conditions to come up with their correlations, and hence the majority of the models are limited to their operating conditions (i.e., they cannot be accounted as universal models).

## 6. Thermal Applications

The previous sections showed how dispersing carbon-based nanomaterials in conventional working fluids could positively affect these liquids properties, especially when it comes to their overall thermal conductivity. On the contrary, this section concentrates on utilizing carbon-based suspensions in three heat and mass transfer systems widely used in the energy sector, namely, PTSCs, nuclear reactors systems, and AC&R systems. This is because the previous attempts that many researchers undertook to enhance the performance of these systems were mainly through design modifications, such as adding turbulators, geometric and construction materials variations, and surface alterations. However, these techniques have reached a point where limited enhancements can be accomplished. Therefore, to break these boundaries for further progress, some scientists have proposed exchanging the working fluids of these thermal applications with nanofluids [293]. This is because employing a working fluid that possesses higher thermal conductivity would eventually improve the heat transfer rate in these systems, as will be demonstrated next.

### 6.1. Parabolic Trough Solar Collectors

A PTSC is part of the existing energy solar systems that utilizes solar radiation (usually emitted from the sun) to generate thermal energy with high efficiency [294]. This happens when reflecting concentrated incident sunlight from its reflector surface, which consists of a parabolically curved mirror polished metal, to a focal line where the receiver or absorber tube containing the working fluid is located. The lower temperature heat transfer fluid, which is usually water or oil, then absorbs the solar heat flux from the attached inner tube surface, and thus causes its temperature to raise. Figure 19 shows an example of a real life PTSC system and its working mechanism in a schematic diagram. Based on the system configuration and the application used, the working fluid temperature in a PTSC can exceed 500 °C at concentrated solar power plants (CSPP), for steam power cycles; or can be lower than 100 °C, for industrial process heat (IPH) applications, such as domestic and industrial water heating [295]. Examples of low temperature requirements (i.e., temperature starting from ≤100 °C) for different industrial processes are shown in Table 6 [296]. Most of the modern designs of PTSC contain a sunlight tracking system that helps improve the efficiency of these systems [297].

Since the primary goals in industrial applications are to reduce the processing time, increase the lifetime of the equipment, and decrease the amount of energy consumption, using PTSC systems, these goals can be fulfilled through improving the rate of heat transfer between the absorber tube and the working fluid. One way of achieving this is by utilizing nanofluids as the heat transfer fluid in the PTSC system [299,300]. This is because, as mentioned earlier, nanofluids have higher thermal conductivity than any known conventional heat transfer fluid, which makes them potential candidates for the future of such heat transfer applications. When using carbon-based particles (e.g., MWCNTs, graphene, or NDs), the effective thermal conductivity significantly increases along with the rate of thermal diffusion and effective viscosity of the suspension. Subsequently, this causes the fluid heat capacity, Reynold’s number (Re), and Prandtl number (Pr) to decrease. In the case of turbulent flow, the Nusselt number (Nu) depends on both Re and Pr. Thus, a decrease in the two aforementioned parameters would result in fewer or smaller eddy formations within the fluid, and hence the level of turbulently in the flow would reduce. Furthermore, since the effective viscosity of a nanofluid is higher than its base fluid, the pressure drop that will be experienced from using such category of fluids in a PTSC system would be higher than that of the conventional base fluids. To overcome this issue, the PTSC system should take into account the thermophysical properties of the suspension used at its design phase. One important thing to consider is that when using nanofluids, as the PTSC working fluid, the absorber tube needs to be transparent so that the dispersed particles can directly absorb the sunlight throughout their cycle [301].

Although the previous facts showed how promising nanofluids could be when used in PTSC systems, the scientific field is still scarce with the amount of published works that investigate carbon-based nanofluids in such system. Most of the work covered on nanofluids were those involving nanoparticles of Al_2_O_3_, CuO, TiO_2_, Fe_2_O_3_, SiO_2_, Cu, SiC, Fe_3_O_4_, and limited other literature were found for CNTs, MWCNTs, and SWCNTs [298]. For instance, Kasaeian et al. [302] explored the overall efficiency enhancement of a pilot PTSC system using MWCNTs–mineral oil suspensions of 0.1 wt % and 0.3 wt %. The researchers found that the 0.1 wt % and 0.3 wt % dispersions had improved the system efficiency by 4–5% and 5–7%, respectively, compared to conventional base fluid (i.e., mineral oil). Furthermore, Kasaeian et al. [303] studied the effect of 0.1, 0.2, and 0.3 vol. % of MWCNTs dispersed in EG, as the working fluid, for a direct absorber solar collector attached to a parabolic trough. They found that the optical efficiency reached up to 71.4%, due to the 0.3 vol. % of MWCNTs particles employed in their heat transfer fluid. In addition, the thermal efficiency of their system was found to be 17% higher, when using the 0.3 vol. % nanofluid, than that obtained from pure EG. Moreover, Mwesigye et al. [304] coupled a Monte Carlo ray tracing (MCRT) optical model along with a computational fluid dynamics (CFD) finite volume method (FVM)-based model to analyze a PTSC, hosting a SWCNTs–Therminon VP-1 suspension, thermal performance. The authors found that raising the particles concentration from 0 to 2.5 vol. % caused the entropy generation to reduce by 70%, with the heat transfer rate to increase by 234%, and the thermal efficiency of the system to improve by 4.4%. In addition, Dugaria et al. [305] designed and modeled the optical efficiency of a direct absorber solar collector (DASC) that is connected to a parabolic trough system. In their experiment, they used 0.006, 0.01, 0.02, and 0.05 g/L of SWCNTs to fabricate their aqueous nanofluids. Their results showed that increasing the nanoparticles concentration to more than 0.05 g/L would cause the thermal efficiency to reduce due to the thermal radiation being mostly contained in the surrounding area between the absorber tube inner surface and the nanofluid. In addition, using nanofluids made of 0.05 g/L SWCNTs caused the thermal efficiency of the system, including the optical losses of the concentrating trough, to reach 90.6% at a reduced temperature range ($T_{m}^{*}$) = 0 K∙m^2^/W and 77.2% at $T_{m}^{*}$ = 0.128 K∙m^2^/W. It is important to note that the thermal efficiency of solar collectors is usually shown in a graph as a function of $T_{m}^{*}$, which is defined for the case of nanofluids as:(12)$$T_{m}^{*}=\frac{\left(T_{m_{nf\;}}-T_{amb}\right)}{DNI}$$
where $T_{m_{nf}}$, $T_{amb}$, and DNI are the mean temperature of the nanofluid, ambient air temperature, and direct normal irradiance, respectively. One of the main aspects for the enhancement in the thermal performance of the two aforementioned published works [304,305] was due to the fact that CNTs, along with other carbon-based materials, possessed extremely high solar absorption characteristics (i.e., more than 90%) [306]. Despite the research investigations that were covered in this section on carbon-based nanofluids usage in PTSC’s, there are only a few other alternatives [298]. To the best of the authors of this article knowledge, there is still a lack of exploration on utilizing ND’s and graphene nanofluids for PTSC’s. This shows that further investigation is required from the researchers working in the solar energy field; especially since, for example, nanofluids of ND base showed to contain remarkable optical and thermal properties when studied in other similar applications [307].

### 6.2. Nuclear Reactors

Nuclear power plants are part of the energy network that has been adopted by many countries across the globe, such as France, USA, UK, Russia, Iran, and UAE, among others to support their growth in energy demands [308]. Unlike most energy sources, the power produced from the fission process of the fuel (i.e., enriched uranium or plutonium) within the nuclear reactor can arguably be considered as one of the solutions for solving the problems associated with climate change and the increasing levels of CO_2_ emissions in the atmosphere and its feasibility for none or low oil producing countries [309]. Nuclear technology has seen significant developments throughout the years to enhance the efficiency of these systems, reduce their construction size, and improve their safety standards [310,311]. Historically, the first generation of commercial nuclear reactors were inaugurated in the 1950s, whereas today, the newly introduced fourth generation of reactors are currently being either planned or under construction. In terms of the working fluid, these reactors can be classified into three main groups (i.e., water-cooled reactors (WCRs), gas-cooled reactors (GCRs), and molten solid cooled reactors (MSR)) [312]. The WCRs can be subdivided into further categories, namely, boiling water reactors (BWRs), pressurized water reactors (PWRs), and pressurized heavy water reactors (PHWRs). Furthermore, the thermal transport concept of the BWR and both PWR and PHWR is similar in the sense that the working fluid, in all cases, absorbs the thermal energy from the fuel when it undergoes an excited state. However, the main difference is that PWR and PHWR use pressurizing systems to maintain the working fluid in its liquid phase, and therefore must be separated from the electrical generating cycle for contamination safety concerns. On the other hand, the working fluid in the BWR is boiled to generate steam that is used directly to provide the needed mechanical power to rotate the steam turbine and generate electricity. In addition to being a thermal energy carrier for power generating purposes, the working fluid also takes the role of extracting heat from the nuclear fuel, which is primarily the main concern related to the safe and economic operation and lifespan of the reactor. In some cases where the cooling rate is insufficient or if the control rods fail to operate properly to stabilize or reduce the reaction process, the reactor can experience a loss-of-coolant accident (LOCA) [313]. In such scenarios, the nuclear fuel needs to be rapidly cooled down, using backup water tanks, to avoid a core meltdown crisis and possibly a hydrogen explosion in the chamber. From the aforementioned, one can generalize the modes of heat transfer inside the rector’s core based on the driving force of the fluid motion into two main categories; the first is flow boiling, which is a forced convection phenomenon that occurs during normal operating conditions. The second is pool boiling, which is a natural convection heat mechanism that takes place following a reactor LOCA state. Enhancing the heat transfer coefficient (HTC) and critical heat flux (CHF), for flow boiling, or increasing the minimum film boiling temperature (T_min_) in pool boiling are essential for optimizing these thermal modes outcomes. Whether it comes to improving the energy efficiency or for safety reasons, the aforementioned shows how crucial the role of the working fluid in a nuclear reactor system. Therefore, utilizing working fluids of enhanced thermophysical properties, such as nanofluids, can help in further advancements in the field of nuclear power plants, especially in WCR systems, if properly handled and understood its role in both nuclear flow boiling and pool boiling [314]. This section demonstrates some of the available studies on nanofluids for both thermal modes (i.e., flow and pool boiling), but focuses more on the pool boiling mode due to its important role in designing an emergency core cooling system.

#### 6.2.1. Nanofluids Influence on Flow Boiling

During the normal operating condition, the thermal efficiency of a reactor system depends mainly on the flow boiling of the working fluid, where the working fluid is forced to flow by means of a pump and buoyancy effects. Flow boiling consists of several flow regimes [315] such as liquid single-phase flow, bubbly flow, slug flow, annular flow, mist flow, and vapor single-phase flow, as shown in Figure 20. The existence of each regime is affected by several factors such as the type of working fluid, surface orientation, degrees of liquid subcooling, system pressure, wall temperature, mass flux, surface microstructure (including porosity), wettability, oxidation, and surface roughness [316,317]. Moreover, this mode of thermal transport can remarkably improve the energy efficiency of the system when the HTC and CHF of the working fluid are enhanced [318,319]. Researchers through their intensive studies, have shown that a passive and a safe approach to achieve higher HTC and CHF can be accomplished using nanofluids in the system [320]. The level of enhancement that accompanies the utilization of nanofluids over their other conventional counterparts is tuned by altering nanoparticle morphology (i.e., shape and size), thermophysical properties and different flow parameters (i.e., mass flow rate, channel size, flow direction, and flow regime) [321,322]. Published reports described that nanofluids of enhanced thermal conductivities also showed better convective flow performance [319,323]. Carbon allotropes based nanofluids have higher thermal conductivities compared to the metallic and oxide based nanofluids [324,325,326]. Therefore, nuclear scientists have already started to investigate the flow boiling performance of various carbon-based nanofluids.

Some of the work done on carbon-based nanofluids in flow boiling includes the research conducted by Hashemi and Noie [328]. They experimentally used MWCNTs–water suspensions produced through the two-step fabrication method and stabilized by adding the AG surfactant, of the 1:1 surfactant to the nanomaterial ratio, to the mixture. The MWCNT used had a 10–20 nm outer diameter and 30 µm length, and base fluid concentrations between 0.0005 and 0.005 vol. %. Furthermore, the stability of their dispersions was determined through the zeta potential method, and the testing section consisted of a horizontal stainless steel (SS) tube of 10 mm (diameter) and 200 cm (length). Their findings showed that the HTC of the as-prepared nanofluids was significantly higher than the base fluid, and that the enhancement in the HTC for both types of fluids was influenced by the changes in heat and mass fluxes. In addition, the CHF in the nanofluids case showed a maximum improvement over pure water by approximately 4.3%, when using the 0.005 vol. % suspensions. The same research group [329] also presented the feasibility of AG stabilized MWCTs–water nanofluids at various concentrations (0.001, 0.005, and 0.01 wt %) for flow boiling inside a 2 m long tube placed horizontally under atmospheric condition. The zeta potential analysis of test samples confirmed the well dispersion of nanoparticles in base fluid. Their report describes that CHF of nanoparticles increased significantly due to particle inclusion as well as the increase in mass flux. The CHF enhancement was observed better than water due to nanoparticle deposition and enhancement in wettability of the heating surface. Another flow boiling study under forced convective and nucleate boiling regions using CNTs nanofluid was conducted by Sarafraz and Abad [323]. Their work was performed using statistical, regression and experimental analyses for commercial heat transfer oil based CNTs nanofluids. The nanofluids were prepared by dispersing 0.1 wt % and 0.3 wt % of the dry powder in therminol 66 for a total duration of 25 min, using a two-step procedure (magnetic stirring 15 min then sonication 10 min). The stability of the suspensions was achieved by adding nonylphenol ethoxylates surfactant of 0.1 vol. % to the mixture matrix, after which the zeta potential of the prepared samples was measured. The results showed that the stability of the prepared nanofluids was maintained for 5 days. Additionally, the presence of carbon nanotube within the oil increased the convective HTC together with a substantial augmentation in the nucleate boiling HTC. The degree of subcooling also increased the HTC in both boiling regions due to enhancements in thermophysical properties of nanofluids. Furthermore, the thermal the thermo-hydraulic performance of the system enhanced by 56% using the carbon nanotube nanofluid compared to the same with pure oil. Sarafraz and Hormozi [330] experimentally investigated flow boiling heat transfer in MWCNT, alumina, and copper oxide flow. Different techniques were implemented for stabilizing based on the type of the dispersed nanomaterial. The results showed that MWCNT–water nanofluids had higher thermal conductivity and boiling thermal performance compared to the other suspensions. It was also found that, as the heat and mass fluxes and the concentration of nanofluids increased, the boiling performance of MWCNT nanofluids intensified. However, the HTC was deteriorated in force convection and nucleate boiling regimes due to the deposition of the nanoparticle on the surface. As a result, the surface roughness decreased over the time since it is affected by the size of nanoparticles, thickness of deposited layer and size of microcavities. This was confirmed by the minimal amount of bubble generation due to reduction in nucleation sites and surface roughness. The researchers finally concluded that the MWCNT–water nanofluids outperformed the other candidates for utilization in cooling applications.

Other studies have found that addition of graphene oxide (GO) into base fluid improve the CHF in flow boiling due to its hydrophilicity feature [331,332]. Lee et al. [331] examined 0.01 vol. % of GO–water nanofluid in round tubes with a length of 0.5 m and an inner diameter of 0.01041 m at two inlet temperatures (25 and 50 °C) and four different mass fluxes (100, 150, 200, and 250 kg/m^2^∙s) for low pressure and low flow scenarios. Comparing to other oxide nanofluids from the literature [333], the research group showed maximum CHF enhancements at mass flux of 250 kg/m^2^∙s increased up to 100% and 72% at fixed temperatures of 25 °C and 50 °C, respectively. This significant improvement was due to the liquid film’s wettability enhancement caused by the deposition of GO nanoparticles. However, Park and Bang [332] reported limited improvement in CHF of up to 20% when testing GO–water nanofluid in advanced light water reactors (ALWRs) at 50 and 100 kg/m^2^.s and subcooling condition of 10 K compared to distilled water. The results showed that GO deposited on the heated surface and changed phase to reduced GO (RGO) during nucleate flow boiling, which might constrain the thermal activity improvement. Zhang et al. [334] examined the deposition of GO in water nanofluids over heating surface with nanoparticle concentration ranging from 0 to 0.05 wt %. They reported that the increase in GO concentration depreciated the heat transfer performances (CHF and HTC) up to 100% and 73% (at 0.05 wt % with 40 mL/min), respectively. In another study, Mohammed [335] varied graphene particle concentration from 0 to 0.5 vol. % in zinc bromide and acetone solution (acetone–ZnBr_2_).The CHF and HTC on the heated surface increased with GO concentration by up to approximately 52% and 58%, respectively. However, the increase in particle concentration involved a decrease in pressure drop up to 11% approximately.

In terms of nanofluids containing NDs, there were a limited amount of literature covering this topic [336,337,338]. For instance, DolatiAsl et al. [338] proposed a mathematical model using Kim et al. [336] results to estimate the different parameters that were affecting the CHF when utilizing ND suspensions. Their correlation only required the properties of the nanomaterial and the vol. % employed to predict the CHF. Furthermore, the numerical findings showed that the most effecting parameters on the CHF were the length of the tube (decreasing) and the mass flux (increasing), whereas the particles concentration and thermal conductivity had the lowest influence. They extended the work by developing another correlation that contains the particles property data, and thus only required an input of the type of the particles (i.e., NDs) and the vol. % to perform the prediction. The new model predicted the actual values with a mean absolute error of 9.8% for CHF ranging from 500 to 2500 kW/m^2^.

#### 6.2.2. Dispersions Effect on Pool Boiling

Pool boiling heat transfer plays a crucial role in numerous technological and industrial applications. It consists of four different regimes [339]: natural convection (single-phase), nucleate boiling, transition boiling, and film boiling as shown in Figure 21. When a surface is sufficiently heated and plunged in a water pool, film boiling regime occurs where the heated surface is physically separated from the coolant by a stable vapor blanket. This region is denoted by the film boiling regime. In this regime, heat transfers by conduction and radiation only leading to a gradual decrease in the surface temperature. The performance of the pool boiling is evaluated by the enhancement in CHF, HTC, T_min_, and vapor film thickness. In literature, the effects of the following parameters have been studied: substrate material [340], surface conditions and oxidation [341,342], system pressure [343,344], initial wall temperature [345], shape and dimension of the testing specimen [346,347], degrees of liquid subcooling [348,349,350], surface wettability and vapor–liquid contact angle [351], surface roughness and wickability [352], and type of quenchant such as water, oil, or nanofluids [353,354]. Recently, researchers have been focused on the effects of the later parameter on pool boiling heat transfer performance.

Review papers have intensively presented research studies that cover the preparation methods of various nanofluids and test their effect on CHF and HTC [321,355,356,357,358,359]. It was mentioned that carbon-based nanomaterials such as graphene dispersed in water enhanced the heat transfer as compared to any other nanoparticle [355]. Nevertheless, there are many reasons that contribute to the CHF enhancement such as surface roughness [360], deposition of nanoparticles [361], concentration of nanoparticle beyond a certain limit [362,363], increase in surface wettability [364], and capillary effect [365]. Some researchers have explained that the enhancement was a result of the occurrence of cavities on the surface due to the deposition of the dispersed nanomaterial on the surface, especially on surfaces of rough structure. Others mentioned that the increase in surface area of the formed porous layer because the nanoparticle accumulation enhanced heat transfer by disturbing the flow. Active nucleation sites decreased with nanoparticle layers, which significantly increase surface wettability, and therefore enhance CHF. As for HTC enhancement, thermophysical properties has a major role on it. It has been observed that nanoparticle affects the thermal conductivity and surface tension of the quenchant whereas viscosity, density, and heat capacity remain nearly constant [366]. A vapor blanket layer formed during film boiling on the heating surface, which significantly affect the conduction heat transfer, and hence the HTC. Yang and Liu [367] found that as the surface tension decreased, the HTC increase due to the reduction in the formation of bubbles and the active nucleation sites. The effect of the modified surface topologies such as surface roughness was observed to enhance HTC [133,368]. Furthermore, the high particle concentration led to nanoparticle deposition on the surface and thus porous surfaces occur. Depending upon either original surface condition or size of nanoparticle, the surface roughness can be increased [360] or decreased [369]. Generally, HTC was found to be maximum when carbon nanoparticles were used for boiling [355]. Since the current work focuses on the effect of carbon-based nanofluids (i.e., graphene, ND, and CNT) on heat transfer application, various selected studies on CHF and HTC in pool boiling heat transfer have been listed in Table 7 and Table 8, respectively.

The third important parameter in pool boiling is T_min_ to be considered in the reactor core under the extreme environment and severe accidents such as LOCA. Understanding this parameter can lead to an improved nuclear cladding performance that provides more efficient and safer future nuclear reactors. Physically, T_min_ is defined as the boundary between film boiling and transition boiling, beyond which temperature liquid loses physical contact with the solid surface and the heat transfer significantly reduces. Fewer studies for the quenching behavior of nanofluids have been conducted in the literature that was focused on T_min_.

A study by Gerardi [387] showed that T_min_ of a quenched indium-tin-oxide (ITO) rod in a nanofluid pool (ND–water) was found to increase by 30 °C compared to the water pool. Another experimental investigation by Fan et al. [388] was performed in aqueous nanofluids in the presence of four CNTs having various lengths and diameters. It was concluded that the accelerated quenching was clearly related to the enhancement in boiling heat transfer. An increase in T_min_ was exhibited for all cases. The modified quenching and boiling behaviors were elucidated by the accumulative changes in surface properties due to the deposition of CNTs. Given the nearly unvaried contact angles, the consistently increased surface roughness and the formation of porous structure seem to be responsible for quenching and boiling enhancement. In order to achieve better performance, the use of longer and thicker CNTs tends to form a highly porous layer, even upon consecutive quenching, which may induce rewetting by the entrapped liquid in the pores and serve as vapor ventilation channels as well. In another experimental study, Fan et al. [389] examined transient pool boiling heat transfer in aqueous GO nanofluids. They tested various dilute concentrations of the nanofluids up to 0.1 wt %. It was shown that the quenching processes could be accelerated using GO nanofluids as compared to pure water. The boiling behavior during quenching was analyzed in relation to the modified surface properties of the quenched surfaces. T_min_ values were found to increase with raising the concentration of GOs compared to the baseline case of pure water. The results suggested that surface property changes due to the deposition of GOs were responsible for the modified boiling behavior of the nanofluids. In addition, the surface wettability was a nondominant factor in most cases. The surface effects of the deposited layer of GOs were strongly dependent on the material properties, finish, and treatment of the original surfaces. Kim et al. [390] quenched metal spheres made from SS and zircaloy in water-based nanofluid containing low concentration (less than 0.1 vol. %) of ND. They showed that film boiling heat transfer in nanofluids was almost identical to that in pure water. However, subsequent quenches proceeded faster due to the gradual accumulation of nanoparticle deposition on the sphere tended to destabilize the vapor film but, T_min_ remained unchanged. A summary of the previous research studies is listed in Table 9.

### 6.3. Air Conditioning and Refrigeration Systems

Air conditioning (AC) is a process used to controls air’s thermal and physical properties and then supply it with cooling or heating to an allocated area from its central plant or rooftop units. It also maintains and controls the temperature, humidity, air cleanliness, air movement, and pressure differential in a space within predefined limits so that conditioned space occupants or products enclosed satisfy comfort and health standards [391]. A typical AC or refrigeration system uses a vapor compression cycle to accomplish cooling or heating. The vapor compression cycle consists mainly of a compressor, an evaporator, a condenser, an expansion device, indoor and outdoor fans, and a working fluid. Additionally, secondary heating and cooling loops are implemented to accommodate more extensive systems, as shown in Figure 22. The AC system is potentially used for providing a clean, healthy, and comfortable indoor environment, and saving energy by developing high-efficiency equipment in residential and industrial sectors. However, none of these uses come without associated challenges. The AC&R systems can be operating with a very high temperature lift (different between heat source and heat sink temperatures). For instance, the AC system operates in hot and dry climate countries needs to maintain indoor temperature as low as 18.3 °C (65° F) [392], whereas the refrigeration system needs to run continuously for long hours to sustain freezing chamber temperatures [393]. As a result, the AC&R systems generate a tremendous amount of heat loss to the environment during the compression process, which increases the pressure ratio across the compressor and degrades its efficiency; it increases the compressor discharge temperature and jeopardizes its reliability. Simultaneously, the cooling demand is compromised and the AC or the refrigeration system strives to provide enough cooling in the unit’s evaporator (or reject heating in the unit’s condenser) and therefore degrades the overall system coefficient of performance (COP).

One of the methods to improve the COP of AC&R systems is to reduce the power consumption in the compressor. Many researchers have already shown that adding nanoparticles to the compressor oil (nanolubricant) reduced its energy consumption because it enhanced the lubricating oil’s tribological and thermal properties, which helped improve the compression process, and therefore increased the system COP [394,395,396,397,398,399,400,401,402]. Lee et al. [403,404] studied the effects of adding nanoparticle to mineral oil. Their results showed that the improvement in the lubricating properties of the mineral oil increases with the addition of the nanoparticle. The authors found that adding the nanoparticle to the compressor oil decreased its friction coefficient by 90%, and thus causing improvement in the compression process and reducing the energy losses in the compressor. Jia et al. [405] investigated the effects of using mineral-based nano-oils in a domestic refrigerator compressor with two different refrigerants, namely, R-134a and R-600. They concluded that the COP values increased by 5.33% when the nano-oil was utilized in the compressor with R-600, whereas no effects were noticed when the same nano-oil was used with R-134a.

Another method to improve the cooling COP is to increase the heat transfer coefficient in the heat exchangers of the AC&R system. Many studies have already shown that mixing nanoparticles with the refrigerant enhanced the heat transfer coefficient of the refrigerant (nanorefrigerant) in the condenser and the evaporator because of the additional nucleate boiling and the higher thermal conductivity of the nanoparticles that enhanced the heat transfer rate, and thus increasing the system COP [286,395,401,406,407,408,409,410,411]. Since carbon-based nanofluids (i.e., ND, graphene, CNTs, etc.) have better performance due to their superior features compared to other known nanomaterials [17,406,412,413,414,415,416,417,418], they could result in significant system performance improvement. Therefore, researchers have further investigated carbon-based nanoparticles for various AC&R applications [419], such as the ones demonstrated in Figure 23. The following sections present a literature review on studies investigating the effect of carbon-based nanoparticles on the thermophysical properties of AC&R refrigerants, followed by a literature review on studies investigating the effect of carbon-based nanoparticle on the AC&R system’s performance.

#### 6.3.1. Influence of Carbon-Based Nanoparticles on the Thermophysical Properties of Working Fluid in AC&R Systems

It is evident from the literature that there are many researchers investigated the performance properties (i.e., heat transfer coefficient and viscosity) of nanoparticles applied to the refrigerants in AC&R systems including copper (Cu), aluminum (Al), nickel (Ni), copper oxide (CuO), zinc oxide (ZnO), aluminum oxide (Al_2_O_3_), titanium oxide (TiO_2_), and other metal nanoparticles [325,419,420,421,422,423,424,425,426,427,428,429,430,431,432,433,434]. However, only a limited number of research work is available for ND, graphene, and CNTs, which can be summarized in Table 10. Park and Jung [435] investigated the possible contribution of CNT on the nucleate boiling heat transfer coefficients of R-123 and R-134a. They reported an enhancement up to 36.6% in nucleate boiling heat transfer coefficients of the nanorefrigerant at low heat flux compared to the baseline refrigerant. However, as the heat flux increases the enhancement decreased due to robust bubble generation that prevented the CNT from penetrating the thermal boundary layer and touch the surface. The flow boiling heat transfer characteristics and pressure drop were also investigated experimentally by Zhang et al. [436], using MWCNT dispersed in the R-123 refrigerant with SDBS surfactant flowing in a horizontal circular tube heat exchanger. Their results showed that the nanorefrigerant heat transfer coefficient and frictional pressure drop increased with the increase of nanoparticle concentration, mass flux, and vapor quality. Similar conclusions were observed by Sun et al. [437] when they investigated MWCNT with R-141b. Jiang et al. [438] studied the influence of CNT diameters and aspect ratios on CNT–R-113 nanorefrigerant. The study involved four different groups of CNTs with different physical dimensions (diameters, length, and aspect ratio). Their experimental results showed that the thermal conductivities of CNT nanorefrigerant increased proportionally with the increase of CNT’s volume fraction and aspect ratio and with the decrease of CNT’s diameter. The maximum increase in the thermal conductivity was about 104% for a volume fraction of 1.0 vol. %. Peng et al. [439] studied the influence of CNT physical dimensions such as diameters, length, and aspect ratios for the CNT–R-113–oil mixture. They used the same four different groups of CNTs with different physical dimensions as Jiang et al. [438] and VG68 ester lubricating oil. An enhancement of up to 61% was obtained in the nucleate pool boiling heat transfer coefficient compared to R-311–oil mixture without CNTs. They also showed that the improvement of the nucleate pool boiling heat transfer coefficient increased as the CNTs length increases and as CNTs outside diameter decreases. The heat transfer performance of MWCNT–oil–R-600A nano-refrigerant in horizontal counter-flow double-pipe heat micro-fin tube heat exchanger, was studied by Ahmadpour et al. [440]. Their experiments covered a wide range of parameters, including mass velocity, vapor quality, and condensation pressure. Their results showed that an increase up to 74.8% in the heat transfer coefficient could be achieved with 0.3% nanoparticles concentration at 90 kg/m^2^.s mass velocity compared to the pure refrigerant. Kumaresan et al. [441] conducted an experimental study on the convective heat transfer characteristics of secondary refrigerant nanofluids in a tubular heat exchanger. The objective of the secondary refrigerant loop is to reduce the primary refrigerant charge in vapor compression refrigeration systems. The nanofluid used in the study consists of MWCNT dispersed in a water-EG mixture. Their results showed that the maximum enhancement in convective heat transfer coefficient was 160% for the nanofluid containing 0.45 vol. % of MWCNT compared to the base fluid. However, the friction factor was also increased by 8.3 times, which might increase the pumping power and reduce the advantage of the increase in the heat transfer coefficient of the nanofluid [442]. Similar findings were attained by Baskar et al. [443] and Wang et al. [444] when they experimentally tested MWCNT–IPA and graphene–EG in a secondary refrigeration loop, respectively.

The dispersion stability of MWCNT in the R-141b refrigerant with the addition of surfactant was investigated by Lin et al. [445]. Three different types of surfactants, including SDBS, hexadecyl trimethyl ammonium bromide, and nonylphenoxpoly ethanol (NP-10), were tested to prevent the aggregation and sedimentation of MWCNTs during the long-term operation. SDBS was found to have an excellent adsorption ability on the MWCNT surface. It was also shown that the relative concentration increased with decreasing MWCNT length or outer diameter and increasing ultrasonication time. The optimal SDBS concentration for the highest dispersion stability increased proportionally with the increase of the initial MWCNT concentration. However, the SDBS might reduce the nanorefrigerant’s thermal conductivity at higher operating temperatures. The thermophysical properties and heat transfer performance of SWCNTs dispersed in the R-134a refrigerant was also investigated by Alawi and Sidik [446]. They found that up to a 43% increase in thermal conductivity can be reached when 5 vol. % of nanoparticle concentration is used in the MWCNT–R-134A nanorefrigerant compared to the pure R-134A refrigerant. Similar to other nanofluids, the thermal conductivity increases with the increase of nanoparticle volume concentrations and with the increase in the temperature of the nano-refrigerant. Moreover, the increase of volume fractions at a constant temperature led to a significant increase in the viscosity and density of the nanorefrigerant.

Dalkilic et al. [447] investigated the stability and viscosity of MWCNTs–polyolester (POE) oil nanolubricants. The study involved using four different refrigeration compressor oil with different values of viscosity (i.e., 32 mm^2^/s, 68 mm^2^/s, 100 mm^2^/s, and 220 mm^2^/s) tested at a maximum temperature of 50 °C and a concentration of MWCNTs up to 1 wt %. They reported a substantial augmentation in viscosity up to 90% compared to the viscosity of the base oil. This could reduce the refrigeration efficiency due to the possible increase in the compressor pumping power. Most of the review studies [325,419,420,431,448] have shown that adding nanoparticles always enhances the heat transfer coefficient of the nanofluid mixture due to the higher thermal conductivity of nanorefrigerant and due to the reduction of the thermal boundary layer thickness caused by the presence of nanoparticles. Additionally, nanoparticles increased the viscosity of the nano-refrigerant causing an increase in the frictional pressure drop and therefore might reduce the AC&R system performance. The review studies of references [325,419,420,431,448] covered only CNTs nanomaterial from the carbon family, and therefore further investigations on other types of carbon-based nanoparticles, such as diamonds and graphene, needs to be conducted.

#### 6.3.2. Influence of Carbon-Based Nanofluids on the COP and Overall Cooling Performance of AC&R Systems

A limited number of studies are available on how ND, graphene, and CNTs improve system COP and cooling capacity, which can be summarized in Table 11.

Abbas et al. [449] examined CNT mixed with POE oil in an R-134a refrigeration unit. They found that the system COP increased by 4.2% with nanoparticle concentration of 0.1 wt %. The experiment was infeasible beyond this concentration because the main challenge was with the agglomeration due to the strong Van der Waals interactions during the preparation phase. Jalili et al. [450] mixed various concentrations of MWCNT with water to assess the cooling performance of the secondary fluid in the evaporator of the refrigeration system. The transient analysis results showed that the evaporator’s inlet temperature increased by 6.5% while the outlet temperature decreased by 14.5% when the water contains 2000 ppm of MWCNT. The significant enhancement in evaporator outlet temperature confirmed the tremendous increase in heat transfer coefficient with MWCNT. According to Kruse and Schroeder [451] and Cremaschi [452], the existence of oil lubricant in heat exchangers acted as insulation and resulted in heat transfer coefficient reduction. However, if the addition of nanoparticles enhanced the oil lubricant, the heat transfer coefficient would be compensated in the heat exchangers due to the improved overall thermophysical properties. Vasconcelos et al. [453] examined MWCNT–water as a secondary fluid in a 4–9 kW refrigeration unit with R-22 as a refrigerant. Due to the high thermal conductivity of the nanofluid, the cooling capacity increased up to 22.2% at the coolant’s inlet temperature range of 30–40 °C. Vasconcelos et al. [453] found no significant reduction in the total power consumption. However, the increase in cooling load helped the compressor power consumption to relatively reduced because of the relative increase in evaporation pressure, and therefore the COP increased up to 33.3%. Kamaraj and Manoj Babu [454] replaced the POE oil with POE–mineral oil nano lubricants containing CNT particles with the amount of 0.1 and 0.2 g/L in a R-134a refrigerator. Besides the reduction in cooling time using the new nanofluids by approximately 40%, the COP was improved by 16.7% for 0.2 g/L of CNT using mineral oil. This is mainly due to the enhancement of the heat transfer coefficient in the evaporator without any significant reduction in the compressor power. Yang et al. [455] analyzed graphene nanosheets blended with SUNISO 3GS refrigerant oil in a R-600a refrigerator/freezer. The authors found that the cooling rate freezing rate improved by approximately 5.6% and 4.7%, respectively. The energy analysis yielded that the three concentrations nanolubricants (10 mg/L, 20 mg/L, and 30 mg/L) helped in reducing the compressor discharge temperature by 2.5%, 3.8%, and 4.6%, and dropping the energy consumption by 14.8%, 18.5%, and 20.4%, respectively. Hence, the energy saving was estimated to be up to 20% compared to using pure refrigerant oil. Indeed, the addition of graphene nanosheets with lubricant oil helped to decrease compressor friction losses. However, using graphene nanosheets as nanolubricant required additional surfactants (dispersants), which might increase the compression power because surfactants increase the viscosity and reduce the thermal conductivity at higher operating temperatures. Pico et al. [456] investigated two mass concentrations of ND–POE in a 17.6 kW vapor compression refrigeration system that used variable-speed compressor and refrigerant R-410A. The results showed that the compressor power consumption remained the same due to the type of compressor (i.e., hermetic scroll). On the other hand, the discharge compressor temperature reduced by approximately 3 °C and 4 °C, while the cooling capacity increased by 4.2% and 7%. Therefore, the overall system COP increased by 4% and 8% at 0.1% and 0.5% mass concentrations, respectively. Furthermore, Pico et al. [457] experimentally investigated the same ND–POE nanolubricant with R-32 as a substitute for R-410A. The results showed that for 0.5% mass concentration of diamond nanoparticle added to POE lubricant, the cooling capacity increased by 2.4% and the discharge compressor temperature decreased by approximately 2 °C, and hence the COP enhanced by 3.2%. The reduction of ND–POE performance with R-32 compared to R-410A can be justified with the low mass flow rate, which affected the oil circulation rate of the system operating with R-32.

## 7. Environmental Consideration and Potential Health Issues

In addition to the studies that focused on the thermal enhancement that nanofluids can provide to energy systems, researchers have also explored the environmental impact and the potential hazardous towards human health from using these kind of suspensions [459]. In terms of environmental concerns, Meyer et al. [460] proposed an exergoenvironmental analysis method for designing an energy conversion system with an as low as possible environmental impact. Furthermore, an Eco-indicator 99 life cycle impact assessment approach was employed by the scholars to evaluate the environmental impact in a quantitative manner. They obtained the ecoindicator point through undertaken a series of analyses such as the exposure and effect analysis, resource and fate analysis, and damage analysis. One of the main element in determining the fate analysis is the toxicity evaluation of the nanomaterial(s) that form(s) the nanofluid. However, the criteria for measuring the level of toxicity is still not clear, and as such uncertainty always exist [461]. In order to resolve the previous issue, Card and Magnuson [462] came with a two-step approach for evaluating the toxicity of nanomaterials in an objective manner. In the first step, the authors collected the available literature that are related to the toxicity of nanomaterials, after which they started to evaluate and rank these studies according to the suitability of the design, documentation of adopted approach, materials used, and research outcome to produce the reliability ‘study score’. As for the next step, the fulfilment of the physicochemical characterization of the nanomaterials is assessed in every study, which is then used to generate a ‘nanomaterial score’. In general, the optimum way to reduce the environmental impact is to lower the utilization of resources and the resulting gas emissions during operation condition. For systems the utilize nanofluids, this can be achieved through enhancing their thermal efficiency, reducing their overall system size, and/or reducing the number of nanomaterials used. The reader is guided to the following sources [463,464,465,466,467,468,469,470,471] for further details on the aforementioned.

On the other hand, users dealing with nanofluids are always under high health risk [472]. This is because in the preparation process of these suspensions there is an inevitable contact between the user and the nanomaterials used, which can therefore enter their blood streams and/or organisms through skin absorption, inhalation, and/or ingestion of these toxic materials. It is important to note that almost all carbon-based nanomaterials, including CNTs, NDs, and graphene, are toxic and that the level of their toxicity increases with the decrease in their size [473]. Moreover, it is important that the reader distinguish between the safety aspect of nanomaterials that are used as medication (or for inner body diagnostic) and those used for other non-medical applications [110,474,475,476,477]. The first are safe and non-toxic when transferred to the human body in the appropriate way, whereas the second have high health risks and should be dealt with safety precautions. Some of the studies that were mainly devoted towards evaluating the harm that nanomaterial can cause to human health can be found in the following literature [473,474,478,479,480,481,482], where the material type, shape, size, surface characteristics charge, curvature, free energy, and functionalized groups were taken into account. Although using personal protective equipment and following the safety handling procedures can lower the user health risks, it is believed that part of the problem associated with commercializing nanofluids is due to the increasing concerns from the potential stakeholders and a lack of sufficient research studies [472,481].

## 8. Discussion and Future Directions

This review article has covered carbon-based nanofluids, from the fabrication stages of the raw nanoparticles materials (i.e., ND, graphene, and CNT) and up to their employment in some of the commonly known thermal applications in the energy industry. In addition, it was shown how dispersions made of carbon allotropes possess the most favorable thermal properties and, when well handled, physical properties compared to any other type of nanofluids or conventional fluids. This is because these carbon-based materials, when dispersed in a base fluid attain unique features such as high thermal conductivity and specific heat capacity, high heat transfer rate, and lower pressure drop in the working system compared to other types of dispersed nanomaterials. Furthermore, the aforementioned suspensions cause the least corrosion and erosion effects on the hosting device [483], all of which are crucial parameters for the operation cycle. Moreover, the influence of the stability of these suspensions on their thermophysical properties was also highlighted along with the development in these properties prediction correlations. Nevertheless, there are still some challenges and gaps in the scientific knowledge that need to be tackled for further advancement in the field and the possible industrial viability of such advanced fluids. Some of these issues are pointed out in this section.

### 8.1. Challenges in Carbon-Based Nanofluids

Carbon allotropes in the nanoscale have shown to be promising in enhancing the thermal performance of liquids when homogeneously dispersed, and their products are commercially available through a wide range of companies. Nevertheless, these powders are very expansive from an economic perspective compared to other sorts of nanomaterials, which makes their utilization quite questionable in the sense of their feasibility towards the targeted application [484]. Therefore, one of the challenges that need to be focused on is how to fabricate large quantities of these nanopowders at minimal production cost. In the current situation and before even introducing such type of nanofluids to the industry, researchers need to initially evaluate the gained performance enhancement and economic benefits of carbon-based nanofluids for each selected application before hands, and hence more work is needed in this area. Some scholars have proposed combining carbon-based nanomaterials with other cheaper types of nanoparticles (e.g., Cu, Al, and Fe) to form hybrid nanofluids containing carbon allotropes, and thus reducing the suspension cost [485,486,487,488]. However, the feasibility of such an approach remains limited, and the consideration of such types of hybrid nanofluids remains in the exploration stages. As it is well-known by now that the favorable thermophysical properties of nanofluids have made such a category of working fluids beneficial when used for enhancing the system performance of many thermal applications. Yet, the stability of the dispersed particles remains a major drawback and thus limiting the widespread of these suspensions. This is because, in an unstable state, the particles tend to cluster into larger forms of agglomerate, and therefore the benefits of the high surface area of the nanoparticles losses its optimum effectiveness on the exposed host (i.e., base fluids). For such a reason, it is essential that any proposed nanofluid to the industry maintains its long-term stability. This is where the preparation phase of the product plays a critical role. In order to overcome this difficulty, scientists have suggested using physical approaches (e.g., sonication) and/or chemical methods, such as surfactants and surface functionalization. Although this can help solve the aggregation problem, the changes caused to the surface of the dispersed particles remain another uncertainty that needs to be understood. In addition, further exploration on combining the two stabilization methods (i.e., physical and chemical routes) need to be conducted. Moreover, a joint international standard database on the thermophysical properties and physical stability of different types of nanofluids, their dispersed nanomaterial(s) concentration, and fabrication approach is strongly needed [198]. This is because even after more than 25 years from the first discovery of nanofluids, scientist are still reporting different thermophysical properties and physical stability for similar synthesized suspensions. In terms of properties prediction, it was shown previously that both effective thermal conductivity and effective viscosity lacks universal correlations and can only be determined through experimental means. However, artificial neural networking that is based on data mining has started to show good accuracy in predicting these properties, but further research is still required in this area [489,490,491,492].

### 8.2. Limitations in Parabolic Trough Solar Collector Systems

In PTSCs, the main challenge is that most of the studies shown in the literature were of pilot-scale tests, and thus further investigations are needed on the real-life application itself. Moreover, systematic studies are required to understand and standardize the influence of operational parameters such as high pressure, high temperature, flow rate, particles concentration, and suspension thermophysical properties on the system performance. Furthermore, there is still a need for a better understanding of the fouling build-up mechanism that is commonly associated with the use of nanofluids in systems of elevated temperatures as this newly introduced thin-film is likely to change the wettability behavior of the surface, and with it the dynamics of the flow [493,494,495]. Knowing this can open the door for introducing systematic washing routines, whether online, offline, or both, that can help in extending the heat transfer efficiency of the PTSC. Other aspects such as the friction factor and pressure drop should also be considered and explored intensively due to the nature of nanofluids having higher effective viscosity than conventional fluids.

### 8.3. Limitations in Nuclear Reactor Systems

As with most heat transfer systems, nuclear reactors were shown to have potential performance benefits from replacing their working fluids with nanofluids. Although on some occasions, certain types of nanofluids cannot compile with such application requirements (e.g., gold and platinum), due to the high temperature nature of such systems and the presence of emitted radiations that effects the dispersed particles [496]. Nevertheless, carbon-based materials have shown the capability of being an acceptable candidate for these systems. Despite that, the common challenge with almost every application that uses this category of suspensions remains rounded on the feasibility of such fluids and their particles clustering issue within the hosting system. When focusing on nuclear reactors, these systems design, sizes, and mechanisms can be seen changing rapidly throughout the past 20 years [308,312,497]. While this shows how this area of science is advancing, it also constrains the exploration capability of researchers working in the field of nanofluids. Thus, scientists specialized in nanofluids cannot investigate the performance enhancement caused by their suspensions on pre-existing reactors. Still, at the same time, they need to take into account the operation lifetime of the facility and understand how the isotopes build up and decay within these systems. This is because such changes in these isotopes could cause different behaviors when exposed to the dispersed particles. In addition, the fact that some of the dispersed particles may deposit on the nuclear fuel surface needs to be also considered and evaluated with respect to the possible corrosion development on the outer surface of the fuel. Moreover, studies on the long-term physical stability of nanofluids, when employed in nuclear reactors, remain unknown and need to be investigated. In addition, further work is needed to determine the effect of surfactants, when used as stabilizers, on the heat transfer rate in such application. This is because most (if not all) of these chemicals cannot withstand high temperature operating conditions. In terms of LOCA scenarios, nanofluids can help stop (or reduce) the level of damages that the fuel of elevated temperature may cause to the facility, but the method in which the newly introduced waste can be dealt with remains questionable and needs to be solved. This is because, unlike conventional liquids, the dispersed particles conserve more radiations, and hence remain radioactive for a very long time before they decay and stabilize. When it comes to pool boiling during quenching, to the best of our knowledge, there is still no existing literature that covers the effect of nanofluids on the minimum film boiling temperature (T_min_) such as what was presented in Section 6.2.2. In general, there is a lack of studies about the impact of nanofluids on T_min_ during quenching. Owing to the importance of T_min_, various types, concentrations, and sizes of nanofluids should have experimented with to investigate their effects on this parameter in specific. Furthermore, most of the investigations that are concern the effect of nanofluids on the CHF uses block plates, flat plates, or wires. However, research work on other geometries is crucial because it is evidence that the CHF will strongly be influenced by it. In addition, the currently employed models (e.g., Zuber’s correlation) fails to accurately predict CHF when using thin wires [498], and therefore scholars need to focus more into developing a universal model that can withstand such limitation.

### 8.4. Limitations in Air Conditioning and Refrigeration Systems

Based on the research conducted by Hu et al. [499], it was shown that some carbon-based nanomaterials need surfactant(s) to ensure long-term stability and avoid agglomeration when they are mixed with the oil–refrigerant in the AC&R system. Additionally, surfactant might help to increase the performance of the AC&R system because it enhances the nucleate pool boiling heat transfer coefficient. However, this improvement is only attained under limited conditions because it is highly dependent on certain specifications (e.g., nanoparticle type, nanoparticle size, nanoparticle concentration, surfactant type, surfactant concentration, nanolubricant concentration, base fluid type, and heat flux). In addition, Cheng and Liu [500] recommended further investigation on nucleate pool boiling and flow boiling for refrigerant based-nanofluids. Therefore, further investigations must be conducted toward carbon-based nanofluids with refrigerant–oil–surfactants for AC&R applications. According to Bahiraei et al. [501] and Dalkılıç et al. [502], nanofluids helped to improve the heat transfer coefficient in heat exchangers (i.e., spiral-type and double-type heat exchangers). However, due to the increase in the nanofluid’s viscosity, the pressure drop can be escalated, especially for low mass flow rates [503,504]. In AC&R systems, the pressure drop is an important factor that needs to be incorporated during the system design phase because any additional increase in the pressure drop in the heat exchangers during the operation of the AC&R system will result in significant degradation in the system COP performance as reported by Sunardi et al. [505] and Tashtoush et al. [506]. Yet, pressure drop due to a carbon-based nanofluids additive has not been studied in AC&R systems and needs to be further investigated. In fact, engineers need robust software tools to design AC&R systems with nanofluid additives. Some studies provided correlations for all thermophysical properties of the nanofluids (heat transfer coefficient, friction factor, thermal conductivity, viscosity, etc.) [507,508,509,510]. However, computational models integrating component models (as employed in Bahman et al. [511] and Loaiza et al. [512]) and the influence of carbon-based nanofluids in AC&R systems are still lacking. In addition, those kinds of models might have the potential to predict the overall system performance and ensure optimal operations. Furthermore, experimental investigations were limited to a specific concentration amount of nanofluids. The optimal concentration for maximum AC&R performance can be obtained numerically with the formerly mentioned computational models. Moreover, the determination of the optimal amount of concentration has not been proposed yet in the literature for carbon-based nanofluids in AC&R systems. In the AC&R system, nanofluid additives ultimately enhance the viscosity of the lubricating oil. Conversely, this might relatively increase the compressor power consumption. Therefore, it is vivid to find the relationship between nanofluid viscosity and compressor pumping power for AC&R application [513,514,515]. A limited number of research employed energy and exergy analysis on carbon-based nanofluids applied in AC&R systems. They have shown that there is a high potential for decreasing the irreversibility with carbon-based nanofluids due to their higher thermal conductivity compared to oxide materials [516]. Therefore, more compressive studies similar to Bahman and Groll [517] are required to identify the AC&R components with major irreversibility when employing nanofluids. Moreover, the literature is scarce in techno-economic analysis for nanofluids in AC&R applications. Although Bhattad et al. [518] showed that nanofluids could result in a higher payback period than the AC&R’s components (i.e., heat exchangers), however, further studies need to be conducted on carbon-based nanofluids because by optimizing the number of nanoparticles with respect to operating condition and stability, it can be more cost-effective. Finally, AC&R systems combine several components, as nanofluid pass through these components, it would be compressed, expanded, or changed phases. All these processes may lead to the possibility of getting nanoparticles to be separated from the carrier fluid during long-term operation and probably degrade the system performance. Therefore, the long-term operating performance of the nanorefrigerant (and nanolubricant) must be investigated.

## 9. Conclusions

This paper provides a comprehensive state-of-the-art review on carbon-based nanofluids, including the initial synthesis methods used for producing carbon nanotubes, graphene, and nanodiamonds, and up to the employment of their dispersions into thermal energy applications, namely; parabolic trough solar collectors, quenching systems, and air conditioning and refrigeration systems. It was shown how some of these nanomaterials could only be fabricated in a dry form, such as high purity nanodiamonds, whereas graphene, for example, can be produced as a dry powder or a suspension. Thus, when selecting the nanofluids’ preparation approach containing these nanomaterials, the available options can be narrowed from two routes to only a single process. Furthermore, the main equations used in calculating the volume concentration that are commonly required for the nanofluid two-step production method were provided. Moreover, the physical stability of the suspension, which is considered as one of the most influential aspect that can dramatically affect the thermophysical properties of any nanofluid, was discussed in terms of its formation mechanism and evaluation approaches. Although there are many advanced ways to characterize the dispersion stability, it was concluded that the photographical capturing method remains the most reliable approach due to its capability of determining both short- and long-term dispersion stability of the mixture in real-time and at high accuracy. Nevertheless, this method is very time-consuming to conduct, especially when the characterized sample is of high state of stability. In addition, chemical methods, such as surfactants and surface functionalization; and physical approaches, namely, ultrasonication, magnetic stirring, homogenizer, and ball milling, were also discussed and shown in how they can be employed to improve the level of particles dispersion within an instable suspension. It was concluded that, unlike the chemical approaches, using physical methods for enhancing the dispersion stability is a better option when it comes to conserving the optimum possible effective thermal conductivity of the nanofluid and that between the available physical routes, the homogenizer can provide the best outcomes. In general, the stability of the suspension does not affect the mixture density nor its specific heat capacity but rather influences both the effective thermal conductivity and effective viscosity of the nanofluid. These two properties were seen to degrade gradually with time due to the nanomaterial’s agglomerations and their sediment formation. Many methods were shown to determine these two properties (i.e., effective thermal conductivity and effective viscosity), either by experimental means or through pre-existing correlations. Still, up to today, the scientific field has failed to provide a universal formula for both of these two properties, and hence the only reliable approach is through experimental analysis. When it comes to replacing conventional working fluids with carbon-based nanofluid in thermal applications (i.e., parabolic trough solar collectors, nuclear reactor systems, and air conditioning and refrigeration systems), it was proven, at least at the lab and pilot-scale, that such advanced fluids are very beneficial in terms of enhancing the overall performance of these systems, and can therefore be seen as strong candidates for such industries when their associated challenges are solved and fully understood.

## Figures and Tables

**Figure 1 nanomaterials-11-01628-f001:**
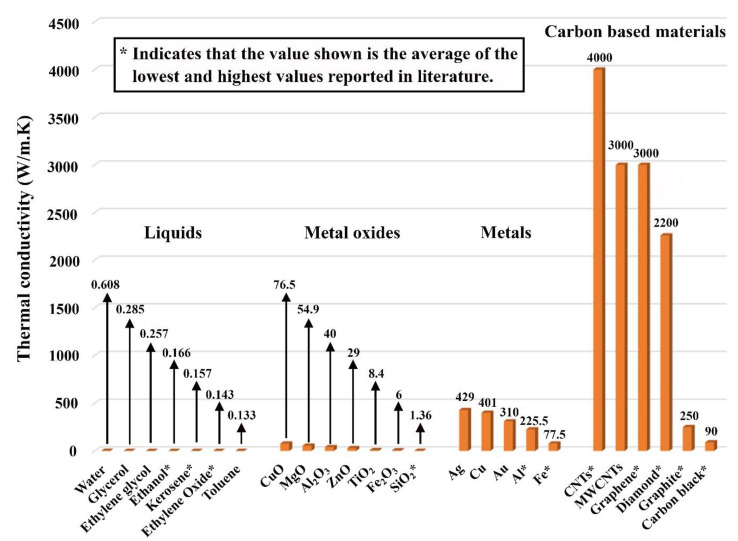
Thermal conductivity of commonly used particles and base fluids for fabricating nanofluids showing an order of magnitude higher in the thermal property for some of the carbon-based materials.

**Figure 2 nanomaterials-11-01628-f002:**
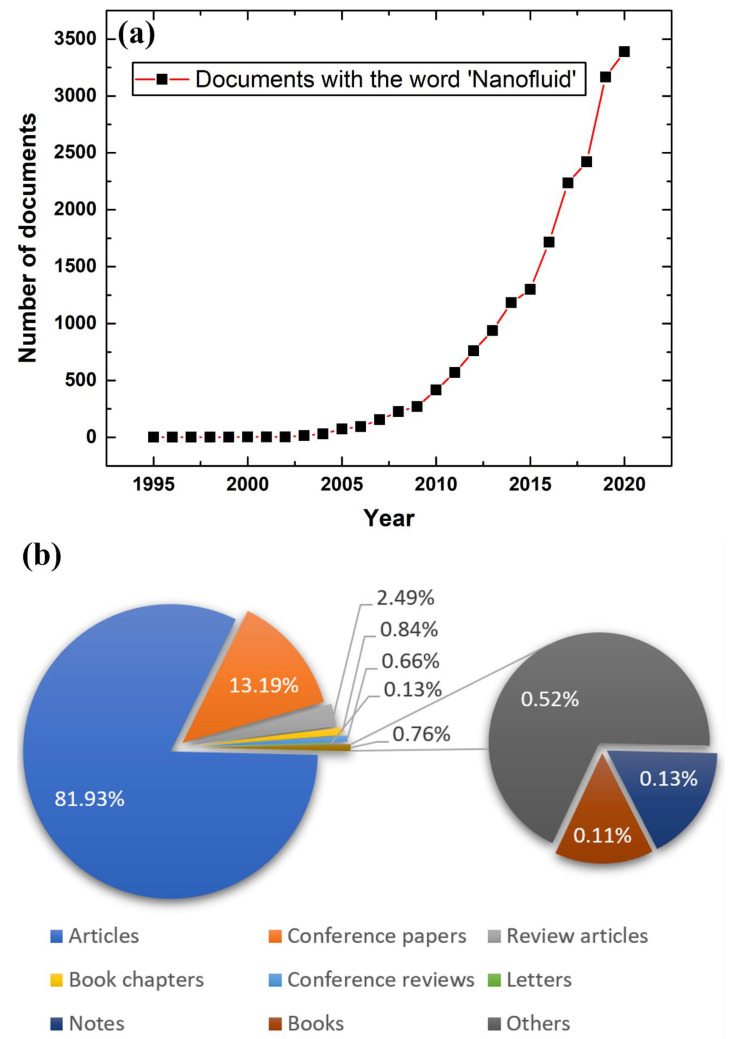
Search result obtained from Scopus database on nanofluids, where (**a**) illustrates the number of published works per year and (**b**) shows the percentage of each type of these documents [35].

**Figure 3 nanomaterials-11-01628-f003:**
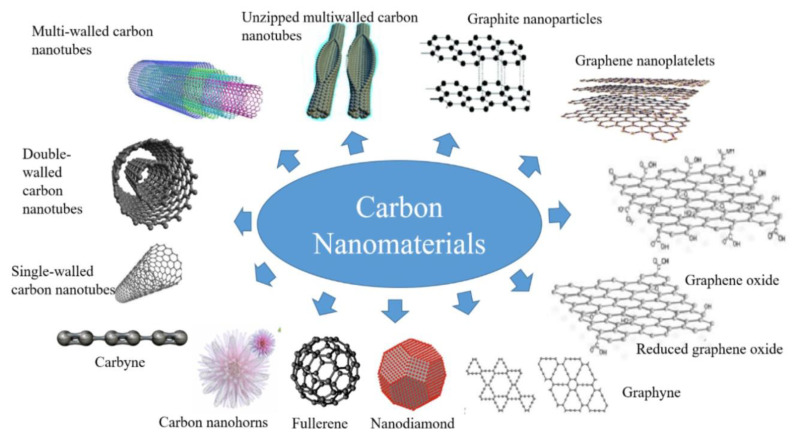
Common allotropes of carbon nanomaterials that grant distinctive thermophysical properties [50].

**Figure 4 nanomaterials-11-01628-f004:**
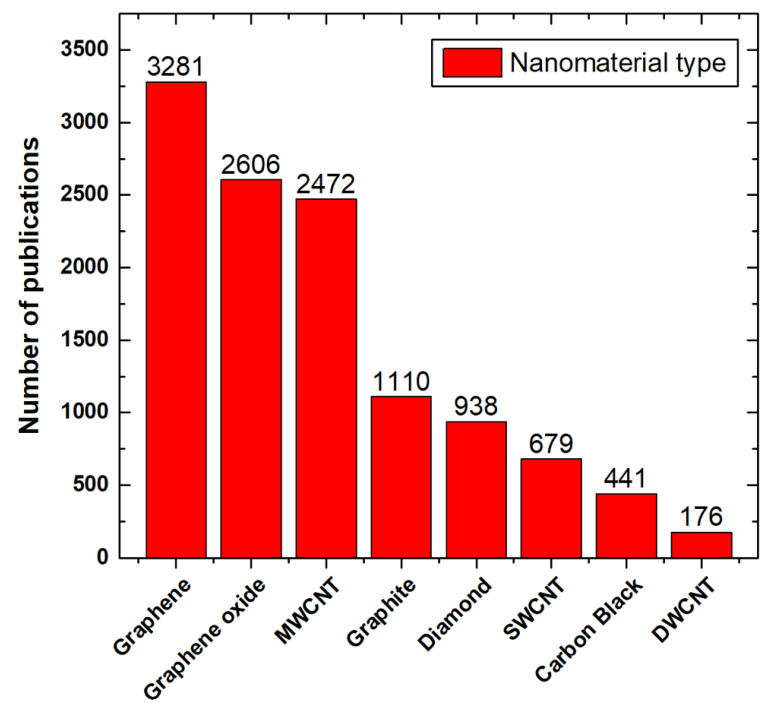
The number of publications available at the Scopus database for common carbon-based material used in nanofluids fabrication [35].

**Figure 5 nanomaterials-11-01628-f005:**
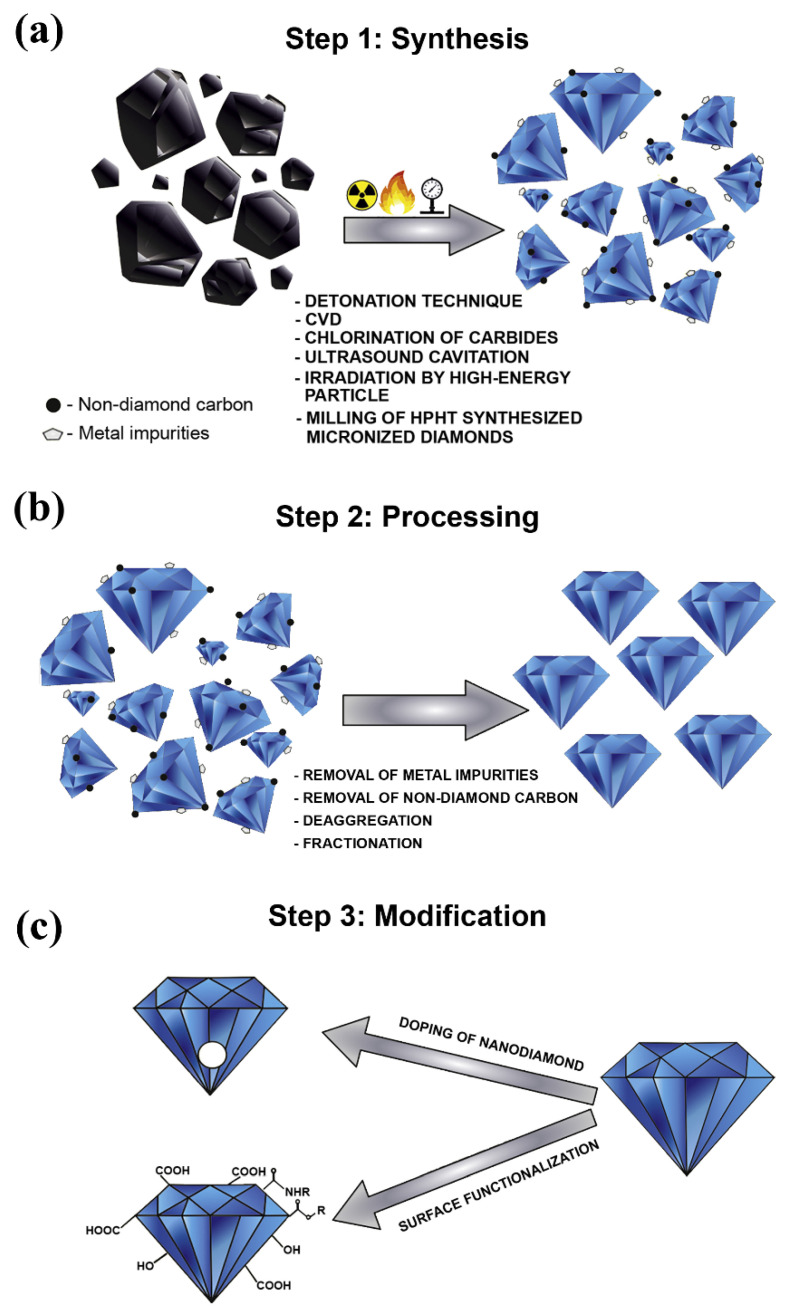
Phases involved in the production process of nanodiamonds, where (**a**) illustrates the synthesis phase, (**b**) demonstrates the processing phase, and (**c**) shows the modification phase. Reproduced with permission from [84]. Elsevier, 2019.

**Figure 6 nanomaterials-11-01628-f006:**
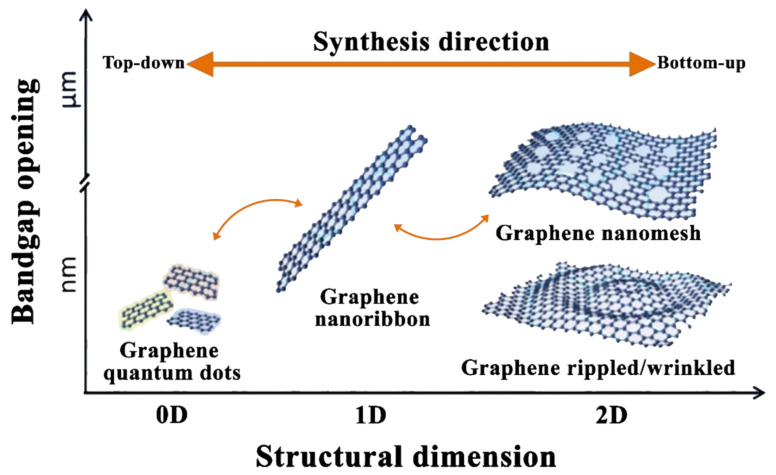
Different forms of graphene based on their dimensionality and bandgap opening, where graphene quantum dots, graphene nanoribbon, and both graphene nanomesh and graphene rippled/wrinkled have a structural dimension of 0D, 1D, and 2D, respectively.

**Figure 7 nanomaterials-11-01628-f007:**
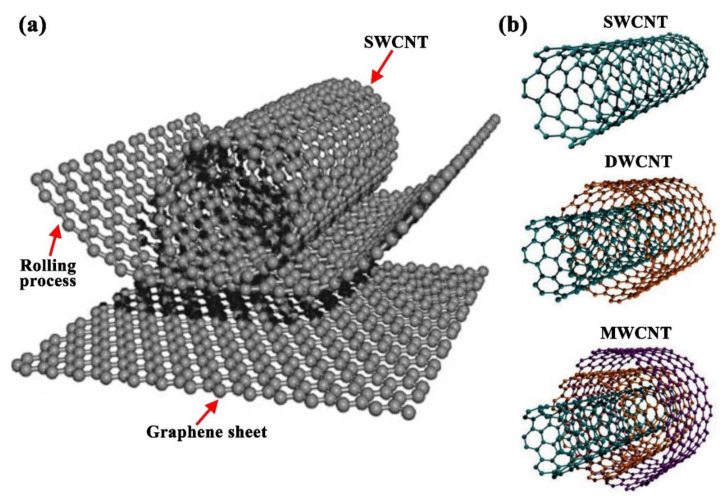
Carbon nanotubes formation and classifications, where (**a**) illustrates the rolling mechanism of graphene sheet into SWCNT and (**b**) demonstrates the three different categories of CNTs, namely SWCNT, DWCNT, and MWCNT.

**Figure 8 nanomaterials-11-01628-f008:**
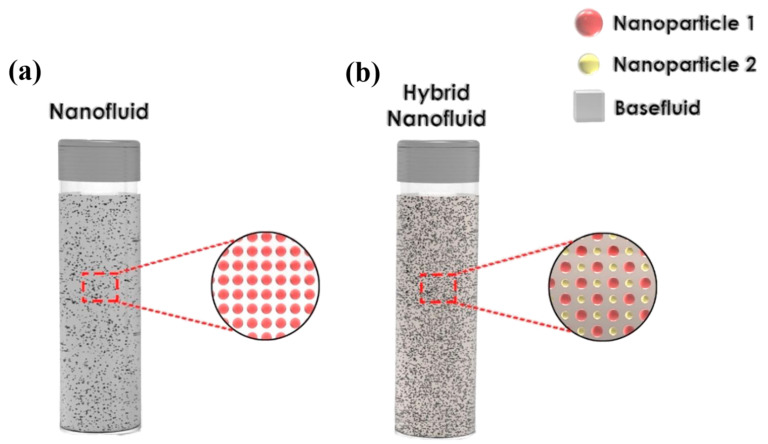
Schematic demonstration to compare between conventional (**a**) and hybrid (**b**) nanofluids that uses the same base fluid.

**Figure 9 nanomaterials-11-01628-f009:**
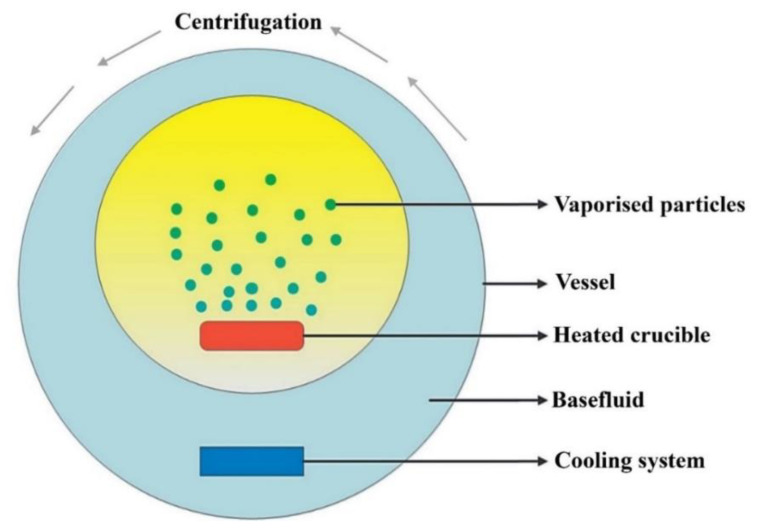
Eastman et al. [21] one-step method of evaporation and centrifugation for nanofluids fabrication [37].

**Figure 10 nanomaterials-11-01628-f010:**
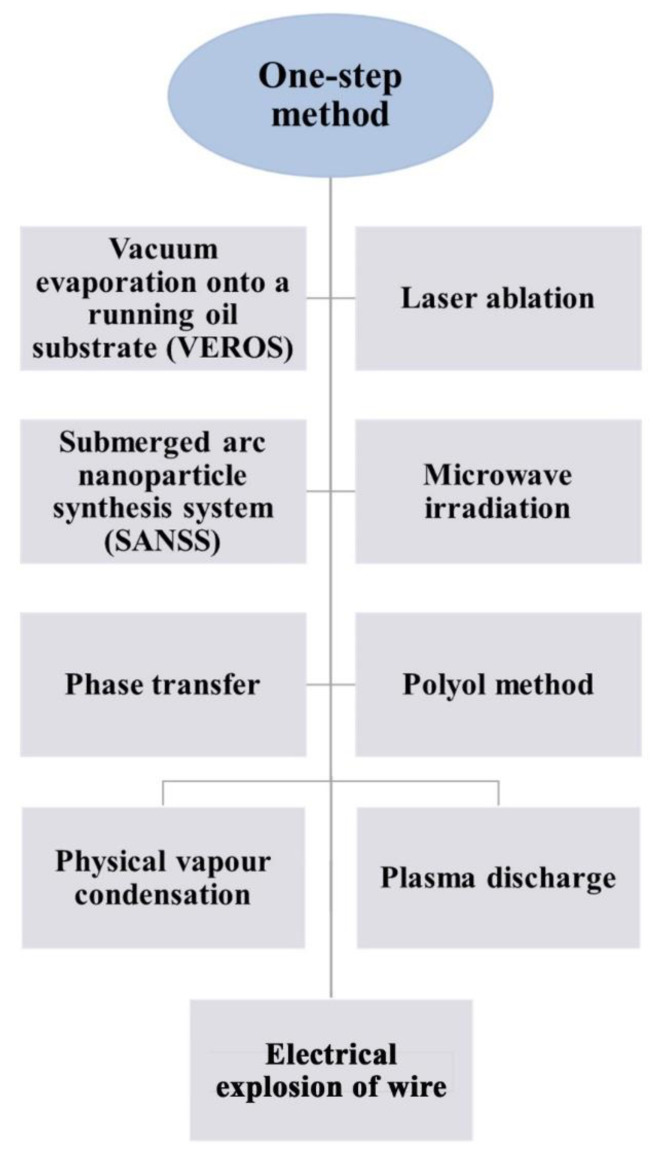
Different nanofluid production approaches that fall under the one-step method.

**Figure 11 nanomaterials-11-01628-f011:**
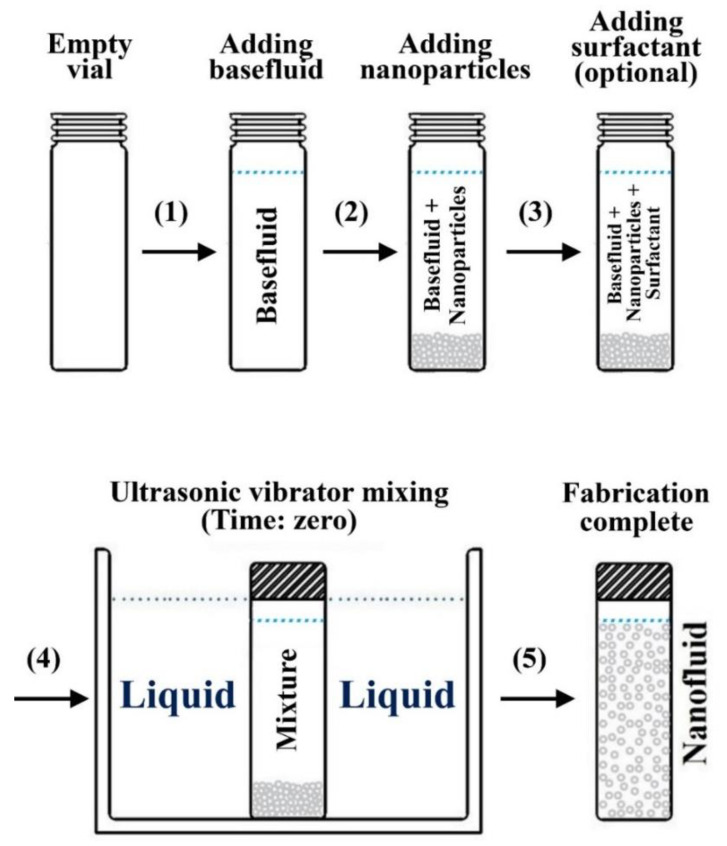
Example of nanofluids two-step preparation using a bath type ultrasonic device.

**Figure 12 nanomaterials-11-01628-f012:**
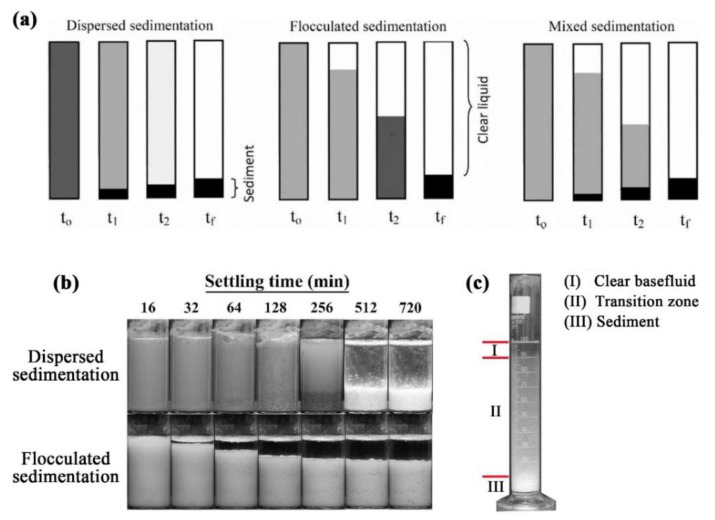
The three types of sedimentation behaviours, where (**a**) shows their schematic mechanism from the starting time (t_o_) to the finishing time (t_f_) [8], (**b**) demonstrates the dispersed and flocculated sedimentation behaviors from Ali et al. [8] experimental work, and (**c**) represents the mixed sedimentation behavior shown in Ma and Alain [174] investigation.

**Figure 13 nanomaterials-11-01628-f013:**
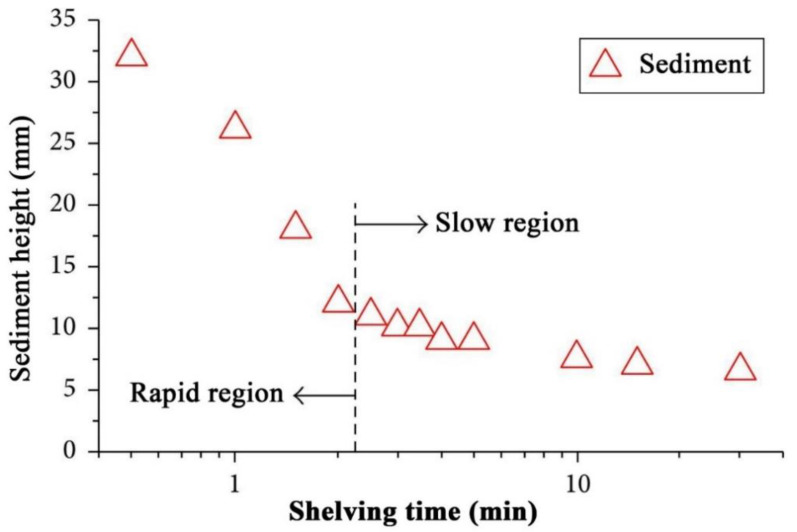
A demonstration of the two sedimentation regions in terms of settling speed, where (the left side) shows the rapid region in which the sediment height changes rapidly, and (the right side) illustrates the slow region, where the changes in the sediment height are very slow to the point where it can be negligible [13].

**Figure 14 nanomaterials-11-01628-f014:**
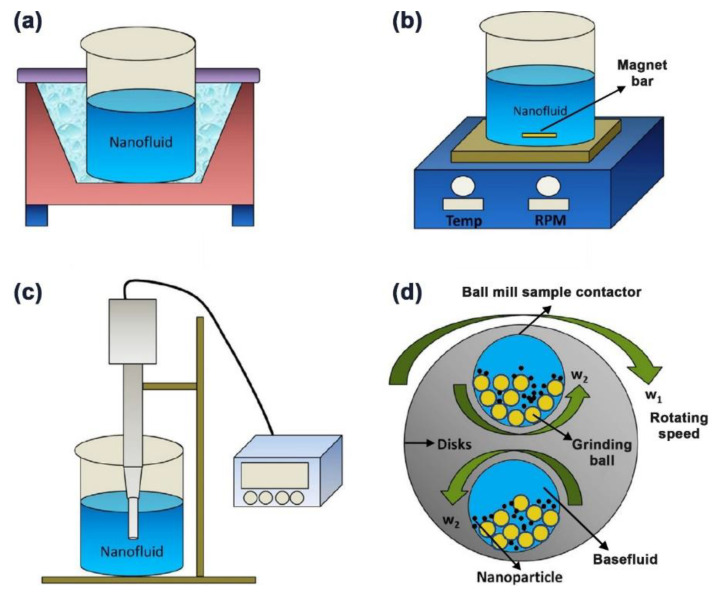
Physical dispersion stability enhancement devices, where (**a**) shows the ultrasonic bath sonicator, (**b**) demonstrate the magnetic stirrer, (**c**) illustrates the homogenizer/prob sonicator, and (**d**) shows the ball milling device. Reproduced with permission from [116]. Elsevier, 2020.

**Figure 15 nanomaterials-11-01628-f015:**
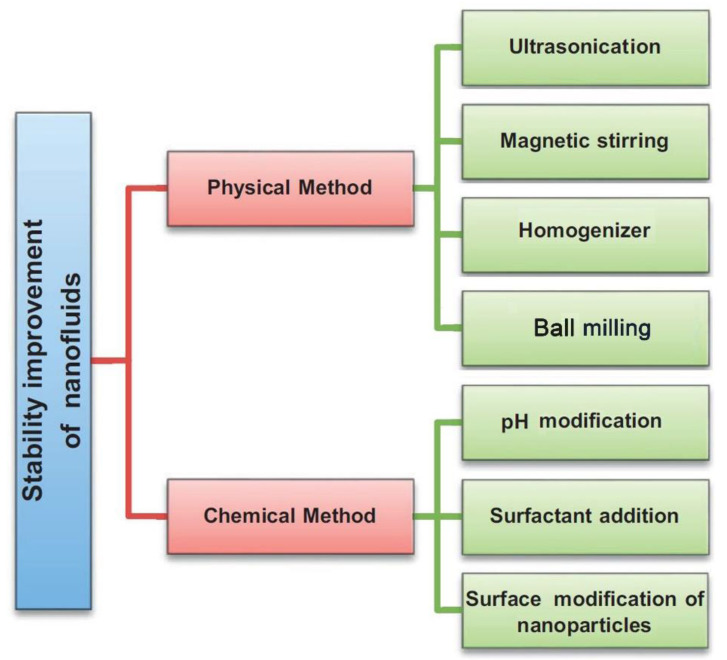
Nanofluids stability improvement methods categorized by their physical and chemical methods.

**Figure 16 nanomaterials-11-01628-f016:**
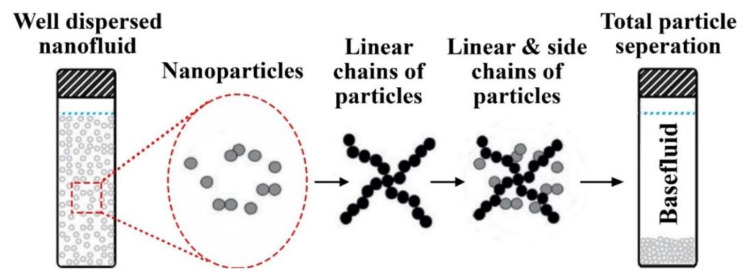
Nanoparticles separation due to the formation of both linear and side chains in the base fluid.

**Figure 17 nanomaterials-11-01628-f017:**
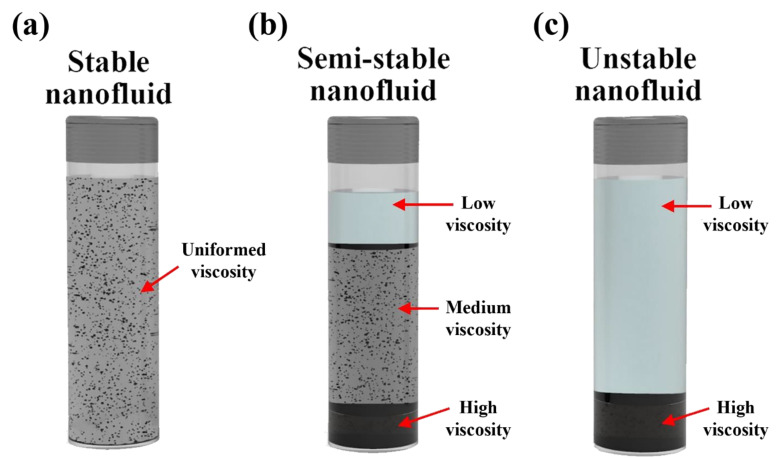
Nanofluids viscosity classification, where (**a**) shows the stable, (**b**) illustrates semistable, and (**c**) demonstrates the unstable cases of the suspension.

**Figure 18 nanomaterials-11-01628-f018:**
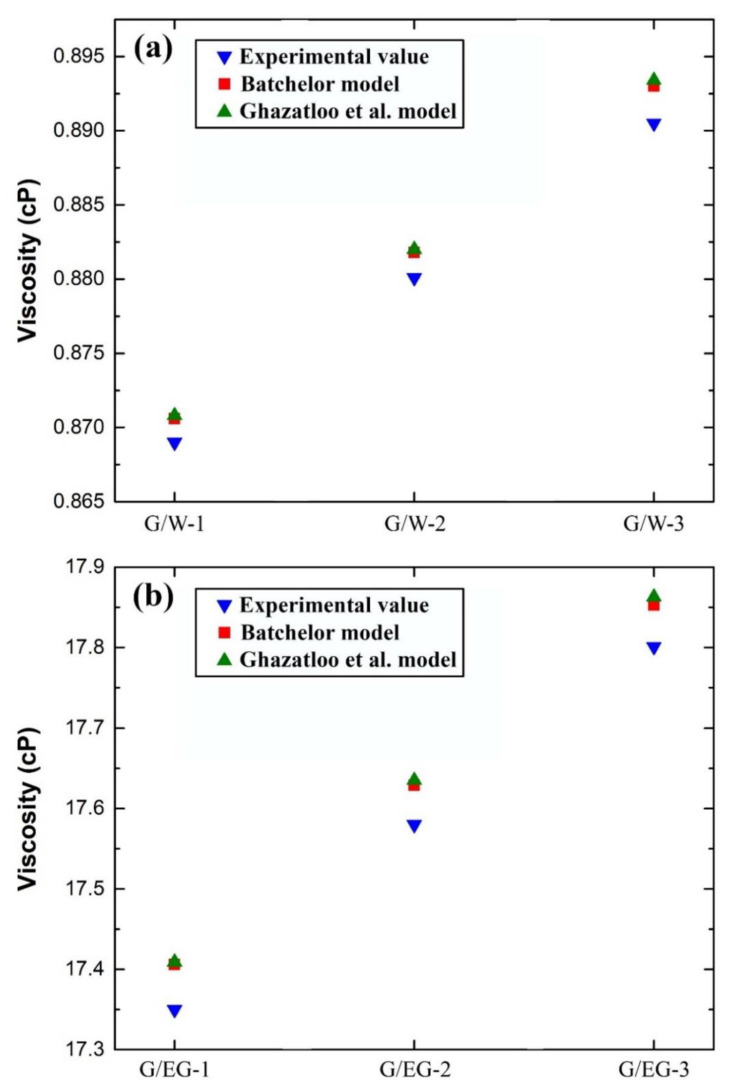
Comparison between the Ghazatloo et al. [269] model, Batchelor [270] model, and experimental effective viscosity, where (**a**) shows the results for graphene–water nanofluid of 0.5 vol. % (G/W-1), 1.0 vol. % (G/W-2), and 1.5 vol. % (G/W-3), and (**b**) illustrates the values for graphene–EG suspensions of 0.5 vol. % (G/EG-1), 1.0 vol. % (G/EG-2), and 1.5 vol. % (G/EG-3).

**Figure 19 nanomaterials-11-01628-f019:**
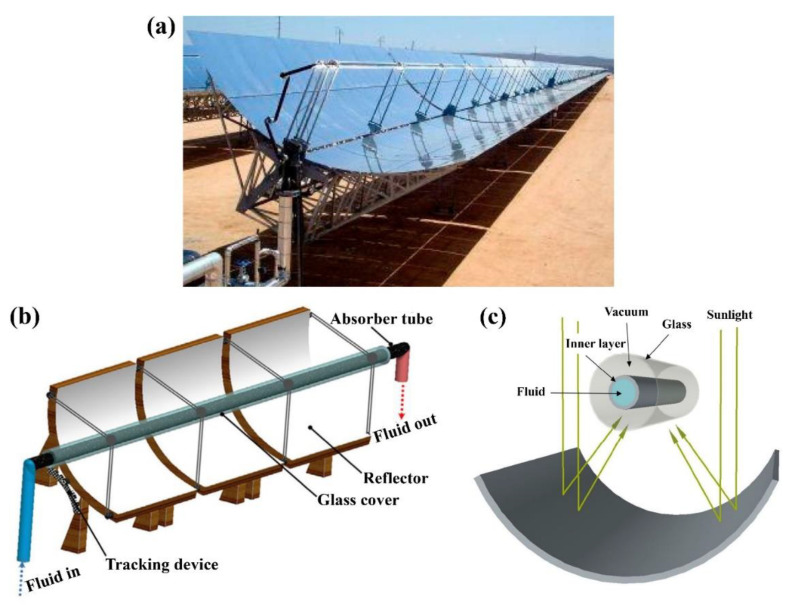
Example of a parabolic trough solar collector system, where (**a**) shows the physical device, (**b**) illustrates its schematic diagram, and (**c**) demonstrates the reflection mechanism of solar radiation on the absorber tube [298].

**Figure 20 nanomaterials-11-01628-f020:**
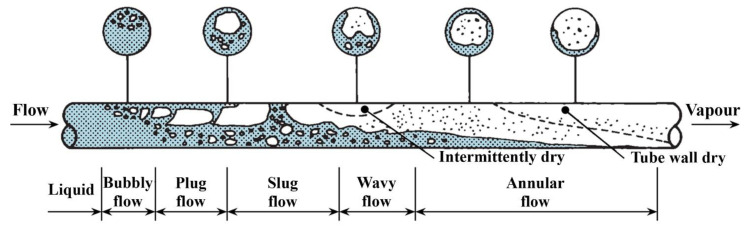
Flow boiling regimes inside a horizontal tube from liquid to vapor phases [327].

**Figure 21 nanomaterials-11-01628-f021:**
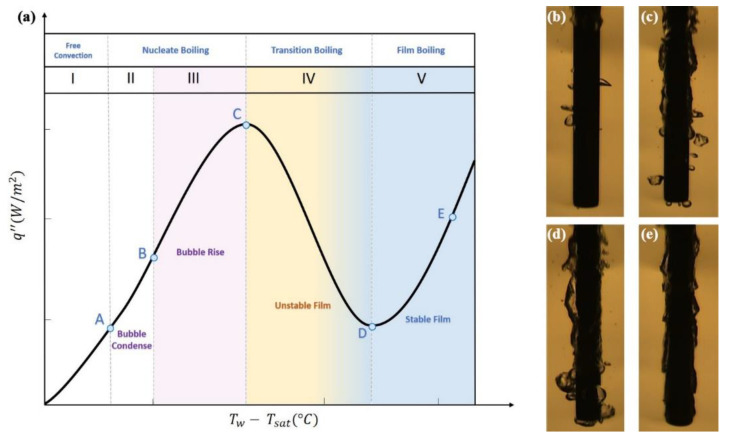
Boiling curve for stagnant water at atmospheric pressure (1 atm), where (**a**) shows the boiling curve and (**b**–**e**) illustrates the bubble formation within the free convection, nucleate boiling, transition boiling, and film boiling regimes.

**Figure 22 nanomaterials-11-01628-f022:**
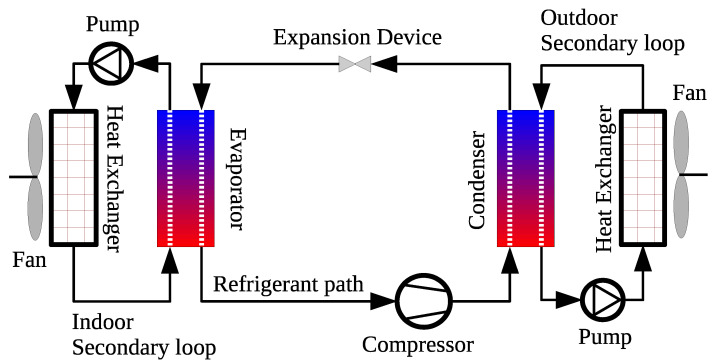
Schematic for a typical vapor compression system with secondary heating and cooling loops.

**Figure 23 nanomaterials-11-01628-f023:**
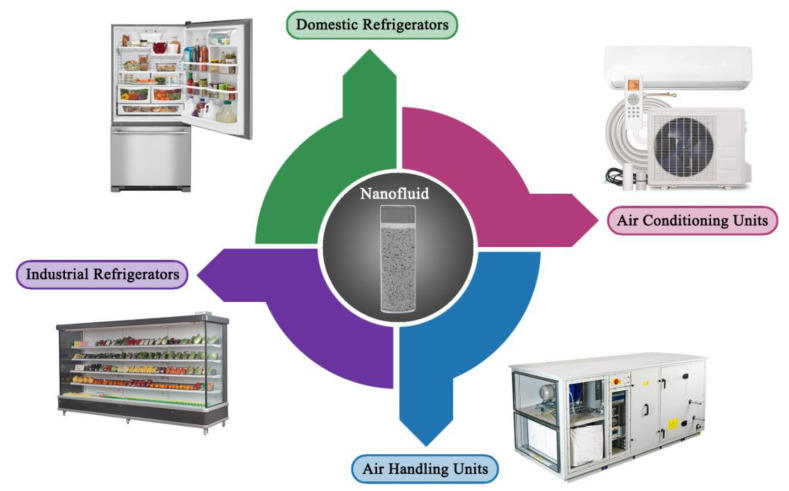
Nanofluids employment in AC&R applications, namely; air conditioning units, air handling units, industrial refrigerators, and domestic refrigerators.

**Table 1 nanomaterials-11-01628-t001:** Fraction calculation Formulae for different forms of nanofluids.

Type of Particles	Type of Base-Fluid	Fraction (%)	Formulae	Ref.	Eq.
Single type	Single type	vol.	$\frac{V_{np}}{V_{np}+V_{bf}}\times 100;$ $\mathrm{or}\;\frac{{\left(\frac{m}{\rho }\right)}_{np}}{{\left(\frac{m}{\rho }\right)}_{np}+{\left(\frac{m}{\rho }\right)}_{bf}}\times 100$	[13,37]	(1)
Single type	Two type	vol.	$\frac{{\left(\frac{m}{\rho }\right)}_{np}\;}{{\left(\frac{m}{\rho }\right)}_{np}+\left[{\left(\frac{m}{\rho }\right)}_{bf1}+{\left(\frac{m}{\rho }\right)}_{bf2}\right]\;}\times 100;$ where bf1 and bf2 have equal volume ratio	[141]	(2)
Two type	Single type	vol.	$\frac{{\left(\frac{m}{\rho }\right)}_{np1}+{\left(\frac{m}{\rho }\right)}_{np2}\;}{\left[{\left(\frac{m}{\rho }\right)}_{np1}+{\left(\frac{m}{\rho }\right)}_{np2}\right]+{\left(\frac{m}{\rho }\right)}_{bf}}\times 100;$ where np1 and np2 have equal volume ratio	[142,143]	(3)
Two type	Two type	vol.	$\frac{{\left(\frac{m}{\rho }\right)}_{np1}+\;{\left(\frac{m}{\rho }\right)}_{np2}\;}{\left[{\left(\frac{m}{\rho }\right)}_{np1}+{\left(\frac{m}{\rho }\right)}_{np2}\right]+\left[{\left(\frac{m}{\rho }\right)}_{bf1}+{\left(\frac{m}{\rho }\right)}_{bf2}\right]\;}\times 100;$ where np1 and np2 have equal volume ratio as well as bf1 and bf2	[144]	(4)

**Table 2 nanomaterials-11-01628-t002:** Published work on nanodiamond, graphene, and carbon nanotubes nanofluids produced using the two-step approach.

Material	Base Fluid	Particles Dimensions (nm)	Particles Concentration	Additional Information	Ref.
ND	EG	30–50	<1.4 vol. %	- Dispersion was performed with an ultrasonic vibration device for 3 h.	[149]
	EG	5–10	0.25–5.0 vol. %	- Purification and surface modification of the particles were done using a mixture of nitric acid, perchloric acid, and hydrochloric acid. - Dispersion was performed via continuous sonication.	[150]
	EG	5–10	0.25–1.0 vol. %	- Purification and surface modification of the particles were done using a mixture of nitric acid and perchloric acid. - Nanofluid pH adjustment: 7–10. - Dispersion was performed by magnetic stirring and ultrasonic sonication for 3 h.	[151]
	EG—water	30–50	0. 5–2.0 vol. %	- Purification and surface modification of the particles were done using a mixture of nitric acid, perchloric acid, and hydrochloric acid. - Base fluid used was a mixture of 55% distilled water and 45% of EG. - Dispersion was performed by sonication for 3 h.	[152]
	EG and mineral oil	5	2.0 g	- NDs were prepared by detonation followed by functionalization. - For the EG base fluid: the particles and 48 g of dimethylsulfoxide (DMSO) were bath sonicated for 30 min then magnetic stirred with 50 mL of glycidol for 24 h. - For the mineral oil base fluid: the particles, 2.0 g of oleic acid, and 63 g of octane were bath sonicated for 1 h	[153]
	Highly refined thermal oil	3–10	0.25–1.0 wt %	- Non-ionic sorbitane trioleate (Span 85) was used as a surfactant in a surfactant to particles ratio of 7:1. - Dispersion was performed by a probe-type sonicator for 1 h.	[154]
	Naphthenic transformer oil (NTO)	10	1.0 g	- The particles, 2.0 g of oleic acid, and 50 g of octane were high energy ultrasonicated for 30 min. - The previous mixture was added to the base fluid then sonicated for an additional 1.0 h.	[155]
	propylene glycol (PG)—water	5–10	0.2–1.0 vol. %	- The particles were initially purified then treated with acid. - The base fluid contained a mixture of PG and water at ratios of 20:80, 40:60, and 60:40, respectively. - Fabrication was performed through a bath type sonicator for 2.0 h.	[156]
Graphene	Water	2–5 *	10 mg/mL	- Graphene powder was produced through a modified hummer method (i.e., mechanical exfoliation) followed by surface treatment. - Nanofluid fabrication was done through mixture centrifugation at 6000 rpm for 10 min.	[157]
	Water	6000–8000 *	0.001–0.01 vol. %	- Graphene powder was initially oxidized using sulfuric acid and nitric acid. - Nanofluid was produced by ultrasonicating the mixture for 2.0 h.	[158]
	Water	2 *	0.025–0.1 wt %	- Graphene powder was initially oxidized using sulfuric acid and nitric acid. - Nanofluid was produced by continuous sonication using a high-power probe type ultrasonicator.	[159]
	EG and water	–	0.005–0.056 vol. %	- Fabricated graphene was treated with acid for better dispersion. - Nanofluid was produced by sonicating the mixture for 30–45 min. - Solution pH value was adjusted to around 6–7.	[160]
	Glycerol	15–50 *	13 wt %	- Graphene was surface functionalized. - Nanofluid was produced by sonicating the mixture for 10 min.	[161]
CNTs	Water	9–15 ^	0.5 wt %	- MWCNTs powder was surface functionalized via nitric and sulfuric acid of 1:3 ratio, respectively. - Nanofluid was produced by probe sonication for 5 min.	[162]
	Vegetable cutting oil	10–20 ^	0.6 vol. %	- Functionalized MWCNTs were used. - Fabrication process consisted of three mixing stages: 1—mechanical mixing for 60 min at 750 rpm, 2—ultrasonic homogenizer for 60 min, and 3—magnetic stirring for 60 min at 1500 rpm.	[163]
	Turbine meter oil	5–16.1 ^	0.05–0.4 wt %	- Triton X100 was added as a surfactant to the base fluid in a ratio of 1:3, respectively. - Fabrication process consisted of: 1—mixing the surfactant with the base fluid using an electric mixer for 20 min at 1500 rpm, 2—adding and dispersing the MWCNTs using the same device for 4 h, 3—additional mixing using a probe sonicator for 2 h.	[164]
	Water	2–4 ^	0.01–0.5 vol. %	- DWCNTs functionalized by carboxylic acid were used. - Nanofluid production was conducted by magnetic stirring for 2.5 h, followed by ultrasonication for 5 h.	[165]
	EG	2–4 ^	0.02–0.6 vol. %	- DWCNTs functionalized by carboxylic acid were used. - Fabrication was performed by magnetic stirring for 2.5 h, then sonication for 6 h.	[166]
	Water	1–2 ^	0.1–0.5 vol. %	- SWCNTs nanofluids were prepared by first adding sodium dodecyl sulfate (SDS) surfactant then mixing with a high-pressure homogenizer for 1 h.	[167]
	Water	0.8–1.6 ^	0.3 vol. %	- Nanofluid production consisted of SWCNTs, sodium deoxycholate surfactant (0.75 vol. %), and the base fluid. - Mixing was conducted by bath sonication for 6 h, followed by probe sonication for 2 h.	[168]

Note: * and ^ refers to graphene sheet thickness and CNTs outer diameter, respectively.

**Table 3 nanomaterials-11-01628-t003:** Examples of surfactants used in nanofluids fabrication categorized by classifications based on their head group charge.

Surfactant Classification	Head Group Charge	Example(s)
Cationic	+ve	Cetyltrimethyl ammonium bromide (CTAB), distearyl dimethyl ammonium chloride (DSDMAC), and benzalkonium chloride (BAC).
Non-ionic	neutral or uncharged	Oleic acid, polyvinylpyrrolidone (PVP), Arabic gum (AG), Tween 80, and oleylamine.
Anionic	−ve	Sodium dodecyl benzenesulfonate (SDBS), and SDS.
Amphoteric	+ve and −ve	lecithin.

**Table 4 nanomaterials-11-01628-t004:** Developments of nanofluids effective thermal conductivity formulas.

Developer/s	Year	Formula	Dependent Parameter	Limitations
Maxwell [231]	1890	$\frac{k_{eff}}{k_{bf}}=\frac{k_{np}+2k_{bf}+2\;f_{V}\;\left(k_{np}-k_{bf}\right)}{k_{np}+2k_{bf}-f_{V}\;\left(k_{np}-k_{bf}\right)};$ $\mathrm{where}\;k_{eff}$$,\;k_{bf}$$,\;\mathrm{and}\;k_{np}$ are the effective thermal conductivity of the nanofluid, base fluid thermal conductivity, and nanoparticles thermal conductivity, respectively.	$f_{V}$	Suited for spherical shaped particles.
Jefferson et al. [232]	1958	$k_{eff}=k_{bf}\left\{\left(1-1.21\;f_{V}{}^{2/3}\right)+0.4875\;f_{V}{}^{1/3}\;\left[\frac{\ln\frac{k_{np}}{k_{bf}}-1}{0.25+\left(0.403\;f_{V}{}^{-1/3}-0.5\right)\left(\ln\frac{k_{np}}{k_{bf}}-1\right)}\right]\right\}\;$	$f_{V}$	The model is used for spherical particles but always underestimate the effective thermal conductivity by 25%.
Hamilton and Crosser [233]	1962	$\frac{k_{eff}}{k_{bf}}=\frac{k_{np}+\left(n-1\right)k_{bf}-\left(n-1\right)\;f_{V}\;\left(k_{bf}-k_{np}\right)}{k_{np}+\left(n-1\right)k_{bf}-f_{V}\;\left(k_{bf}-k_{np}\right)}$	$f_{V}$ and n	Preferred for spherical and cylindrical shaped particles with *n* = 3/ψ, where *n* and ψ are the empirical shape factor and particle sphericity, respectively. For perfectly spherical particles ψ = 1.
Wasp et al. [234]	1977	$\frac{k_{eff}}{k_{bf}}=\frac{k_{np}+2k_{bf}-2\;f_{V}\;\left(k_{bf}-k_{np}\right)}{k_{np}+2k_{bf}+f_{V}\;\left(k_{bf}-k_{np}\right)}$	$f_{V}$	Particles should have a sphericity of ≤1.
Yu and Choi [235]	2003	$\frac{k_{eff}}{k_{bf}}=\frac{k_{np}+2k_{bf}+2\;f_{V}\;\left(k_{np}-k_{bf}\right){\left(1+\beta \right)}^{3}}{k_{np}+2k_{bf}-f_{V}\;\left(k_{bf}-k_{np}\right){\left(1+\beta \right)}^{3}}$; where β is the ratio of the nanolayer thickness to the particle radius.	$f_{V}$, interfacial particle layer, and radius	Modified version of the Maxwell [231] model for spherical particles. The main problem is that it is inadequate the non-linear trend of thermal conductivity.
Xuan et al. [236]	2003	$\frac{k_{eff}}{k_{bf}}=\frac{k_{np}+2k_{bf}-2\;f_{V}\;\left(k_{bf}-k_{np}\right)}{k_{np}+2k_{bf}+f_{V}\;\left(k_{bf}-k_{np}\right)}+\frac{f_{V}\;\rho_{np}\;C_{np}}{2k_{bf}}\;\sqrt{\frac{k_{B}T}{3\pi r_{c}}}\nu$; where $k_{B}$ is the Boltzmann constant (1.381 × 10^−23^ J/K), *T* is the temperature of the mixture, $r_{c}$ is the particle apparent radius, and ν is the kinematic viscosity of the liquid.	$f_{V}$, $\rho_{np}$, $C_{np}$, T, $r_{c}$, and ν	Hard to predict the thermal conductivity for linear temperatures.
Nan et al. [237]	2003	$\frac{k_{eff}}{k_{bf}}=\frac{3+f_{V}\left(\frac{k_{np}}{k_{bf}}\right)}{3-2f_{V}}$	$f_{V}$	Can only be used with CNTs nanofluids.
Kumar et al. [218]	2004	$\frac{k_{eff}}{k_{bf}}=1+c\frac{2k_{B}\;T{\;f}_{V}{\;r}_{m}}{\pi \;\nu \;d_{np}^{2}\;k_{bf}}\;\left(1-{\;f}_{V}\right){\;r}_{\mathrm{np}}$; - For none-spherical particles: $d_{np}=\frac{6V_{np}}{A_{np}}$; - For CNTs: $d_{np}=\frac{1.5\;ab}{a+\left(\frac{b}{2}\right)}$; where c is a constant value from 2.9 to 3.0, ${\;r}_{m}$ is the radius of the fluid medium particles, $r_{\mathrm{np}}$ is the particles radius, $d_{np}$ is the nanoparticles mean diameter, $V_{np}$ is the volume of the particles, $A_{np}$ is the area of the particles, a is the length of the CNT, and b is the outer diameter of the CNT.	$f_{V}$ dimensions of the particles, T, and *ν*	The Brownian motion has the dominative effect on the thermal conductivity prediction over all other factors.
Jang and Choi [213]	2004	$k_{eff}=k_{bf}\left(1-{\;f}_{V}\right)+k_{np}\;{\;f}_{V}+3C_{1}\left(\frac{d_{bf}}{d_{np}}\right)k_{bf}{\;f}_{V}\;Re_{d_{np}}^{2}\;Pr;$ - $Re_{d_{np}}=\frac{C_{R.M.}\;d_{np}}{\nu }$; - $C_{R.M.}=\frac{k_{B}\;T}{3\;\pi \mu d_{np}\;\ell_{bf}}$; where *C*_1_ is a proportional constant, *d_bf_* is the diameter of the base fluid molecule, $Re_{d_{np}}$ is the Reynolds number as defined above, *Pr* is the Prandtl number, *C_R.M._* is the nanoparticle random motion velocity, and $\ell_{bf}$ is the mean-free path of the base fluid molecule.	${\;f}_{V}$ dimensions of the particles, T, *ν*, and $\ell_{bf}$	Both conduction and convection heat transfer are accounted for, while the heating duration is much higher.
Yu and Choi [238]	2004	$\frac{k_{eff}}{k_{bf}}=1+\frac{nf_{V_{e}}A}{1-f_{V_{e}}A};$ - $f_{V_{e}}=r\;f_{V}$; - $A=\frac{1}{3}\;\sum_{j=a,b,c}\frac{k_{pj}-k_{bf}}{k_{pj}-\left(n-1\right)k_{bf}}\;$; where $f_{V_{e}}$ is the equivalent volume concentration of complex ellipsoids particles, r is the volume ratio, a, b, and c are the semi-axes of the particle (for sphere a=b=c), A is a parameter that reflects the equation shown above, and $k_{pj}$ is the equivalent thermal conductivity of the ellipsoids particle.	$f_{V}$, n, and interfacial resistance	This is a renovated Hamilton and Crosser [233] model with *n* = 3/ψ^−α^, where α is an empirical parameter that depends on both particle sphericity and the particle to liquid thermal conductivity ratio. In addition, this model includes the interface layer between the particles and the surrounding liquid but cannot predict the nonlinear behaviour of the thermal conductivity.
Prasher et al. [239]	2005	$\frac{k_{eff}}{k_{bf}}=\left(1+A^{\prime }\;f_{V}\;Re^{m^{\prime }}\;Pr^{0.333}\right)\;\frac{\left(1+2\alpha \right)+2\;f_{V}\left(1-\alpha \right)\;}{\left(1+2\alpha \right)-{\;f}_{V}\left(1-\alpha \right)};$ - $\alpha =\frac{2\;R_{b}\;K_{m}}{d_{np}}$; where $A^{\prime }$ is a constant that is independent of the type of base fluid, $m^{\prime }$ is a constant that depends on the base fluid type, Re is the Reynolds number, α is a parameter that reflects the equation shown above, $R_{b}$ is the impact of interfacial resistance with a magnitude in the range of 0.77 × 10^−8^ to 20 × 10^−8^ ${\mathrm{Km}}^{2}\;W^{-1},$ and *K_m_* is the matrix conductivity.	$f_{V}$, $R_{b}$, and $d_{np}$	Only considers the dispersed particles convection effect.
Xue [240]	2005	$\frac{k_{eff}}{k_{bf}}=\frac{1-{\;f}_{V}+2{\;f}_{V}\left(\frac{k_{np}}{k_{np}-k_{bf}}\right)\;\ln\left(\frac{k_{np}+k_{bf}}{2k_{bf}}\right)}{1-{\;f}_{V}+2{\;f}_{V}\left(\frac{k_{bf}}{k_{np}-k_{bf}}\right)\;\ln\left(\frac{k_{np}+k_{bf}}{2k_{bf}}\right)}$	${\;f}_{V}$	Suitable for nanofluids made of dispersed CNTs.
Murshed et al. [241]	2006	keffkbf=[1+0.27 fV43(knpkbf−1)] [1+0.52 fV1−fV13(knpkbf−1)]1+fV43(knpkbf−1) (0.52 fV1−fV13+0.27 fV13+0.27)	${\;f}_{V}$	The particles need to be uniformly dispersed in the suspension for appropriate effective thermal conductivity prediction.
Vajjha et al. [242]	2010	$k_{eff}=\frac{k_{np}+2k_{bf}-2\left(k_{bf}-k_{np}\right){\;f}_{V}}{k_{np}+2k_{bf}\;\left(k_{bf}-k_{np}\right){\;f}_{V}}\;k_{bf}+5\times 10^{4}\beta {\;f}_{V}\;\rho_{bf}C_{p_{bf}}\sqrt{\frac{k_{B}\;T}{\rho_{np}\;d_{np}}}\;ƒ\;\left(T,{\;f}_{V}\right);$ - $f\;\left(T,{\;f}_{V}\right)=\left(2.8217\times 10^{-2}{\;f}_{V}+3.917\times 10^{-3}\right)\left(\frac{T}{T_{o}}\right)+\left(-3.0669\times 10^{-2}{\;f}_{V}-3.91123\times 10^{-3}\right)$; where β is the fraction of the liquid volume that moves with the particle, $ƒ\;\left(T,\;f_{V}\right)$ is a function that depends on the fluid temperature and particles concentration as defined above, and $T_{o}$ is a reference temperature that equals 273 K	${\;f}_{V}$, particles type, and base fluid temperature	Limited to nanofluids of temperatures between 295 and 363 K.
Xing et al. [243]	2016	$k_{eff}=\left(1+\frac{{\;\eta^{\prime }\;f}_{V}}{\frac{3k_{bf}}{\;\eta^{\prime }k_{33}^{c}}+3H\left(\;\eta^{\prime }P\right)}\right)k_{bf}+0.5{\;f}_{V}\;\rho_{CNT}\;C_{p_{CNT}}\;\sqrt{\frac{k_{B}\;T}{3\;\pi \;\mu \;r_{m}}};$ - $\;\eta^{\prime }=\left[\left(2\times 10^{8}\right)a^{2}-13.395\;a+0.2533\right]{\;f}_{V}{}^{-\left(6988.1\;a+0.1962\right)}\;$; - $H=\frac{1}{P^{2}-1}\;\left[\frac{P}{\sqrt{P^{2}-1}}\;\ln\left(P+\sqrt{P^{2}-1}\right)-1\right];$ - $P=\frac{a}{b}$; - $k_{33}^{c}=\frac{k_{np}}{1+\frac{2R_{k}\;k_{np}}{a}}$; where $\eta^{\prime }$ is the modified straightness ratio, *H* is a factor reflected by the equation defined above, *P* is the CNT length to diameter ratio, $k_{33}^{c}$ is the equivalent thermal conductivity of the CNT along the longitudinal axes, $R_{k}$ is the Kaptiza radius and is equal to 8 × 10^−8^ m^2^ K/W, μ is the dynamic viscosity, $\rho_{CNT}$ is the density of the CNT, and is the $C_{p_{CNT}}$ specific heat capacity of the CNT.	$f_{V}$, T, and aspect ratio	Can only be used for CNTs suspensions. Furthermore, not all of the parameters are accounted in the correlation, while the effect of the micro-motion is the most significant parameter.
Gao et al. [244]	2018	$\frac{k_{eff}}{k_{bf}}=\frac{3+\eta^{2}{\;f}_{V}\;}{\left[k_{bf}\;\left(\frac{2\;R_{b}}{L}+13.4\sqrt{t}\right)\right]\left(3-{\eta \;f}_{V}\right)}\;;$ where L is the length of the nanoplatelet, *t* is the nanoplatelet thickness, and η is the average flatness ratio of the graphene nanoplatelet.	${\;f}_{V}$, L, t, $R_{b}$, and η.	This model is designed for suspensions of water, as the base fluid, and graphene nanoplatelet.
Li et al. [245]	2019	$\frac{k_{eff}}{k_{bf}}=\frac{k_{pe}+2k_{bf}+2\left(k_{pe}-k_{bf}\right){\left(1-\frac{t_{nl}}{r_{np}}\right)}^{3}{\;f}_{V}}{k_{pe}+2k_{bf}-\left(k_{pe}-k_{bf}\right){\left(1-\frac{t_{nl}}{r_{np}}\right)}^{3}{\;f}_{V}}\;;$ where $k_{pe}$ is the equivalent particle thermal conductivity, and $t_{nl}$ is the thickness of the nanolayer surrounding the particle.	$f_{V}$, $t_{nl}$, $r_{np}$, and fluid temperature	This model is a modified form of the Yu and Choi model with the nanolayer constant value changed to quadratic.
Jóźwiak et al. [246]	2020	$\frac{k_{eff}}{k_{bf}}=\frac{{\;\omega f}_{V}\left(k_{np}-\omega k_{bf}\right)\left(\gamma_{1}^{2}-\gamma^{2}+1\right)+\left(k_{np}+\omega k_{bf}\right)\gamma_{1}^{2}\left[{\;f}_{V}\gamma^{2}\left(\omega -1\right)+1\right]}{\gamma_{1}^{2}\left(k_{np}+\omega k_{bf}\right)-\left(k_{np}-\omega k_{bf}\right){\;f}_{V}\;\left(\gamma_{1}^{2}+\gamma^{2}-1\right)}\;;$ - $\omega =\frac{k_{IN}}{k_{bf}}$; - $\gamma =1+\frac{t_{nl}}{r_{CNT}}$; - $\gamma_{1}=1+\frac{t_{nl}}{2r_{CNT}}$; where ω, γ, $\gamma_{1}$ are factors representing the equations shown above, $k_{IN}$ is the interfacial nanolayer thermal conductivity, and $r_{CNT}$ is the radius of a single CNT.	${\;f}_{V}$, and particles morphology	This is a modified version of the Murshed et al. [241] model, which is suitable for ionic liquid nanofluids (also known as ionanofluids) with dispersed CNTs.

**Table 5 nanomaterials-11-01628-t005:** Developments of nanofluids effective viscosity formulas.

Developer/s	Year	Formula	Dependent Parameter	Limitations
Einstein [271]	1906	${µ}_{eff}={µ}_{bf}\left(1+2.5f_{V}\right)$	$f_{V}$	Suited for suspensions of <0.02 vol. % and spherical shaped particles.
Hatschek [272]	1913	${µ}_{eff}={µ}_{bf}\left(1+2.5f_{V}\right)$	$f_{V}$	Designed for suspensions with up to 40 vol. % of spherical particles but does not account for the size of the dispersed particle. The formula also showed very large deviation from the actual viscosity value.
Saitô [273]	1950	${µ}_{eff}={µ}_{bf}\left(1+\frac{1.25f_{V}}{1-\frac{f_{V}}{0.87}}\right)$	$f_{V}$	Preferred for dispersions of small spherical particles and is affected by the Brownian motion of the particles.
Mooney [274]	1951	${µ}_{eff}={µ}_{bf}\;\exp\left(\frac{2.5f_{V}}{1-CFf_{V}}\right);$ where CF is the self-crowding factor.	$f_{V}$, and CF	This is an extended version of the Einstein’s [271] formula that can be used for suspensions of spherical particles with any concentration. The downside is that the modeled suspension needs to meet the functional equation so that the ${µ}_{eff}$ can be independent of the stepwise sequence of adding further particles concentrations.
Brinkman [275]	1952	${µ}_{eff}={µ}_{bf}{\left(1-f_{V}\right)}^{-2.5}$	$f_{V}$	Enhanced form of the previous Einstein [271] formula, where it can be used for particles concentrations of up to 4 vol. %.
Roscoe [276]	1952	${µ}_{eff}={µ}_{bf}{\left(1-Sf_{V}\right)}^{S^{\prime }}$; where S is a constant that is equal to 1 (for very diverse particles sizes), −2.5 (for similar particles sizes and <0.05 vol. %), and 1.35 (for higher vol. %); and $S^{\prime }$ is a constant that is equal to −2.5 (for the very diverse particles sizes case and the >0.05 vol. % suspension) and 1 (for the <0.05 vol. % of similar sized particles).	$f_{V}$	Can be used with any dispersion concentration but the particles need to be of spherical shape.
Maron and Pierce [277]	1956	${µ}_{eff}={µ}_{bf}{\left(1-\frac{f_{V}}{f_{p}}\right)}^{-2};$ where $f_{p}$ is the packing fraction of the particles.	$f_{V}$, and $f_{p}$	Suitable for suspensions of small spherical particles and of similar sizes.
Krieger and Dougherty [278]	1959	${µ}_{eff}={µ}_{bf}{\left(1-\frac{f_{V}}{f_{p}}\right)}^{-2.5{\;f}_{p}\;}$;	$f_{V}$, and $f_{p}$	For dispersed spherical particles of ≤0.2 vol. %, but the model does not account for the particle’s interfacial layers and their aggregation.
Frankel and Acrivos [279]	1967	${µ}_{eff}={µ}_{bf}\left(\frac{9}{8}\right)\left[\frac{{\left(\frac{f_{V}}{f_{m}}\right)}^{\frac{1}{3}\;}}{1-{\left(\frac{f_{V}}{f_{m}}\right)}^{\frac{1}{3}\;}}\right]$; where $f_{m}$ is the maximum attainable concentration.	$f_{V}$	Employed for uniform spherical particles and assumes that the rise in viscosity with the increase in particles concentration is due to their hydrodynamic interactions.
Nielson [280]	1970	${µ}_{eff}={µ}_{bf}\;\exp\left(\frac{f_{V}}{1-f_{p}}\right)$	$f_{V}$, and $f_{p}$	This is a modified form of the Einstein’s [271] formula but it lacks accurate suspension viscosity prediction.
Brenner and Condiff [281]	1974	${µ}_{eff}={µ}_{bf}\;\left[1+f_{V}\left(2+\frac{0.312s}{\ln2s-1.5}-\frac{0.5}{\ln2s-1.5}-\frac{1.872}{s}\right)\right];$ where s is the axis aspect ratio of the dispersed particle.	${\;f}_{V}$, aspect ratio, and shear rate	Shows good prediction capability for dispersed particles of rod shape but less effective for other shapes.
Jeffrey and Acrivos [282]	1976	${µ}_{eff}={µ}_{bf}\;\left[3+\frac{4}{3}\left(\frac{s^{2}f_{V}}{\ln\frac{\pi }{f_{V}}}\right)\right]$	${\;f}_{V}$, and aspect ratio	Designed for suspensions of rod-shaped particles.
Batchelor [270]	1977	${µ}_{eff}={µ}_{bf}\left(1+2.5f_{V}+6.2\;f_{V}{}^{2}\right)$	${\;f}_{V}$, and Brownian motion	The model includes the interaction between the particles but fails to provide good prediction agreement.
Graham [283]	1981	${µ}_{eff}={µ}_{bf}\left(\frac{9}{4}\right)\;{\left[1+\left(\frac{h}{\;d_{np}}\right)\right]}^{-1}\left[\frac{1}{\left(\frac{h}{0.5\;d_{np}}\right)}-\frac{1}{1+\left(\frac{h}{0.5\;d_{np}}\right)}-\frac{1}{{\left[1+\left(\frac{h}{0.5\;d_{np}}\right)\right]}^{2}}\right]+\left[1+\left(\frac{5}{2}\right){\;f}_{V}\right];$ where h is the minimum separation distance between the surface of two spherical particles.	${\;f}_{V}$, $d_{np}$, and h	Suitable for spherical particles only and has good prediction agreement with Einstein [271] formula when very low particles concentrations are used or when ${µ}_{eff}$ is very close to that of ${µ}_{bf}$.
Kitano et al. [284]	1981	${µ}_{eff}={µ}_{bf}{\left(1-\frac{f_{V}}{f_{p}}\right)}^{-2}$	$f_{V}$, and $f_{p}$	Similar to the Maron and Pierce [277] formula but the $f_{p}$ value is preliminarily defined numerically and is better suited for two phase mixtures.
Bicerano et al. [285]	1999	${µ}_{eff}={µ}_{bf}\left(1+\left[\eta \right]{\;f}_{V}+k_{H}\;f_{V}{}^{2}\right);$ where [η] is the intrinsic viscosity, and $k_{H}$ is the Huggins coefficient.	${\;f}_{V}$, [η], and $k_{H}$	More determined towards analyzing the relation between particles concentration and ${µ}_{eff}$.
Wang et al. [286]	1999	${µ}_{eff}={µ}_{bf}\left(1+7.3{\;f}_{V}+123\;f_{V}{}^{2}\right)$	${\;f}_{V}$	Simple model that was formed from a set of experimental results obtained from modifying the suspension particles size and concentration.
Masoumi et al. [248]	2009	${µ}_{eff}={µ}_{bf}+\frac{\rho_{np}}{72\delta \;Fun.}\left(\frac{1}{2r_{np}}\sqrt{\frac{18k_{B}T}{2\pi \rho_{np}r_{np}}}\right)\left(2r_{np}{}^{2}\right);$ - $\delta =2r_{np}\left(\sqrt[{3}]{\frac{\pi }{6{\;f}_{V}}}\right)$; where δ is the distance between the particles, and Fun. is a correction function.	${\;f}_{V}$, T, $\rho_{np}$, particle size, and Brownian motion	The formula is bound by the experimental conditions that were used in its development.
Chevalier et al. [250]	2009	${µ}_{eff}={µ}_{bf}{\left[1-\frac{f_{V}}{f_{p}}\;{\left(\frac{D_{a}}{2r_{np}}\right)}^{3-d_{f}}\right]}^{-2};$ where $D_{a}$ is the average diameter of the aggregates, and $d_{f}$ is the fractal dimension, which depends on the shape of the dispersed particles, the type of agglomeration, and the shear flow. $f_{p}$ and $D_{a}$ are usually set to 0.65, for random packing of spheres, and 1.8, respectively.	$f_{V}$, $f_{p}$, $r_{np}$, and $d_{f}$	This model depends on the agglomerate size, and thus it is not optimum for determining the ${µ}_{eff}$ for stabile suspensions.
Chandrasekar et al. [190]	2010	${µ}_{eff}=1-Coef._{1}\;{\left(\frac{f_{V}}{1-f_{V}}\right)}^{Coef._{2}};$ where $Coef._{1}$ and $Coef._{2}$ are regression coefficients that can be obtained from preliminary experimental results.	Specific area, $\rho_{np}$, $\rho_{nf}$, and sphericity of the particles	Depends on preliminary experimental results to set-up the unknown coefficients.
Bobbo et al. [287]	2012	${µ}_{eff}={µ}_{bf}\left(1+Coef._{1}{\;f}_{V}+Coef._{2}\;f_{V}{}^{2}\right)$	${\;f}_{V}$, and $r_{np}$	Developed for single-walled carbon nanohorn (SWCNH) and TiO_2_ nanofluids based on the Batchelor formula and experimental measurements of the ${µ}_{eff}$ at a range of temperatures from 283.2 to 353.2 K, and concentrations from 0.01 to 1 wt %.
Esfe et al. [288]	2014	${µ}_{eff}={µ}_{bf}\left(1.1296+38.158{\;f}_{V}-0.0017357\;T\right)$	${\;f}_{V}$, and T	Limited for water based MWCNTs nanofluids of 0–1 vol. %.
Aberoumand et al. [289]	2016	${µ}_{eff}={µ}_{bf}\left(1.15+1.061{\;f}_{V}-0.5442\;f_{V}{}^{2}+0.1181\;f_{V}{}^{3}\right)$	${\;f}_{V}$	Used for low temperature oil based suspensions.
Akbari et al. [290]	2017	${µ}_{eff}={µ}_{bf}\left(-24.81+3.23\;T^{0.08014}\exp\left(1.838{\;f}_{V}{}^{0.002334}\right)-0.0006779\;T^{2}+0.024\;f_{V}{}^{3}\right)$	${\;f}_{V}$, and T	Suitable for nanofluids of <3 vol. % and of temperature ≤50 °C.
Esfe et al. [291]	2019	${µ}_{eff}=6.35+2.56\;f_{V}-0.24\;T-0.068\;Tf_{V}+0.905\;f_{V}{}^{2}+0.0027{\;T}^{2}$	${\;f}_{V}$, and T	Suitable for MWCNTs and TiO_2_ hybrid nanofluids of ${\;f}_{V}$ between 0.05 and 0.85 vol. %.
Ansón-Casaos et al. [292]	2020	${µ}_{eff}={µ}_{bf}{\left(1-\frac{\chi }{2}f_{V}\right)}^{-2};$ where χ is equal to 2.5 for spherical particles or can be replaced by a function, $f\left(r_{np}\right)$, to determine the suspension property containing 1D and 2D dispersed solids.	${\;f}_{V}$, and χ	Suitable for SWCNTs and graphene oxide.
Ilyas et al. [154]	2020	${µ}_{eff}={µ}_{bf}\;\exp\left(\frac{FP._{1}}{T-FP._{2}}\right)+FP._{3}f_{V}\;\exp\left(\frac{FP._{4}}{T}\right)-FP._{5}f_{V}{}^{2};$ where $FP._{1}$, $FP._{2}$, and $FP._{4}$ are the temperature fitting parameters in Kelvin, whereas $FP._{3}$ and $FP._{5}$ are the dynamic viscosity fitting parameters in Pa.s. The values of these parameters (i.e., $FP._{1}$ to $FP._{5}$) can be found in the published source.	${\;f}_{V}$, FP and T	Suitable for ND dispersed in thermal oil and is valid for the range of 0 ≤ $f_{V}$ ≤ 1 and 298.65 ≤ T (K) ≤ 338.15.

**Table 6 nanomaterials-11-01628-t006:** Examples of different industrial processes that utilize parabolic trough solar collector systems and their temperature requirements.

Industry	Process	Required Temperature Range (°C)
Dairy	Boiler feed water	60–90
Agricultural products	Drying	80–200
Textile	Drying	100–130
Chemistry	Petroleum	100–150
Desalinization	Heat transfer fluid	100–250

**Table 7 nanomaterials-11-01628-t007:** Summary of selected CHF enhancement for various pool boiling studies in water base.

Ref.	Nanofluid	Concentration/Particle Size	Heating Surface	CHF Enhancement%
[370]	CNT	0.1–0.3 wt %	–	Enhanced
[371]	CNT	0.01–0.05 vol. %	Cu block	38.2
[372]	CNT	0.5–4 wt %	Cu plate	60–130
[361]	CNT	0.05 vol. %	SS foil	108 122
[373]	CNT	1.0 vol. %	SS tube	29
[374]	MWCNT	0.01–0.02 wt %	SS cylinder	Enhanced
[371]	MWCNT	0.0001–0.05 vol. %	Cu block	200
[375]	MWCNT	0.1–0.3 wt %	Microfin Cu disk	95
[372]	f-MWCNT	0.5–4 wt %	Cu plate	200
[162]	f-MWCNT	0.25–1 wt %	SS tube	37.5
[376]	f-MWCNT	0.01–0.1 wt %	Cu disk	271.9
[377]	f-MWCNT	0.01 vol. %	Cu block	98.2
[378]	f-SWCNT	2.0 wt	Ni-Cr wire	300
[379]	GO	≤0.001 wt %	Copper plate	Enhanced
[380]	GO	0.0005 wt %	Ni-Cr wire	320
[381]	GO	0.001 vol. %	–	179
[382]	GO	0.0001, 0.0005, 0.0010, and 0.005 wt %	Ni wire	Enhanced
[383]	GO	0.01 vol. %	Ni-Cr wire	–
[366]	ND	1 g/L	Cu plate	Enhanced
[366]	ND	<1 g/L	Cu plate	Deterioration
[384]	ND	0.01–0.1 vol. %	SS plate	Unchanged
[384]	ND	0.01 vol. %	SS plate	11

Note: f-SWCNT, and f-MWCNT refer to functionalized SWCNT, and functionalized MWCNT, respectively.

**Table 8 nanomaterials-11-01628-t008:** Summary of selected HTC enhancement for various pool boiling studies in water base.

Ref.	Nanofluid	Concentration	Heating Surface	CHF Enhancement%
[373]	MWCNT	1.0 vol. %	Cu block	28.7
[371]	MWCNT	0.0001–0.05 vol. %	Cu block	38.2
[385]	MWCNT	0.25%, 0.5%, and 1.0 vol. %	Ni-Cr wire	320
[375]	MWCNT	0.1–0.3 wt %	Microfin Cu disk	77
[372]	f-MWCNT	0.5–4 wt %	Cu plate	130
[162]	f-MWCNT	0.25–1 wt %	SS tube	66
[376]	f-MWCNT	0.01–0.1 wt %	Cu disk	38.5
[377]	f-MWCNT	0.01 vol. %	Cu block	10.15
[386]	Graphene	0.1 and 0.3 wt %	Cu	96

**Table 9 nanomaterials-11-01628-t009:** Summary of selected T_min_ enhancement for various pool boiling studies in water based nanofluids.

Ref.	Nanomaterial(s)	Heating Surface	$T_{min,water}\;({}^{\circ}C)$	$T_{min,nanofluid}\;({}^{\circ}C)$
[387]	ND (0.01 vol. %)	ITO	230	260
[388]	CNT-1 CNT-2 CNT-3 CNT-4 (0.5 wt.%)	316L SS sphere	215 218 218 219	241, 294, 303, 328, 335 211, 229, 277, 281, 287 228, 246, 254, 262, 264 231, 238, 243, 254, 256
[389]	GO (0.0001 wt %)	SS sphere	230	236.1
	GO (0.001 wt %)			239.6
	GO (0.005 wt %)			235.7
	GO (0.01 wt %)			235.6
	GO (0.05 wt %)			233.1
	GO (0.1 wt %)			235.9
[390]	Al_2_O_3_ SiO_2_ ND (0.1 vol. %)	SS	249, 247, 249, 247, 250, 250, 251	244, 343, 345, 394, 348, 399, 389 251, 330, 368, 368, 377, 389, 397 252, 252, 250, 253, 255, 264, 279
	Al_2_O_3_ SiO_2_ ND (0.1 vol. %)	Zr	267, 272, 253, 272, 260, 266, 253	287, 347, 354, 400, 401, 411, 412 282, 323, 362, 372, 415 278, 275, 269, 269, 274, 283, 272

**Table 10 nanomaterials-11-01628-t010:** List of studies related to carbon-based nanoparticles effect with working fluid in AC&R systems.

Reference	Nanofluid	Test Conditions	Nanoparticle		
			Concentration	Diameter (nm)	Length (µm)
Park and Jung [435]	CNT–R-123 CNT–R-134a	Heat Flux 20–60 kW/m^2^	1.0 vol. %	20	1
Zhang et al. [436]	MWCNT–R-123	Heat Flux –	0.02–0.20 vol. %	30–70	2–10
Sun et al. [437]	MWCNT–R-141b	Mass flux 100 to 350 kg/(m^2^s)	0.059, 0.117 and 0.176 vol. %	8	10–30
Jiang et al. [438]	CNT–R-113	Temperature 303 K	0.2–1.0 vol. %	15–80	1.5–10
Peng et al. [439]	CNT–POE–R-113	Heat Flux 10–80 kW/m^2^	0.1–1 wt %	15–80	1.5–10
Ahmadpour et al. [440]	MWCNT–mineral oil–R-600A	Heat Flux –	0.1-.3 wt %	5–15	50
Kumaresan et al. [441]	MWCNT–EG–water	Temperature 273–313K	0.15–0.45 vol. %	30–50	10–20
Baskar et al. [443]	MWCNT–propanol + isopropyl alcohol	Temperature 273–303K	0.15–0.3 vol. %	–	–
Wang et al. [444]	Graphene–EG	Temperature 328–333K	0.01–1 wt %	–	5–15
Lin et al. [445]	MWCNT–R-141b	–	250–750 mg/L	15–80	1.5–10
Alawi and Sidik [446]	SWCNT–R-134a	Temperature 300–320 K	1.0–5.0 vol. %	20	–
Dalkilic et al. [447]	MWCNT–POE	Temperature 288–323 K	0.01–0.1 wt %	10–30	–

**Table 11 nanomaterials-11-01628-t011:** List of studies related to carbon-based nanoparticles effect on AC&R systems performance.

Ref.	Nanofluid	Nanoparticle			Compressor Discharge Temperature	Compressor Power	Cooling Capacity	COP
		Concentration	Diameter (nm)	Length (µm)				
Abbas et al. [449]	CNT–POE–R-134a	0.01–0.1 wt %	–	–	–	Reduced by 2.2%	–	Improved by 4.2%
Jalili et al. [450]	MWCNT–water	0–2000 ppm	10–20	5–15	–	–	–	–
Kamaraj and Manoj Babu [454]	CNT–POE–mineral oil–R-134a	0.1 and 0.2 g/L	13	–	Negligible reduction	Negligible reduction	Improved by 16.7%	Improved by 16.7%
Vasconcelos et al. [453]	MWCNT–water–R-22	0.035–0.212 vol. %	1–2	5–30	–	Negligible reduction	Improved by 22.2%	Improved by 27.3–33.3%
Pico et al. [456]	ND–POE–R-410A	0.1 and 0.5 mass %	3–6	–	Reduced by 3–4 °C	Negligible reduction	Improved by 4.2–7%	Improved by 4–8%
Pico et al. [457]	ND–POE–R-32	0.1 and 0.5 mass %	3–6	–	Reduced by 1.2–2 °C	Negligible reduction	Improved by 1–2.4%	Improved by 1–3.2%
Yang et al. [455]	Graphene–SUNISO 3GS–R-600a	10, 20, and 30 mg/L	100–3000	–	Reduced by 2.5–4.6%	Reduced by 14.8–20.4%	Improved by 5.6%	–
Rahman et al. [458]	SWCNT–R-407c	5 vol. %	–	–	–	Reduced by 4%	–	Improved by 4.3%

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
