# Peer review of "Carbon-Based Nanofluids and Their Advances towards Heat Transfer Applications—A Review"

_nanomaterials, 2021, doi:10.3390/nano11061628_

Round 1

Reviewer 1 Report

This paper shows a nice review of Carbon-Based Nanofluids. There are some issues that need to address:

- 512 References! Fantastic!

- what makes this review different from the other and from the most recent ones?

- section of drawbacks and future could be increased quality of the manuscript.

- There are some grammatical errors, please carefully check the whole manuscript.

Congratulations to the authors for their professional work. This article can be very useful for researchers and students.

Author Response

This paper shows a nice review of Carbon-Based Nanofluids. There are some issues that need to address:

- 512 References! Fantastic!

The authors thank the respected reviewer for his kind words and for supporting us. Thank you very much.

What makes this review different from the others and from the most recent ones?

Thank you for your question. What makes this review different is that it guides the reader all the way from the carbon-based nanomaterials fabrication and all the way towards three major thermal applications in the industry and it provides the recent gap in the scientific knowledge, where the reader can tackle in his future research.

Section of drawbacks and future could be increased quality of the manuscript.

We thank the respected reviewer for his comment. We have taken this into consideration in our revised version. Kindly check Section 8.1 (Page 83 of 125) and Section 8.3 (Page 85 of 125) in our revised manuscript. Thank you very much.

There are some grammatical errors, please carefully check the whole manuscript.

We apologize for any inconveniences. We have revised the English language of the whole manuscripts in its revised version. Thank you very much.

Congratulations to the authors for their professional work. This article can be very useful for researchers and students.

The authors would like to thank the respected reviewer very much for his time and for his encouragements.

Thank you very much.

Reviewer 2 Report

Accept

Author Response

The authors would like to thank the respected reviewer very much for the time and kind support.

Thank you very much.

The authors

Reviewer 3 Report

Heat transfer nanofluids is very interesting and challenging field. Authors try to give a comprehensive review about carbon based nanofluids and their advances towards heat transfer applications. The efforts are worthwhile.  

However, it seems there are more references about carbon based nanofluids. Authors may want to cite and give summary and comments for the manuscript.

1. Enhanced thermal conductivity by the magnetic field in heat transfer nanofluids containing carbon nanotube,   Haiping Hong, Brian Wright, Jesse Wensel, Sungho Jin, Xiang Rong Ye, Walter Roy, Synthetic Metals, 157 (2007) 437-440.

2. Magnetic field enhanced thermal conductivity in heat transfer nanofluids containing Ni-coated single wall carbon nanotubes, Brian Wright , Dustin Thomas, Haiping Hong , Lori Groven, Jan Puszynski, Duke Edward, Xiangrong Ye , Sungho Jin, Applied Physics Letter, 91, 173116, 2007

3. Enhanced thermal conductivity by aggregation in heat transfer nanofluids containing metal oxide and carbon nanotube, Jesse Wenzel, Brian Wright, Dustin Thomas, Wayne Douglas, Bert Mannhalter, William Cross, Haiping Hong, Jon Kellar, Pauline Smith, Walter Roy, Applied Physics Letter, 92, 023110, 2008

4.  Alignment of single wall carbon nanotubes comprising magnetically sensitive metal oxides in heat transfer nanofluids, Haiping Hong, Xinning Luan, Mark horton, Chen Li,  GP Peterson, Thermochimica Acta, 525, 87-92, 2011

5. A Novel Approach to Fabricate Carbon Nanomaterials-Nanoparticle Solids through Aqueous Solutions and their Applications, Younes, H. Hong, H. and Peterson, G. P., Nanomanufacturing and Metrology, DOI, 10.1007/s41871-020-00094-z

Author Response

We would like to thank the respected reviewer for his respected comment and valuable advice and time. Kindly note that the authors have taken into account all of the respected reviewer suggestions in their revised version of the manuscript. Kindly check Pages 38 to 39, Lines 608 to 627 in the revised manuscript. Please note that all changes were made with red text.

Thank you very much.

The Authors

This manuscript is a resubmission of an earlier submission. The following is a list of the peer review reports and author responses from that submission.

Round 1

Reviewer 1 Report

  1. The reviewer thinks English is not the authors’ first language. The quality of the writing is needed to improve. The wrong structure, as well as punctuation in some sentences, prevents proper understanding.
  2. Please obtain permissions for figures’ reproduction.
  3. It will e more informative if a part of the paper covers the biofuels containing nanomaterials.
  4. Some references are older than 2015 and therefore, they are abolished. May the reviewer ask the authors to change these references to newer ones? Some suggested papers are as below: Heat transfer evaluation of a micro heat exchanger cooling with spherical carbon-acetone nanofluid. Effects on thermophysical properties of carbon based nanofluids: experimental data, modelling using regression, ANFIS and ANN.  Study of synthesis, stability and thermo-physical properties of graphene nanoplatelet/platinum hybrid nanofluid.  Investigation of heat transfer performance and friction factor of a counter-flow double-pipe heat exchanger using nitrogen-doped, graphene-based nanofluids.  A novel comprehensive experimental study concerned graphene oxide nanoparticles dispersed in water: Synthesise, characterisation, thermal conductivity measurement and present a new approach of RLSF neural network.
  5. When reviewing the references, a strong impression can be created that the manuscript should be submitted to another journal: To give journal readers a sense of continuity, the reviewer encourages the authors to identify Nanomaterials publications of similar research in your papers. Please, do a literature check of the papers published in recent years (2020 and even 2021) and relate the content of relevant papers to the results and findings presented in your publication. The authors can also reference articles in print using their DOI.
  6. Add some critical results to the introduction part.
  7. Refine the keywords.
  8. Carbon-based nanomaterials have a vast application in enhancing the quality of biofuels. To know more about the topic, you study 2020 & 2021 related papers like: Investigation on the effect of cottonseed oil blended with different percentages of octanol and suspended MWCNT nanoparticles on diesel engine characteristics. Effect of nano-graphene oxide and n-butanol fuel additives blended with diesel—Nigella sativa biodiesel fuel emulsion on diesel engine characteristics.
  9. The conclusion part should be summarized.
  10. Add a part to the article and talk about the environmental issues of carbon-based nanomaterials. You can see this paper for more information: Recent Advances in Using Nanofluids in Renewable Energy Systems and the Environmental Implications of their Uptake.
  11. Do you really think Fig.2 is necessary for the manuscript?

Author Response

The reviewer thinks English is not the authors’ first language. The quality of the writing is needed to improve. The wrong structure, as well as punctuation in some sentences, prevents proper understanding.

We thank the respected reviewer for the provided comment. Kindly note that the English language was proofed in the revised version of the manuscript. Thank you very much.

Please obtain permissions for figures’ reproduction.

Figure permissions were obtained as following:

Fig.1 – No copy rights issue – Made By Authors

Fig.2 – No copy rights issue – Made By Authors

Fig.3 – No copy right issue – Made By Authors

Fig.4 – No copy rights issue – Open Access Source

Fig.5 – No copy rights issue – Made By Authors

Fig.6 – Permission sent to MDPI Nanomaterials

Fig.7 – No copy rights issue – Made By Authors

Fig.8 – No copy rights issue – Made By Authors

Fig.9 – No copy rights issue – Made By Authors

Fig.10 – No copy rights issue – Open Access Source

Fig.11 – No copy rights issue – Made By Authors

Fig.12 – No copy rights issue – Made By Authors

Fig.13 – No copy rights issue – Open Access Source

Fig.14 – No copy rights issue – Open Access Source

Fig.15 – Permission sent to MDPI Nanomaterials

Fig.16 – No copy rights issue – Made By Authors

Fig.17 – No copy rights issue – Made By Authors

Fig.18 – No copy rights issue – Made By Authors

Fig.19 – No copy rights issue – Made By Authors

Fig.20 – No copy rights issue – Open Access Source

Fig.21 – No copy rights issue – Open Access Source

Fig.22 – No copy rights issue – Made By Authors

Fig.23 – No copy rights issue – Made By Authors

Fig.24 – No copy rights issue – Made By Authors

It will be more informative if a part of the paper covers the biofuels containing nanomaterials.

We thank the respected reviewer for the provided suggestion. Unfortunately, the authors background is scoped around heat transfer devices, and therefore we do not feel qualified to write about the use of biofuels that contains dispersed nanomaterials. However, we have included some info on the topic in our revised introduction section (kindly see Page 11, lines 146-150) with the following references (i.e., ref. 55 and 56) so that the respected reader can have a better insight on the topic from the experts in the biofuel field:

  • Khan, H., Soudagar, M. E. M., Kumar, R. H., Safaei, M. R., Farooq, M., Khidmatgar, A., & Taqui, S. N. (2020). Effect of nano-graphene oxide and n-butanol fuel additives blended with diesel—Nigella sativa biodiesel fuel emulsion on diesel engine characteristics. Symmetry, 12(6), 961.
  • Soudagar, M. E. M., Afzal, A., Safaei, M. R., Manokar, A. M., EL-Seesy, A. I., Mujtaba, M. A., & Goodarzi, M. (2020). Investigation on the effect of cottonseed oil blended with different percentages of octanol and suspended MWCNT nanoparticles on diesel engine characteristics. Journal of Thermal Analysis and Calorimetry, 1-18.

Thank you very much.   

Some references are older than 2015 and therefore, they are abolished. May the reviewer ask the authors to change these references to newer ones? Some suggested papers are as below: Heat transfer evaluation of a micro heat exchanger cooling with spherical carbon-acetone nanofluid. Effects on thermophysical properties of carbon based nanofluids: experimental data, modelling using regression, ANFIS and ANN.  Study of synthesis, stability and thermo-physical properties of graphene nanoplatelet/platinum hybrid nanofluid.  Investigation of heat transfer performance and friction factor of a counter-flow double-pipe heat exchanger using nitrogen-doped, graphene-based nanofluids.  A novel comprehensive experimental study concerned graphene oxide nanoparticles dispersed in water: Synthesise, characterisation, thermal conductivity measurement and present a new approach of RLSF neural network.

We Thank the respected reviewer for the suggested articles, which we totally agree that they will improve the quality of the in-hand manuscript. Kindly note that the recommended references have been included in the update version of the manuscript. Thank you very much.   

When reviewing the references, a strong impression can be created that the manuscript should be submitted to another journal: To give journal readers a sense of continuity, the reviewer encourages the authors to identify Nanomaterials publications of similar research in your papers. Please, do a literature check of the papers published in recent years (2020 and even 2021) and relate the content of relevant papers to the results and findings presented in your publication. The authors can also reference articles in print using their DOI.

We thank the respected reviewer for the respected remark. Kindly note that we have added the following references, which are related to our work and have been published recently in MDPI Nanomaterials.

  • You, X., & Li, S. (2021). Fully Developed Opposing Mixed Convection Flow in the Inclined Channel Filled with a Hybrid Nanofluid. Nanomaterials, 11(5), 1107.
  • Abed, N., Afgan, I., Iacovides, H., Cioncolini, A., Khurshid, I., & Nasser, A. (2021). Thermal-Hydraulic Analysis of Parabolic Trough Collectors Using Straight Conical Strip Inserts with Nanofluids. Nanomaterials, 11(4), 853.
  • Ambreen, T., Saleem, A., & Park, C. W. (2021). Homogeneous and Multiphase Analysis of Nanofluids Containing Nonspherical MWCNT and GNP Nanoparticles Considering the Influence of Interfacial Layering. Nanomaterials, 11(2), 277.
  • Giwa, S. O., Sharifpur, M., Ahmadi, M. H., Sohel Murshed, S. M., & Meyer, J. P. (2021). Experimental Investigation on Stability, Viscosity, and Electrical Conductivity of Water-Based Hybrid Nanofluid of MWCNT-Fe2O3. Nanomaterials11(1), 136.
  • Freitas, E., Pontes, P., Cautela, R., Bahadur, V., Miranda, J., Ribeiro, A. P., & Moita, A. S. (2021). Pool Boiling of Nanofluids on Biphilic Surfaces: An Experimental and Numerical Study. Nanomaterials, 11(1), 125.
  • Karagiannakis, N. P., Skouras, E. D., & Burganos, V. N. (2020). Modelling Thermal Conduction in Nanoparticle Aggregates in the Presence of Surfactants. Nanomaterials, 10(11), 2288.
  • Mannu, R., Karthikeyan, V., Velu, N., Arumugam, C., Roy, V. A., Gopalan, A. I., ... & Kannan, V. (2021). Polyethylene glycol coated magnetic nanoparticles: Hybrid nanofluid formulation, properties and drug delivery prospects. Nanomaterials11(2), 440.
  • Martínez-Merino, P., Sánchez-Coronilla, A., Alcántara, R., Martín, E. I., Carrillo-Berdugo, I., Gómez-Villarejo, R., & Navas, J. (2020). The role of the interactions at the tungsten disulphide surface in the stability and enhanced thermal properties of nanofluids with application in solar thermal energy. Nanomaterials, 10(5), 970.

Add some critical results to the introduction part.

- We thank the respected reviewer for his comment and positive feedback. Kindly note that we have taken the respected reviewer remark along with the other respected reviewers towards improving the introduction section. Thank you very much.

Refine the keywords.

- Keywords have been refined as suggested by the respected reviewer. Thank you.

Carbon-based nanomaterials have a vast application in enhancing the quality of biofuels. To know more about the topic, you study 2020 & 2021 related papers like: Investigation on the effect of cottonseed oil blended with different percentages of octanol and suspended MWCNT nanoparticles on diesel engine characteristics. Effect of nano-graphene oxide and n-butanol fuel additives blended with diesel—Nigella sativa biodiesel fuel emulsion on diesel engine characteristics.

We thank the respected reviewer for his suggestion. Unfortunately, the authors have no background on biofuel, and therefore we feel that we are not qualified to write about the use of biofuels that contains dispersed nanomaterials. However, we have included some info on the topic in our revised introduction section (kindly see Page 11, lines 146-150) with the suggested references (i.e., ref. 55 and 56) so that the respected reader can have a better insight on the topic from the experts in the biofuel field.

We do apologies for that and thank you very much for your valuable suggestion.

The conclusion part should be summarized.

Conclusion Section has been summarized as suggested by the respected reviewer and some of the detail texts were removed. Thank you.

Add a part to the article and talk about the environmental issues of carbon-based nanomaterials. You can see this paper for more information: Recent Advances in Using Nanofluids in Renewable Energy Systems and the Environmental Implications of their Uptake.

We thank the respected reviewer for his suggestion. We were pleased to take this suggestion into consideration in our revised manuscript. Kindly check Section 7 (Environmental Consideration and Potential Health Issues), which we have recently added to our manuscript. Thank you very much for your suggestion.

Do you really think Fig.2 is necessary for the manuscript?

Thank you for your question. We truly believe that this figure is important in many ways, such as to guide post graduates and new researchers in the field towards the pioneers in the field, potential reviewers, potential examiners/supervisors, and most importantly we need to give some credit to those whom are working persistently in order to enrich the nanofluid field. Therefore, we kindly ask the respected reviewer to allow us to maintain Fig. 2.

Thank you very much.

Reviewer 2 Report

This paper shows a good review of carbon-based nanofluids in heat transfer applications, there are some issues that need to address:

- Introduction is written simply, most recent research and innovation in nanofluids performances should be reviewed to show the gap of knowledge. Introduction should be extended with recently research papers.

- 499 References! Fantastic! Also, authors can cite the following work in the introduction which is closely related to their work and recently reported:

"Comprehensive study concerned graphene nano-sheets dispersed in ethylene glycol: Experimental study and theoretical prediction of thermal conductivity." Powder Technology 386 (2021): 51-59.

"A review on the control parameters of natural convection in different shaped cavities with and without nanofluid." Processes 8, no. 9 (2020): 1011.

what makes this review different from the others and from the most recent ones?

- section of drawbacks and future could be increased quality of the manuscript.

There are some grammatical errors, please carefully check the whole manuscript.

- The article size is very long; the additional description should be deleted and the additional tables should be merged.

Congratulations to the authors for their professional work. This article can be very useful for researchers and students.

Author Response

This paper shows a good review of carbon-based nanofluids in heat transfer applications, there are some issues that need to address:

- Introduction is written simply, most recent research and innovation in nanofluids performances should be reviewed to show the gap of knowledge. Introduction should be extended with recently research papers.

We thank the respected reviewer for his comment and positive feedback. Kindly note that we have taken the respected reviewer’s remarks along with the other respected reviewers towards improving the introduction section. Thank you very much.

- 499 References! Fantastic! Also, authors can cite the following work in the introduction which is closely related to their work and recently reported:

"Comprehensive study concerned graphene nano-sheets dispersed in ethylene glycol: Experimental study and theoretical prediction of thermal conductivity." Powder Technology 386 (2021): 51-59.

"A review on the control parameters of natural convection in different shaped cavities with and without nanofluid." Processes 8, no. 9 (2020): 1011.

We thank the respected reviewer for the suggestion. We do agree that the proposed two recent references would surely be an added value to our manuscript. As such, kindly note that we have included them in our revised version. Thank you very much.

What makes this review different from the others and from the most recent ones?

Thank you for your question. What makes this review different is that it guides the reader all the way from the carbon-based nanomaterials fabrication and all the way towards three major thermal applications in the industry and it provides the recent gap in the scientific knowledge, where the reader can tackle in his future research.

Section of drawbacks and future could be increased quality of the manuscript.

We thank the respected reviewer for his suggestion. We have taken this into consideration, but in an indirect way within our revised manuscript. Kindly note that we have included a stand-alone section, Section 7 (Environmental Consideration and Potential Health Issues), which is usually seen included as part of the drawbacks and future section, but we wanted it to be a stand-alone section for better deliverability purposes. Thank you very much

There are some grammatical errors, please carefully check the whole manuscript.

We apologise for any inconveniences. We have revised the English language of the whole manuscripts in its revised version. Thank you very much.

The article size is very long; the additional description should be deleted and the additional tables should be merged.

We thank the respected reviewer for his suggestion. The journal has not limit on the number of pages or the materials content. We also would like to be comprehensive about our literature survey. Therefore, we have found that adopting the detailed and informative form of a review article is very favourable by many readers in the field (based on the number of downloads and citations), and thus we tried to write our review article in a similar way to those published by elite scientific publishers. We kindly ask the respected reviewer to allow us to maintain the size of the manuscript. We do apologize for any inconveniences. Thank you very much for your highly respected suggestion.

Congratulations to the authors for their professional work. This article can be very useful for researchers and students.

The authors would like to thank the respected reviewer very much for his time and for his encouragements.

Thank you very much 

Round 2

Reviewer 1 Report

Accept as is

Author Response

Thank you very much.

The Authors

Reviewer 2 Report

The author has addressed my comments and the manuscript is now suitable for publication. 

Author Response

Thank you very much.

The Authors